# Estimating Causal Effects Identifiable from a Combination of Observations and Experiments

**Yonghan Jung**[1], **Iván Díaz**[2], **Jin Tian**[3], and **Elias Bareinboim**[4]

[1]Purdue University    jung222@purdue.edu
[2]New York University    ivan.diaz@nyu.edu
[3]Iowa State University    jtian@iastate.edu
[4]Columbia University    eb@cs.columbia.edu

## Abstract

Learning cause and effect relations is arguably one of the central challenges found throughout the data sciences. Formally, determining whether a collection of observational and interventional distributions can be combined to learn a target causal relation is known as the problem of *generalized identification* (or *g-identification*) [Lee et al., 2019]. Although g-identification has been well understood and solved in theory, it turns out to be challenging to apply these results in practice, in particular when considering the estimation of the target distribution from finite samples. In this paper, we develop a new, general estimator that exhibits multiply robustness properties for g-identifiable causal functionals. Specifically, we show that any g-identifiable causal effect can be expressed as a function of generalized multi-outcome sequential back-door adjustments that are amenable to estimation. We then construct a corresponding estimator for the g-identification expression that exhibits robustness properties to bias. We analyze the asymptotic convergence properties of the estimator. Finally, we illustrate the use of the proposed estimator in experimental studies. Simulation results corroborate the theory.

## 1 Introduction

Performing causal inferences is a crucial aspect of scientific research with broad applications ranging from the social sciences to economics, biology to medicine. It provides a set of principles and tools to draw causal conclusions from a combination of observations and experiments. Two significant tasks in the realization of these inferences are causal effect identification and estimation. *Causal effect identification* concerns determining the conditions under which one can infer the causal effect $P(Y = y|do(X = x))$ (shortly, $P(y|do(x))$) of the treatment $X = x$ on the outcome $Y = y$ from a combination of available data distributions and a causal graph depicting the data-generating process [Pearl, 2000, Bareinboim and Pearl, 2016]. *Causal effect estimation* aims to develop an estimator for the identified causal effect expression using a set of finite samples.

Recent advances in the literature on generalized causal effect identification (g-identification) have developed algorithms that can identify causal effects by using a set of observational and experimental distributions and a causal graph. The result is an expression of the causal effect as a function of available observational and experimental distributions [Bareinboim and Pearl, 2012, Lee et al., 2019]. For concreteness, consider some practical scenarios that exemplify g-identification.

**Example 1.** Many studies have investigated how a training program's eligibility ($X$) affects future salary ($Y$) (e.g.,[Glynn and Kashin, 2017]). Actual registration in the program ($Z$) determines the salary, and experimental studies have looked into how Z affects Y (e.g., [LaLonde, 1986]). Eligibility is determined by past average income ($W$), which is associated with both $Z$ and $Y$. The causal

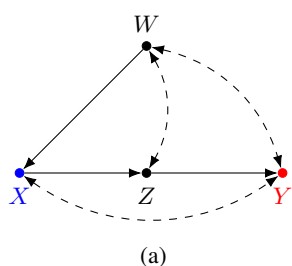
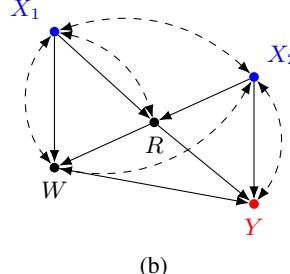

|(a)|(b)|

Figure 1: Causal graphs of examples 1 and 2. The nodes representing the treatment and the outcome are marked in blue and red, respectively.

graph in Fig. 1a shows the data-generating process, with bidirected edges indicating unmeasured confounders affecting the variables. According to Lee et al. [2019], the causal effect $P(y|do(x))$ can be identified by combining the experimental distribution on $Z$ (denoted $P(\cdot|do(z))$) with the observational distribution $P$. It's given as $P(y|do(x)) = \sum_{z,w} P(y|do(z))P(z|w,x)P(w)$. ∎

**Example 2.** There have been many experimental studies on the effect of an antihypertensive drug ($X_1$) on blood pressure ($W$) (e.g., Hansson et al. [1999]) and on the effect of using an anti-diabetic drug ($X_2$) on cardiovascular disease ($Y$) (e.g., Ajjan and Grant [2006], Kumar et al. [2016]). $R$ is a set of mediators. Their relations are depicted in Fig. 1b. Recent studies report that simultaneously taking antihypertensive and anti-diabetic drugs may be harmful [Ferrannini and Cushman, 2012]. This motivates the study of the combined causal effect of both treatments (i.e., $P(y|do(x_1, x_2))$) by combining the two experimental studies (i.e., from $P(\cdot|do(x_1))$ and $P(\cdot|do(x_2))$). According to Lee et al. [2019], it turns out that $P(y|do(x_1, x_2)) = \sum_{r,w} P(y|r, w, do(x_2))P(r|x_2, do(x_1)) \sum_{x'_2} P(w|r, x'_2, do(x_1))P(x'_2|do(x_1))$, which means that the joint treatment effects can be computed using the two experimental studies on $X_1$ and $X_2$. ∎

On the other hand, causal effect estimation has mainly focused on limited identification scenarios, relying on stringent assumptions such as the no unmeasured confounder assumption. Beyond these restrictions, recent progress has been made in developing statistically appealing estimators from observational data for any identification functional given by the complete identification algorithms [Jung et al., 2020a,b, 2021b,a, Bhattacharya et al., 2022, Xia et al., 2021]. While these estimators are capable of estimating any identification expression from observational data, they are not yet sufficiently advanced to estimate g-identification, which involves multiple observations and experiments.

Recently, Jung et al. [2023] generalized existing doubly robust estimators [Mises, 1947, Bickel et al., 1993, Robins and Rotnitzky, 1995, Bang and Robins, 2005, Robins et al., 2009, van der Laan and Gruber, 2012, Luedtke et al., 2017, Chernozhukov et al., 2018, Rotnitzky et al., 2021] to estimate covariate adjustments (e.g., back-door adjustment [Pearl, 1995], sequential back-door (SBD) adjustment [Pearl and Robins, 1995] or multi-outcome SBD (mSBD) [Jung et al., 2021b]) in the g-identification setting, where the expression is in the form of covariate adjustment involving multiple experimental distributions. However, the covariate adjustments only cover a limited portion of all g-identifiability scenarios as in Examples (1,2). On a different thread, Xia et al. [2023] developed a neural network-based estimation framework capable of taking a combination of observational/experimental data. Still, the derived estimators do not possess the doubly robustness property. In other words, there is still a gap between g-identification and causal effect estimation.

In this paper, our goal is to bridge the gap between g-identification and causal effect estimation. Specifically, this paper presents a framework for estimating identification expressions using multiple sets of samples from both observational and interventional distributions. This framework is a generalization of the results in Jung et al. [2021b] since our results reduce to theirs when only observational data is available. Furthermore, our work subsumes the results in Jung et al. [2023] when the identification functional takes the form of covariate adjustments.

The contributions of our paper are as follows:

1. We show that any causal effects identifiable by g-identification can be expressed as a function of generalized mSBD adjustments. We provide a systematic procedure for specifying the function.

2. After developing a doubly robust estimator for generalized mSBD adjustments, we construct
   an estimation framework for any g-identifiable causal effects, which enjoys multiply robustness
   against model misspecification and bias. Experimental studies corroborate our results.

## 1.1 Preliminaries

We use bold letters ($\mathbf{X}$) to denote a random vector and $X$ a random value. Each random vector is represented with a capital letter ($\mathbf{X}$) and its realized value with a small letter ($\mathbf{x}$). Given a set $\mathbf{X} = \{X_1, \cdots, X_n\}$ aligned by an order $\prec$ such that $X_i \prec X_j$ for $i < j$, we denote $\overline{\mathbf{X}}^i := \{X_1, \cdots, X_i\}$ and $\overline{\mathbf{X}}^{i:j} := \{X_i, \cdots, X_j\}$. For a discrete vector $\mathbf{X}$, we use $\mathbb{1}_{\mathbf{x}}(\mathbf{X})$ to represent the indicator function such that $\mathbb{1}_{\mathbf{x}}(\mathbf{X}) = 1$ if $\mathbf{X} = \mathbf{x}$; $\mathbb{1}_{\mathbf{x}}(\mathbf{X}) = 0$ otherwise. We use $[n] := \{1, \cdots, n\}$ a collection of index. For a discrete vector $\mathbf{V}$, we use $P(\mathbf{v}) := P(\mathbf{V} = \mathbf{v})$ where $P$ is a distribution. We use $\mathbb{E}_P[f(\mathbf{V})] := \sum_{\mathbf{v} \in \mathfrak{S}_{\mathbf{V}}} f(\mathbf{v})P(\mathbf{v})$ for a function $f$, where $\mathfrak{S}_{\mathbf{V}}$ denote the support of $\mathbf{V}$. We will use $\mathfrak{D}_{\mathbf{V}}$ to denote the domain of $\mathbf{V}$. For a sample set $D := \{\mathbf{V}_{(i)}\}_{i=1}^n$ where $\mathbf{V}_{(i)}$ denotes the $i$th samples, we use $\mathbb{E}_D[f(\mathbf{V})] := (1/n)\sum_{i=1}^n f(\mathbf{V}_{(i)})$. We use $\|f\|_P := \sqrt{\mathbb{E}_P[\{f(\mathbf{V})\}^2]}$. If a function $\hat{f}$ is a consistent estimator of $f$ having a rate $r_n$, we will use $\hat{f} - f = o_P(r_n)$. We will say $\hat{f}$ is $L_2$-consistent if $\|\hat{f} - f\|_P = o_P(1)$. We will use $\hat{f} - f = O_P(1)$ if $\hat{f} - f$ is bounded in probability. Also, $\hat{f} - f$ is said to be bounded in probability at rate $r_n$ if $\hat{f} - f = O_P(r_n)$. We use the typical graph terminology $pa(\mathbf{C})_G, ch(\mathbf{C})_G, de(\mathbf{C})_G, an(\mathbf{C})_G$ to represent the union of $\mathbf{C}$ with its parents, children, descendants, ancestors in the graph $G$. We use $\text{pre}(\mathbf{C}; G)$ to denote the union of the predecessors of $C_i \in \mathbf{C}$ given a topological order $\prec_G$ over a graph $G$. We use $G(\mathbf{C})$ to denote the subgraph of $G$ over $\mathbf{C}$. Throughout the paper, we will assume a fixed topological order $\prec_G$ over $\mathbf{V}$ on $G$. ∎

**Structural Causal Models (SCMs).** We use Structural Causal Models (SCMs) as our framework [Pearl, 2000, Bareinboim et al., 2022]. An SCM $\mathcal{M}$ is a quadruple $\mathcal{M} = \langle \mathbf{U}, \mathbf{V}, P(\mathbf{U}), F \rangle$. $\mathbf{U}$ is a set of exogenous (latent) variables following a joint distribution $P(\mathbf{U})$. $\mathbf{V}$ is a set of endogenous (observable) variables whose values are determined by functions $F = \{f_{V_i}\}_{V_i \in \mathbf{v}}$ such that $V_i \leftarrow f_{V_i}(pa_i, u_i)$ where $PA_i \subseteq V$ and $U_i \subseteq U$. Each SCM $\mathcal{M}$ induces a distribution $P(\mathbf{V})$ and a causal graph $G = G(\mathcal{M})$ over $\mathbf{V}$ in which there exists a directed edge from every variable in $PA_i$ to $V_i$ and dashed-bidirected arrows encode common latent variables (e.g., see Fig. 1a). Performing an intervention fixing $\mathbf{X} = \mathbf{x}$ is represented through the do-operator, $do(\mathbf{X} = \mathbf{x})$, which encodes the operation of replacing the original equations of $X$ (i.e., $f_X(pa_x, u_x)$) by the constant $x$ for all $X \in \mathbf{X}$ and induces an interventional distribution $P(\mathbf{V}|do(\mathbf{x}))$. ∎

**Experimental Distributions and Samples** To clarify the connection between the experimental samples where the randomization is applied to $\mathbf{Z} \subseteq \mathbf{V}$ and the distribution $P_{\mathbf{z}}(\mathbf{V} \backslash \mathbf{z})$, we introduce the notation $P_{\sigma(\mathbf{Z})}(\mathbf{V})$ where $\sigma(\mathbf{Z})$ denotes that $\mathbf{Z}$ is randomized. The distribution $P_{\sigma(\mathbf{Z})}(\mathbf{V})$ is a distribution induced by the SCM in which the original equation $Z \leftarrow f_Z(pa_z, u_z)$ for $Z \in \mathbf{Z}$ is replaced to the function assigning the value to $Z = z$ at random without depending on other endogenous variables $PA_Z$; e.g., $Z = 1$ and 0 at probability 0.5 for each. We note that $P := P_{\sigma(\emptyset)}$ when observational. For any set $\mathbf{A}, \mathbf{B}, \mathbf{Z} \subseteq \mathbf{V}$, the interventional distribution can be represented as $P(\mathbf{A}|do(\mathbf{z}), \mathbf{B}) = P_{\sigma(\mathbf{Z})}(\mathbf{A}|\mathbf{Z} = \mathbf{z}, \mathbf{B})$ by the definition of the $do$-operator and $P_{\sigma(\mathbf{Z})}$ distribution. We use $P_{\mathbf{z}}(\mathbf{A}|\mathbf{B}) := P_{\sigma(\mathbf{Z})}(\mathbf{A}|\mathbf{Z} = \mathbf{z}, \mathbf{B})$ to highlight that the distribution is induced from the randomization and conditioning on $\mathbf{Z} = \mathbf{z}$. The experimental samples from randomization $\sigma(\mathbf{Z})$ induces samples $D_{\sigma(\mathbf{Z})}$ following $P_{\sigma(\mathbf{Z})}(\mathbf{V})$. We use $D_{\mathbf{z}}$ to denote the subsample of $D_{\sigma(\mathbf{Z})}$ fixing $\mathbf{Z} = \mathbf{z}$, which follows $P_{\mathbf{z}}(\mathbf{V})$. ∎

**g-identification.** Let $\mathbb{Z} := \{\mathbf{Z}_i\}_{i=1}^m$ denote a collection of variables where $\mathbf{Z}_i$ can be an empty set. Let $\mathbb{P} := \{P_{\sigma(\mathbf{Z}_i)}(\mathbf{V}), \mathbf{Z}_i \in \mathbb{Z}\}$, a collection of distributions inducing experimental samples from trials randomizing $\mathbf{Z}_i \in \mathbb{Z}$. A causal effect $P(\mathbf{y}|do(\mathbf{x}))$ is said to be *g-identifiable* from $\mathbb{P}$ in a causal graph $G$ if $P(\mathbf{y}|do(\mathbf{x}))$ is uniquely computable from the combination of distributions in $\mathbb{P}$ in any SCM that induces $G$ [Lee et al., 2019, Def. 4]. The complete g-identification algorithm developed by Lee et al. [2019] identifies the causal effect by decomposing so-called *confounded components* (c-component). A *c-component* is a maximal set of variables where every pair is connected by a bidirectional path composed of bidirectional edges ($V_i \leftrightarrow V_j$). For example, graphs in Figs. (1a, 1b) form a single c-component since bidirectional paths connect any pairs of variables. For any sets $\mathbf{C} \subseteq \mathbf{V}$, the quantity $Q[\mathbf{C}] := P(\mathbf{c}|do(\mathbf{v} \backslash \mathbf{c}))$ is called a *c-factor*. To identify the causal effect

$P(\mathbf{y}|do(\mathbf{x}))$ from $\mathbb{P}$ and $G$, the g-identification algorithm in [Lee et al., 2019, Algo. 1] (and rewrote in Algo. 1) rewrites the causal effect as a marginalization over a product of c-factors, $P(\mathbf{y}|do(\mathbf{x})) = \sum_{\mathbf{d}\backslash\mathbf{y}\in\mathfrak{S}_{\mathbf{D}\backslash\mathbf{Y}}}\prod_{i=1}^{k_d}Q[\mathbf{D}_i]$, where $\mathbf{D} \coloneqq an(\mathbf{Y})_{G(\mathbf{V}\backslash\mathbf{X})}$ and $\mathbf{D}_i$ are c-components in $G(\mathbf{D})$, and identifies each $Q[\mathbf{D}_i]$ from $\mathbb{P}$. ∎

## 1.2 Problem Statement

This paper aims to develop an estimation framework for the g-identifiable causal effect $P(\mathbf{y}|do(\mathbf{x}))$ identified as a function of distributions in $\mathbb{P}$ from experimental samples $\mathbb{D} \coloneqq \{D_{\mathbf{Z}_i}\sim P_{\sigma(\mathbf{Z}_i)}(\mathbf{V})\in\mathbb{P}\}$. We impose the following regularity assumptions:

**Assumption 1** (**Regularity**). *For variables $\mathbf{V}$ and distributions $P_{\sigma(\mathbf{Z})}\in\mathbb{P}$, the following conditions hold: (1) All variables in $\mathbf{V}$ are discrete; (2) $P_{\sigma(\mathbf{Z})}(\mathbf{v}) > c, \forall \mathbf{v}\in\mathfrak{D}_{\mathbf{V}}$ for some $c\in(0,1)$.*

We discuss the relaxation of the regularity assumption in Appendix C. This relaxation allows some subset of variables in $\mathbf{V}$ can be a mixture of continuous and discrete random variables. Due to space constraints, all proofs are provided in Appendix B.

## 2 Expressing Causal Effects as a Combination of mSBD Adjustments

In this section, we present an algorithm that expresses any g-identifiable causal effects as a combination of *marginalization/multiplication/divisions* of adjustment functionals defined in the following. We begin by formally defining the generalized multi-outcome sequential back-door adjustment (g-mSBD) functional, which strictly generalizes the mSBD adjustment proposed by Jung et al. [2021b]:

**Definition 1** (**generalized-mSBD adjustment (g-mSBD)**). Let $(\mathbf{W},\mathbf{R})$ be a disjoint pair in $\mathbf{V}$ topologically ordered as $(\mathbf{W},\mathbf{R}) = \{\mathbf{R}_0,W_1,\cdots,\mathbf{R}_{m-1},W_m,\mathbf{R}_m\}$ by $\prec_G$, where $\mathbf{R}_i$ can be empty. Let $\overline{\mathbf{W}}^{i-1} \coloneqq \{W_j\}_{j=1}^{i-1}$ and $\overline{\mathbf{R}}^{i-1} \coloneqq \{\mathbf{R}_j\}_{j=0}^{i-1}$ for $\forall i\in[m]$. Let $\mathbf{C}\subseteq\mathbf{W}$. Let $\mathbb{Z}_0\subseteq\mathbb{Z}$ be some set such that $\forall\mathbf{Z}\in\mathbb{Z}_0,\mathbf{W}\cap\mathbf{Z}=\emptyset$. Let $\texttt{seq}(\mathbb{Z}_0)$ denote a sequence $(\mathbf{z}_1,\cdots,\mathbf{z}_m)$ where $\mathbf{z}_i$ denotes some realization of $\mathbf{Z}_i\in\mathbb{Z}_0$ (same $\mathbf{z}_i$ could appear multiple times in the sequence). Then, the g-mSBD adjustment is expressed as an operator $A_0[\mathbf{W},\mathbf{C},\mathbf{R};\mathbb{Z}_0,\texttt{seq},G](\mathbf{w}\backslash\mathbf{c},\mathbf{r})$ defined by

$$A_0[\mathbf{W},\mathbf{C},\mathbf{R};\mathbb{Z}_0,\texttt{seq}](\mathbf{w}\backslash\mathbf{c},\mathbf{r}) \coloneqq \sum_{\mathbf{c}\in\mathfrak{S}_{\mathbf{C}}}\prod_{i:W_i\in\mathbf{W}}P_{\mathbf{z}_i}(w_i|\overline{\mathbf{w}}^{i-1},\overline{\mathbf{r}}^{i-1}\backslash\mathbf{z}_i). \tag{1}$$

The g-mSBD adjustment specializes to the mSBD adjustment [Jung et al., 2021b] when $\mathbb{Z}_0=\emptyset$. The g-mSBD adjustment can be viewed as a variant of the g-formula [Robins, 1986] involving multiple distributions. The power of the g-mSBD adjustment lies in its ability to express the c-factor:

**Lemma 1** (**c-component Identification** [Jung et al., 2021b]). *Let $\mathbf{S}$ denote a c-component in $G_i \coloneqq G(\mathbf{V}\backslash\mathbf{Z}_i)$ for some $\mathbf{Z}_i\in\mathbb{Z}$. Let $\mathbf{R} \coloneqq pa(\mathbf{S})_{G_i}\backslash\mathbf{S}$. Let $(\mathbf{S},\mathbf{R})$ be ordered as $(\mathbf{R}_0,S_1,\cdots,\mathbf{R}_{m-1},S_m)$ by $\prec_G$. Let $\mathbf{A}\subseteq\mathbf{S}$ denote a set satisfying $\mathbf{A}=an(\mathbf{A})_{G_i(\mathbf{S})}$. Let $\mathbf{C}\coloneqq(\mathbf{S}\backslash\mathbf{A})$. Let $\mathbb{Z}_0\coloneqq\{\mathbf{Z}_i\}$ and $\texttt{seq}(\mathbb{Z}_0)$ be a sequence of $\mathbf{z}_i$ repeating $m$ times. Then, the c-factor $Q[\mathbf{A}]$ is g-identifiable as follows:*

$$Q[\mathbf{A}] = A_0[\mathbf{S},\mathbf{C},\mathbf{R};\mathbb{Z}_0\coloneqq\{\mathbf{Z}_i\},\texttt{seq}](\mathbf{a},\mathbf{r}) = \sum_{\mathbf{c}\in\mathfrak{S}_{\mathbf{C}}}\prod_{j:V_j\in\mathbf{S}}P_{\mathbf{z}_i}(v_j|\overline{\mathbf{s}}^{j-1},\overline{\mathbf{r}}^{j-1}\backslash\mathbf{z}_i). \tag{2}$$

We propose an identification algorithm, Algo. 1, which expresses any causal effect as a combination of marginalizations, multiplications, and divisions of g-mSBD operators. Here are some results used for the g-mSBD operation. An example of using these results is provided in Appendix A.

**Lemma 2** (**Marginalization**). *Let $A_0[\mathbf{W},\mathbf{C},\mathbf{R};\mathbb{Z}_0,\texttt{seq}](\mathbf{w}\backslash\mathbf{c},\mathbf{r})$ denote the g-mSBD operator in Def. 1. Let $\mathbf{W}_0\subseteq\mathbf{W}\backslash\mathbf{C}$. Let $\mathbf{W}_{mar}\subseteq\{\mathbf{W}_0,\mathbf{C}\}$ denote the vector formed by the following procedure: Starting from $\mathbf{W}_{mar}=\emptyset$, for $j=m,\cdots,1$, $\mathbf{W}_{mar}=\mathbf{W}_{mar}\cup\{W_j\}$ if (1) $W_j\in\{\mathbf{W}_0,\mathbf{C}\}$ and (2) $\exists k\in\{j,\cdots,m\}$ such that $\mathbf{R}_j,\cdots,\mathbf{R}_{k-1}=\emptyset$, $\overline{\mathbf{W}}^{k+1:m}\subseteq\mathbf{W}_{mar}$, and $\mathbf{Z}_k=\cdots=\mathbf{Z}_j$ and $\mathbf{z}_k=\cdots=\mathbf{z}_j$. Let $\mathbf{W}'\coloneqq\mathbf{W}\backslash\mathbf{W}_{mar}$, $\mathbf{R}'\coloneqq pre(\mathbf{W}';G)\cap\mathbf{R}$ and $\mathbf{C}'\coloneqq\{\mathbf{W}_0,\mathbf{C}\}\backslash\mathbf{W}_{mar}$. Let $\mathbb{Z}'\subseteq\mathbb{Z}_0$ denote the collection of $\mathbf{Z}_i$ corresponding to the variable in $\mathbf{W}'$, and $\texttt{seq}'$ the corresponding sequence. Then,*

$$\sum_{\mathbf{w}_0\in\mathfrak{S}_{W_0}}A_0[\mathbf{W},\mathbf{C},\mathbf{R};\mathbb{Z}_0,\texttt{seq}](\mathbf{w}\backslash\mathbf{c},\mathbf{r}) = A_0[\mathbf{W}',\mathbf{C}',\mathbf{R}';\mathbb{Z}',\texttt{seq}'](\mathbf{w}'\backslash\mathbf{c}',\mathbf{r}'). \tag{3}$$

**Algorithm 1:** GID $(\mathbf{x}, \mathbf{y}, \mathbb{Z}, \mathbb{P}, G)$

**Input:** $\mathbf{x}, \mathbf{y}, \mathbb{Z} \coloneqq \{\mathbf{Z}_i\}, \mathbb{P} \coloneqq \{P_{\sigma(\mathbf{Z}_i)}(\mathbf{V}), \ \forall \mathbf{Z}_i \in \mathbb{Z}\}, G$
**Output:** Expression of $P(\mathbf{y}|do(\mathbf{x}))$ w.r.t. distributions in $\mathbb{P}$

1   **If** $\exists \mathbf{Z}_i \in \mathbb{Z}$ such that $P(\mathbf{y}|do(\mathbf{x})) = P_{\mathbf{z}_i}(\mathbf{y})$ for some $\mathbf{z}_i \in \mathfrak{D}_{\mathbf{Z}_i}$, **then return** $P_{\mathbf{z}_i}(\mathbf{y})$.
2   Let $\mathbf{V} \leftarrow an(\mathbf{Y}); P(\mathbf{v}) \leftarrow P(an(\mathbf{Y}));$ and $G \leftarrow G(an(\mathbf{Y}))$.
3   Let $\mathbf{D} \coloneqq an(\mathbf{Y})_{G(\mathbf{V}\setminus\mathbf{X})}$.
4   Find the $C$-component of $G(\mathbf{D})$: $\mathbf{D}_1, \cdots \mathbf{D}_{k_d}$.
5   **foreach** $\mathbf{D}_j \in \{\mathbf{D}_1, \cdots \mathbf{D}_{k_d}\}$ **do**
6     **foreach** $\mathbf{Z}_i \in \mathbb{Z}$ **do**
7       Find the c-component $\mathbf{S}_j^i$ in $G(\mathbf{V}\setminus\mathbf{Z}_i)$ such that $\mathbf{D}_j \subseteq \mathbf{S}_j^i$.
8       $Q[\mathbf{S}_j^i] = A_0[\mathbf{S}_j^i, \emptyset, \mathbf{R}_j^i; \mathbb{Z}_j^i \coloneqq \{\mathbf{Z}_i\}, \mathtt{seq}_j^i](\mathbf{s}_j^i, \mathbf{r}_j^i)$, where $\mathbf{R}_j^i \coloneqq pa(\mathbf{S}_j^i)_{G(\mathbf{V}\setminus\mathbf{Z}_i)}\setminus\mathbf{S}_j^i$. //
      By Lemma 1
9       Run $Q[\mathbf{D}_j] = \textsc{subID}(\mathbf{D}_j, \mathbf{S}_j^i, Q[\mathbf{S}_j^i], G(\mathbf{S}_j^i))$.
10      **If** $Q[\mathbf{D}_j] \neq$ FAIL, **then break**.
11    **end**
12    **If** $Q[\mathbf{D}_j] =$ FAIL, **then return** FAIL.
13   **end**
14   $P(\mathbf{y}|do(\mathbf{x})) = \sum_{\mathbf{d}\setminus\mathbf{y}\in\mathfrak{S}_{\mathbf{D}\setminus\mathbf{Y}}} \prod_{j=1}^{k_d} Q[\mathbf{D}_j]$. // Apply Lemmas (2,3,4) if viable
15   **return** $P(\mathbf{y}|do(\mathbf{x}))$
a.1   **Procedure** $\textsc{subID}(\mathbf{C}, \mathbf{T}, Q[\mathbf{T}], G(\mathbf{T}))$
a.2    Let $\mathbf{A} \coloneqq an(\mathbf{C})_{G(\mathbf{T})} = \{A_1, A_2, \cdots, A_{n_a}\}$ such that $A_1 \prec_G \cdots \prec_G A_{n_a}$ in $G(\mathbf{T})$.
a.3    Let $Q[\mathbf{A}] = \sum_{\mathbf{t}\setminus\mathbf{a}\in\mathfrak{S}_{\mathbf{T}\setminus\mathbf{A}}} Q[\mathbf{T}]$. // Apply Lemma 2 if viable
a.4    **If** $\mathbf{A} = \mathbf{C}$, **then return** $Q[\mathbf{A}]$.
a.5    **If** $\mathbf{A} = \mathbf{T}$, **then return** FAIL.
a.6    **else**
a.7     Let $\mathbf{S}$ be the c-component in $G(\mathbf{A})$ such that $\mathbf{C} \subseteq \mathbf{S}$.
a.8     Let $Q[\mathbf{S}] \coloneqq \prod_{\{i:A_i\in\mathbf{S}\}} \frac{\sum_{\mathbf{b}_{i+1}\in\mathfrak{S}_{\mathbf{B}_{i+1}}} Q[\mathbf{A}]}{\sum_{\mathbf{b}_i\in\mathfrak{S}_{\mathbf{B}_i}} Q[\mathbf{A}]}$ for $\mathbf{B}_i \coloneqq \mathbf{A}\setminus\overline{\mathbf{A}}^{i-1}$. // Apply
    Lemmas (2,3,4) if viable
a.9     **return** $\textsc{subID}(\mathbf{C}, \mathbf{S}, Q[\mathbf{S}], G(\mathbf{S}))$
a.10   **end**

---

This lemma provides a graphical criterion where $\sum_{\mathbf{w}_0 \in \mathfrak{S}_{W_0}} A_0[\mathbf{W}, \mathbf{C}, \mathbf{R}; \mathbb{Z}_0, \mathtt{seq}](\mathbf{w}\setminus\mathbf{c}, \mathbf{r})$ is given as a g-mSBD operator.

**Lemma 3 (Multiplication).** *Let* $A_0^i \coloneqq A_0[\mathbf{W}_i, \emptyset, \mathbf{R}_i; \mathbb{Z}_i, \mathtt{seq}^i](\mathbf{w}_i, \mathbf{r}_i) \coloneqq \prod_{j=1}^{m^i} P_{\mathbf{z}_j^i}(w_{i,j}|\overline{\mathbf{w}}_i^{j-1}, \overline{\mathbf{r}}_i^{j-1}\setminus\mathbf{z}_j^i)$ *for* $i \in \{1,2\}$ *where* $\mathtt{seq}^i \coloneqq (\mathbf{z}_j^i)_{j=1}^{m^i}$. *Let* $\mathbf{W} \coloneqq \mathbf{W}_1 \cup \mathbf{W}_2$. *Let* $\mathbf{R} \coloneqq (\mathbf{R}_1 \cup \mathbf{R}_2)\setminus\mathbf{W}$. *Let* $(\mathbf{W}, \mathbf{R})$ *be ordered by* $\prec_G$. *Let* $\mathbb{Z} \coloneqq \mathbb{Z}_1 \cup \mathbb{Z}_2$. *Assume the following: (1)* $\mathbf{W}_1 \cap \mathbf{W}_2 = \emptyset$; *and (2)* $\forall W_j \in \mathbf{W}, \exists W_{i,k} \in \mathbf{W}_i$ *such that* $(\overline{\mathbf{W}}^{j-1}, \overline{\mathbf{R}}^{j-1}) = (\overline{\mathbf{W}}_i^{k-1}, \overline{\mathbf{R}}_i^{k-1})$. *Let* $\mathtt{seq} \coloneqq (\mathbf{z}_j)_{j:W_j\in\mathbf{W}}$ *where* $\mathbf{z}_j = \mathbf{z}_k^i$ *for all* $j$. *Then,*

$$A_0^1 \times A_0^2 = A_0[\mathbf{W}, \emptyset, \mathbf{R}; \mathbb{Z}, \mathtt{seq}](\mathbf{w}, \mathbf{r}) = \prod_{j:W_j\in\mathbf{W}} P_{\mathbf{z}_j}(w_j|\overline{\mathbf{w}}^{j-1}, \overline{\mathbf{r}}^{j-1}\setminus\mathbf{z}_j). \quad (4)$$

This lemma provides a graphical criterion where a product $A_0^1 \times A_0^2$ is given as a g-mSBD operator.

**Lemma 4 (Division).** *Let* $A_0^i \coloneqq A_0[\mathbf{W}_i, \emptyset, \mathbf{R}_i; \mathbb{Z}_i, \mathtt{seq}^i](\mathbf{w}_i, \mathbf{r}_i) \coloneqq \prod_{j=1}^{m^i} P_{\mathbf{z}_j^i}(w_{i,j}|\overline{\mathbf{w}}_i^{j-1}, \overline{\mathbf{r}}_i^{j-1}\setminus\mathbf{z}_j^i)$ *for* $i \in \{1,2\}$ *where* $\mathtt{seq}^i \coloneqq (\mathbf{z}_j^i)_{j=1}^{m^i}$. *Let* $\mathbf{W} \coloneqq \mathbf{W}_1\setminus\mathbf{W}_2$. *Let* $\mathbf{R} \coloneqq (\mathbf{R}_1 \cup \mathbf{W}_2) \cap pre(\mathbf{W}; G)$. *Assume the following: (1)* $\mathbf{W}_2 \subseteq \mathbf{W}_1$; *and (2)* $\forall W_j \in \mathbf{W}, \exists W_{1,k} \in \mathbf{W}_1$ *such that* $(\overline{\mathbf{W}}^{j-1}, \overline{\mathbf{R}}^{j-1}) = (\overline{\mathbf{W}}_1^{k-1}, \overline{\mathbf{R}}_1^{k-1})$, $\mathbf{Z}_{i,k} = \mathbf{Z}_j$ *and* $\mathbf{z}_{i,k} = \mathbf{z}_j$. *Then,*

$$A_0^1/A_0^2 = A_0[\mathbf{W}, \emptyset, \mathbf{R}; \mathbb{Z}_1, \mathtt{seq}^1](\mathbf{w}, \mathbf{r}) = \prod_{j:W_j\in\mathbf{W}} P_{\mathbf{z}_j}(w_j|\overline{\mathbf{w}}^{j-1}, \overline{\mathbf{r}}^{j-1}\setminus\mathbf{z}_j). \quad (5)$$

This lemma provides a graphical criterion where a product $A_0^1/A_0^2$ is given as a g-mSBD operator.

We have rewritten the identification algorithm proposed by Lee et al. [2019] as Algo. 1 to express the g-identifiable causal effect as a combination of marginalizations, multiplications, and divisions of g-mSBD. It's worth of noting that the identification algorithm proposed by Lee et al. [2019] and Algo. 1 are equivalent.

**Theorem 1** (**Expression of g-Identifiable Causal Effects**). *Algo. 1 returns any g-identifiable causal effects as a function of a set $\{A_0^k\}$ of g-mSBD adjustment operators in the form*

$$P(\mathbf{y}|do(\mathbf{x})) = f(\{A_0^k\}_{k=1}^K), \tag{6}$$

*where the function $f(\cdot)$ applies marginalization, multiplication, or division over g-mSBD operators in $\{A_0^k\}$ as specified by Algo. 1.*

For concreteness, we demonstrate the application of Algo. 1 for Figs. (1a,1b), where the effects $P(\mathbf{y}|do(\mathbf{x}))$ are g-identifiable. Detailed and visually friendly demonstrations are described in Appendix A.

**Example 3** (**Application of Algo. 1 to Example 1**). Note $\mathbb{Z} = \{\emptyset, Z\}$. **Line 3-4:** $\mathbf{D} = \{Z, Y\}$ where $\mathbf{D}_1 \coloneqq \{Z\}$ and $\mathbf{D}_2 \coloneqq \{Y\}$. **Line 5-13:** Identify $Q[\mathbf{D}_1]$ from $\mathbf{Z}_1 = \emptyset$ as follow. Note $\mathbf{D}_1 \subseteq \mathbf{S}^0$, where $\mathbf{S}^0 \coloneqq \mathbf{V}$ where $Q[\mathbf{S}^0] = A_0[\mathbf{S}^0, \emptyset, \emptyset; \mathbb{Z}^0 \coloneqq \emptyset, \emptyset](\mathbf{s}^0, \emptyset) = P(\mathbf{v})$. Run $Q[\mathbf{D}_1] = \text{SUBID}(\mathbf{D}_1, \mathbf{S}^0, Q[\mathbf{S}^0], G)$ and obtain $Q[\mathbf{D}_1] = A_0^1 \coloneqq A_0[\{W, Z\}, W, X; \emptyset, \emptyset](z, x) = \sum_{w \in \mathfrak{S}_W} P(z|x, w)P(w)$. Lemmas (2,3,4) are used in running the sub-procedure. Now, identify $Q[\mathbf{D}_2]$ from $\mathbf{Z}_2 = \{Z\}$ as follow. Note $\mathbf{D}_2 \subseteq \mathbf{S}^1 \coloneqq \{W, X, Y\}$, the c-component in $G(\mathbf{V}\backslash Z)$. Note $Q[\mathbf{S}^1] = A_0[\mathbf{S}^1, \emptyset, \emptyset; \mathbb{Z}^1 \coloneqq \{Z\}, \text{seq}^1](\mathbf{s}^1, \emptyset) = P_z(w, x, y)$, where $\text{seq}^1 = (z, z, z)$. We run $Q[\mathbf{D}_2] = \text{SUBID}(\mathbf{D}_2, \mathbf{S}^1, Q[\mathbf{S}^1], G(\mathbf{S}^1))$, and obtain $Q[\mathbf{D}_2] = A_0^2 \coloneqq A_0[Y, \emptyset, \emptyset; \mathbb{Z}^1, \text{seq}^1](y, \emptyset) = P_z(y)$. Lemma 2 is used in the sub-procedure. **Line 14-15:** $P(y|do(x)) = \sum_{z \in \mathfrak{S}_z} A_0^1 A_0^2$. ∎

**Example 4** (**Application of Algo. 1 to Example 2**). Note $\mathbb{Z} = \{X_1, X_2\}$. **Line 3-4:** $\mathbf{D} = \{R, W, Y\}$ where $\mathbf{D}_1 \coloneqq \{R\}$, $\mathbf{D}_2 \coloneqq \{W\}$, and $\mathbf{D}_3 \coloneqq \{Y\}$. **Line 5-13:** In $G(\mathbf{V}\backslash X_1)$, $\mathbf{D}_1 = \mathbf{S}_1^1 \coloneqq \{R\}$. $Q[\mathbf{D}_1] = Q[\mathbf{S}_1^1] = A_0^1 \coloneqq A_0[R, \emptyset, X_2; \mathbb{Z}^1 \coloneqq \{X_1\}, \text{seq}^1](r, x_2) = P(r|do(x_1), x_2)$, where $\text{seq}^1 = (x_1)$. In $G(\mathbf{V}\backslash X_1)$, $\mathbf{D}_2 \subseteq \mathbf{S}_2^1 \coloneqq \{X_2, W, Y\}$. $Q[\mathbf{S}_2^1] = A_0[\mathbf{S}_2^1, \emptyset, R; \mathbb{Z}^2 \coloneqq \{X_1\}, \text{seq}^2](\mathbf{s}_2^1, r) = P_{x_1}(x_2)P_{x_1}(w|x_2, r,)P_{x_1}(y|x_2, w, r)$ where $\text{seq}^2 = (x_1, x_1, x_1)$. Run $Q[\mathbf{D}_2] = \text{SUBID}(\mathbf{D}_2, \mathbf{S}_2^1, Q[\mathbf{S}_2^1], G(\mathbf{S}_2^1)) = A_0^2 \coloneqq A_0[\{X_2, W\}, X_2, R; \mathbb{Z}^2, \text{seq}^2](w, r) = \sum_{x_2' \in \mathfrak{S}_{X_2}} P_{x_1}(w|r, x_2',)P_{x_1}(x_2')$ where $\text{seq}^2 = (x_1, x_1)$. Lemma 2 is used in the sub-procedure. Since $\text{SUBID}(\mathbf{D}_3, \mathbf{S}_2^1, Q[\mathbf{S}_2^1], G(\mathbf{S}_2^1))$ return FAIL, we find the c-component $\mathbf{S}_1^2 \coloneqq \{Y\}$ where $\mathbf{D}_3 = \mathbf{S}_1^2$. Note $Q[\mathbf{D}_3] = Q[\mathbf{S}_1^2] = A_0^3 \coloneqq A_0[Y, \emptyset, \{R, W\}; \mathbb{Z}^3 \coloneqq \{X_2\}, \text{seq}^3](y, \{r, w\}) = P_{x_2}(y|w, r)$, where $\text{seq}^3 = (x_2)$. **Line 14-15:** Applying Lemma 3, $A_0^{13} \coloneqq A_0^1 \times A_0^3 = A_0[\{R, Y\}, \emptyset, \{X_2, W\}; \mathbb{Z}^{13} \coloneqq \{X_1, X_2\}, \text{seq}^{13}](\{r, y\}, \{x_2, w\}) = P_{x_1}(r|x_2,)P_{x_2}(y|r, w)$, where $\text{seq}^{13} = (x_1, x_2)$. Then, $P(y|do(x_1, x_2)) = \sum_{r, w \in \mathfrak{S}_{R, W}} A_0^{13} A_0^2$. ∎

## 3 Estimating g-Identifiable Causal Effects

In this section, we develop an estimator for $P(\mathbf{y}|do(\mathbf{x}))$ using samples $\mathbb{D} \coloneqq \{D_{\sigma(\mathbf{Z}_i)} \sim P_{\sigma(\mathbf{Z}_i)}(\mathbf{V}) \in \mathbb{P}\}$ obtained from randomized experiments and observations (where $\mathbf{Z}_i = \emptyset$). We use $P_{\sigma(\mathbf{Z})}$ instead of $P_{\mathbf{z}}$ to highlight the distribution $D_{\sigma(\mathbf{Z}_i)} \in \mathbb{D}$ follows.

We first introduce an estimator for the g-mSBD adjustment that exhibits the doubly robust property. The nuisance parameters for the g-mSBD adjustment are defined as follows:

**Definition 2** (**Nuisances for g-mSBD**). *Nuisances for g-mSBD $A_0$ in Eq. (1) are $\{\mu_0^{i+1}, \pi_0^i\}_{i=1}^{m-1}$ defined as follows. Let $\mu_0^{m+1} = \mu^{m+1} \coloneqq \mathbb{1}_{\mathbf{w}\backslash\mathbf{c}}(\mathbf{W}\backslash\mathbf{C})$. For $i = m-1, \cdots, 1$,*

$$\mu_0^{i+1}(\overline{\mathbf{W}}^i, \overline{\mathbf{R}}^{1:i}) \coloneqq \mathbb{E}_{P_{\sigma(\mathbf{Z}_{i+1})}}\left[\mu_0^{i+2}(\overline{\mathbf{W}}^{i+1}, \mathbf{r}_{i+1}, \overline{\mathbf{R}}^{1:i})|\overline{\mathbf{W}}^i, \overline{\mathbf{R}}^{1:i}, \mathbf{r}_0, \mathbf{z}_{i+1}\right] \tag{7}$$

$$\pi_0^i(\overline{\mathbf{W}}^i, \overline{\mathbf{R}}^{1:i}) \coloneqq \frac{P_{\sigma(\mathbf{Z}_i)}(\overline{\mathbf{W}}^i, \overline{\mathbf{R}}^{1:i-1}|\mathbf{z}_i, \mathbf{r}_0)}{P_{\sigma(\mathbf{Z}_{i+1})}(\overline{\mathbf{W}}^i, \overline{\mathbf{R}}^{1:i-1}|\mathbf{z}_{i+1}, \mathbf{r}_0)} \frac{\mathbb{1}_{\mathbf{r}_i}(\mathbf{R}_i)}{P_{\sigma(\mathbf{Z}_{i+1})}(\mathbf{R}_i|\overline{\mathbf{W}}^i, \overline{\mathbf{R}}^{1:i-1}, \mathbf{z}_{i+1}, \mathbf{r}_0)}. \tag{8}$$

**Remark 1** (**Simplification of Nuisances**). *Although the nuisances $\pi_0^i$ may seem complicated, they can be simplified in several important special cases. For example, $\pi_0^i = \mathbb{1}_{\mathbf{r}_i}(\mathbf{R}_i)/P_{\sigma(\mathbf{Z})}(\mathbf{R}_i|\overline{\mathbf{W}}^i, \overline{\mathbf{R}}^{i-1}, \mathbf{z}, \mathbf{r}_0)$ if $\mathbb{Z} = \{\mathbf{Z}\}$ for any $\mathbf{Z} \subseteq \mathbf{V}$ where $\mathbf{Z}$ is possibly empty.*

In general, employing off-the-shelf classification methods for density ratio estimation is feasible, leveraging the techniques outlined in Section 5.4 of Díaz et al. [2021].

We now introduce a g-mSBD estimator exhibiting the robustness properties using these nuisances. This estimator is motivated by the double/debiased machine-learning style estimators [Chernozhukov et al., 2018, 2022]:

**Definition 3** (**DR-g-mSBD Estimators**). Let $D_{\sigma(\mathbf{Z}_i)}$ for $\mathbf{Z}_i \in \mathbb{Z}$ denote the experimental samples from randomizing the variable $\mathbf{Z}_i$. Let $\overline{D}_{\mathbf{z}_i}$ for $\mathbf{z}_i \in \mathfrak{D}_{\mathbf{Z}_i}$ denote the subsamples of $D_{\sigma(\mathbf{Z}_i)}$ fixing $\mathbf{R}_0 \backslash \mathbf{Z}_i = \mathbf{r}_0 \backslash \mathbf{z}_i$ and $\mathbf{Z}_i = \mathbf{z}_i$. The DR-g-mSBD estimator $\hat{A}$ for the g-mSBD adjustment $A_0[\mathbf{W}, \mathbf{C}, \mathbf{R}; \mathbb{Z}_0 := \{\mathbf{Z}_i\}_{i=1}^m, \texttt{seq} := (\mathbf{z}_i)_{i=1}^m](\mathbf{w}\backslash\mathbf{c}, \mathbf{r})$ is defined as follows:

1. Randomly partition $\overline{D}_{\mathbf{z}_i}$ into $\{\overline{D}_{\mathbf{z}_i,\ell}\}_{\ell \in [L]}$; i.e., $\overline{D}_{\mathbf{z}_i} = \cup_{\ell=1}^L \overline{D}_{\mathbf{z}_i,\ell}$, $\forall \mathbf{Z}_i \in \mathbb{Z}$ and $\mathbf{z}_i \in \mathfrak{D}_{\mathbf{Z}_i}$.

2. For each fold $\ell \in [L]$, let $\mu_\ell^{i+1}$ denote learned $\mu_0^{i+1}$ using $\overline{D}_{\mathbf{z}_{i+1}} \backslash \overline{D}_{\mathbf{z}_{i+1},\ell}$ for $i = m, \cdots, 2$; and $\pi_\ell^i$ learned $\pi_0^i$ for $i = 1, \cdots, m-1$. Define $\check{\mu}_\ell^{i+1} := \mu_\ell^{i+1}(\overline{\mathbf{W}}^i, \mathbf{r}_i, \overline{\mathbf{R}}^{1:i-1})$ and $\overline{\pi}_\ell^i := \prod_{j=1}^i \pi_\ell^j$.

3. Estimate $\hat{A} := \hat{A}(\{\mu_\ell^{j+1}, \pi_\ell^j\}_{j \in [m-1], \ell \in [L]}) := (1/L)\sum_{\ell=1}^L \hat{A}_\ell(\{\mu_\ell^{j+1}, \pi_\ell^j\}_{j \in [m-1]})$ where

$$\hat{A}_\ell := \hat{A}_\ell(\{\mu_\ell^{j+1}, \pi_\ell^j\}_{j \in [m-1]}) := \sum_{j=1}^{m-1} \mathbb{E}_{\overline{D}_{\mathbf{z}_{j+1},\ell}}\left[\overline{\pi}_\ell^j \{\check{\mu}_\ell^{j+2} - \mu_\ell^{j+1}\}\right] + \mathbb{E}_{\overline{D}_{\mathbf{z}_{1,\ell}}}\left[\check{\mu}_\ell^2\right], \quad (9)$$

where $\mathbb{E}_{\overline{D}_{\mathbf{z}_j,\ell}}[\cdot]$ is an empirical average over samples $\overline{D}_{\mathbf{z}_j,\ell}$.

We now analyze the doubly robustness property of this estimator.

**Proposition 1** (**Asymptotic Analysis of g-mSBD Estimators**). *Assume that the nuisance estimates $\mu_\ell^i$ and $\pi_\ell^i$ are $L_2$-consistent; i.e., $\|\mu_\ell^{i+1} - \mu_0^{i+1}\|_{P_{\sigma(\mathbf{Z}_{i+1})}} = o_{P_{\sigma(\mathbf{Z}_i)}}(1)$, $\|\check{\mu}_\ell^{i+2} - \check{\mu}_0^{i+2}\|_{P_{\sigma(\mathbf{Z}_{i+1})}} = o_{P_{\sigma(\mathbf{Z}_{i+1})}}(1)$ and $\|\pi_\ell^i - \pi_0^i\|_{P_{\sigma(\mathbf{Z}_{i+1})}} = o_{P_{\sigma(\mathbf{Z}_{i+1})}}(1)$ for $i = 1, \cdots, m-1$, and $\|\check{\mu}_\ell^2 - \check{\mu}_0^2\|_{P_{\sigma(\mathbf{Z}_1)}} = o_{P_{\sigma(\mathbf{Z}_1)}}(1)$. Let $n_i := |\overline{D}_{\mathbf{z}_i}|$ for $i \in \{1, \cdots, m\}$. Then,*

$$\hat{A} - A_0 = \sum_{i=1}^m R_i + \frac{1}{L}\sum_{\ell=1}^L \sum_{i=1}^{m-1} O_{P_{\sigma(\mathbf{Z}_{i+1})}}\left(\|\mu_\ell^{i+1} - \mu_0^{i+1}\|\|\pi_\ell^i - \pi_0^i\|\right), \quad (10)$$

*where $R_i$ is a random variable such that $n_i^{1/2} R_i$ converges in distribution to a mean-zero normal random variable.*

This estimator possesses a doubly robustness property since the estimator is bounded in probability at rate $n^{-1/2}$ (for $n := \min\{n_1, \cdots, n_m\}$, whenever $O_{P_{\sigma(\mathbf{Z}_{i+1})}}\left(\|\mu_\ell^{i+1} - \mu_0^{i+1}\|\|\pi_\ell^i - \pi_0^i\|\right) = O_P(n_{i+1}^{-1/2})$ for all $i$.

We now construct an estimator for the g-identification expression using the DR-g-mSBD estimator defined in Def. 3. The resulting estimator is called the MR-gID estimator:

**Definition 4** (**MR-gID Estimator**). The MR-gID estimator $\hat{\psi}$ for the identification expression of the causal effect $\psi_0 := f(\{A_0^k\}_{k=1}^K)$ in Theorem 1 is given as follows: For each $A_0^k$ composing $f(\{A_0^k\}_{k=1}^K)$, let $\hat{A}^k := \hat{A}^k(\{\mu_{k,\ell}^{j+1}, \pi_{k,\ell}^j\}_{j \in [m^k-1], \ell \in [L]})$ denote the DR-g-mSBD estimator with nuisance estimates $\{\mu_{k,\ell}^{j+1}, \pi_{k,\ell}^j\}$ for the true nuisances $\{\mu_{k,0}^{j+1}, \pi_{k,0}^j\}$. Then,

$$\hat{\psi} := f(\{\hat{A}^k\}_{k=1}^K). \quad (11)$$

We impose assumptions on the identification expression and its nuisances for further analysis.

**Assumption 2** (**Analysis of MR-gID** ). *The identification function $f(\{A^k\}_{k=1}^K)$ in Thm. 1 and each nuisances $\{\mu_{k,\ell}^{i+1}, \pi_{k,\ell}^i\}_{k,\ell}$ for $\hat{A}^k$ satisfy the following properties:*

1. ***Twice differentiability**: $f(\{A^k\}_{k=1}^K)$ is twice continuously Frechet differentiable w.r.t. $\{A^k\}_{k=1}^K$ w.r.t. $\{A^k\}_{k=1}^K$.*

2. **Boundedness**: $\forall k \in [K]$ and $\forall \mathbf{Z}_i \in \mathbb{Z}$, $\nabla_{A^k} f(\{A_0^j\}_{j=1}^K)[\hat{A}^k - A_0^k] = O_{P_{\sigma(\mathbf{Z}_i)}}(\hat{A}^k - A_0^k)$.

3. $L_2$-**Consistency**: $\|\mu_{k,\ell}^{i+1} - \mu_{k,0}^{i+1}\|_{P_{\sigma(\mathbf{Z}_{i+1}^k)}} = o_{P_{\sigma(\mathbf{Z}_{i+1}^k)}}(1)$, $\|\check{\mu}_{k,\ell}^{i+2} - \check{\mu}_{k,0}^{i+2}\|_{P_{\sigma(\mathbf{Z}_{i+1}^k)}} = o_{P_{\sigma(\mathbf{Z}_{i+1}^k)}}(1)$, $\|\pi_{k,\ell}^i - \pi_{k,0}^i\|_{P_{\sigma(\mathbf{Z}_{i+1}^k)}} = o_{P_{\sigma(\mathbf{Z}_{i+1}^k)}}(1)$, and $\|\check{\mu}_{k,\ell}^2 - \check{\mu}_{k,0}^2\|_{P_{\sigma(\mathbf{Z}_1^k)}} = o_{P_{\sigma(\mathbf{Z}_1^k)}}(1)$.

Assumption 2 is imposed to limit the error of the MR-gID, which is a linear function of the errors of each DR-g-mSBD estimator.

**Theorem 2 (Asymptotic Analysis of MR-gID).** *Suppose Assumption 2 holds. Let* $n_{k,i} := |\overline{D}_{\mathbf{z}_i^k}|$ *for* $\mathbf{Z}_i^k \in \mathbb{Z}$ *and* $\mathbf{z}_i^k \in \mathfrak{D}_{\mathbf{Z}_i^k}$. *Let* $\hat{\psi}$ *denote the MR-gID estimator in Def. 4 for the causal effect* $\psi_0 := f(\{A_0^k\}_{k=1}^K)$ *in Theorem 1. Then, the error of* $\hat{\psi}$ *is given as*

$$\hat{\psi} - \psi_0 = \sum_{k=1}^K \sum_{i=1}^{m^k} O_{P_{\sigma(\mathbf{Z}_i^k)}}(n_{k,i}^{-1/2}) + \frac{1}{L} \sum_{k=1}^K \sum_{\ell=1}^L \sum_{i=1}^{m^k-1} O_{P_{\sigma(\mathbf{Z}_i^k)}}(\|\mu_{k,\ell}^{i+1} - \mu_{k,0}^{i+1}\|\|\pi_{k,\ell}^i - \pi_{k,0}^i\|).$$

(12)

We highlight that the MR-gID $\hat{\psi}$ exhibits robustness property since $\hat{\psi} - \psi_0$ for $\psi_0 = P(\mathbf{y}|do(\mathbf{x}))$ is bounded at rate $n^{-1/2}$ (for $n = \min\{n_{k,i}\}$ and $P \in \mathbb{P}$) even when all nuisances $\{\mu_{k,\ell}^{i+1}, \pi_{k,\ell}^i\}$ are bounded at slower $n^{-1/4}$ rate. Furthermore, the MR-gID estimator exhibits multiply robustness, as the error of the MR-gID is a linear function of the error of DR-g-mSBD, which demonstrates the doubly robustness property. Formally,

**Corollary 2 (Multiply Robustness (Corollary of Thm. 2)).** *Suppose (1) Assumption 2 holds; (2) Either* $\pi_{k,\ell}^i = \pi_{k,0}^i$ *or* $\mu_{k,\ell}^j = \mu_{k,0}^j$ *for* $j = i+1, \cdots, m^k$ *for all* $i, \ell, k$; *and (3) all nuisances* $\{\pi_{k,\ell}^i, \mu_{k,\ell}^{i+1}\}_{i,\ell,k}$ *are bounded by some constant. Then, the MR-gID* $\hat{\psi}$ *(Def. 4) is consistent to* $\psi_0$.

For concreteness, we illustrate the application of Thm. 2 for Examples (1, 2). Detailed procedures are provided in Appendix A.

**Example 5 (Application of Thm. 2 to Example 1).** *Recall that* $P(y|do(x)) = f(\{A_0^1, A_0^2\}) := \sum_{z \in \mathfrak{S}_Z} A_0^1 A_0^2$. *The nuisance set for* $A_0^1$ *is* $\mu_{1,0}^2(X, W) := \mathbb{E}_P[\mathbb{1}_z(Z)|X, W]$ *and* $\pi_{1,0}^1(X, W) := \mathbb{1}_x(X)/P(X|W)$. *Then, the estimator for* $A_0^1$ *is* $\hat{A}^1 := \hat{A}^1(\{\mu_{1,\ell}^2, \pi_{1,\ell}^1\}_{\ell \in [L]})$ *defined in Def. 3. The nuisance set for* $A_0^2$ *is* $\mu_{2,0}^2 := \mathbb{E}_{P_{\sigma(Z)}}[\mathbb{1}_y(Y)]$. *Then, the estimator for* $A_0^2$ *is* $\hat{A}^2 := \hat{A}^2(\{\mu_{2,\ell}^2\}_{\ell \in [L]})$. *Then, the estimator is constructed as Def. 4, as* $f(\{\hat{A}^1, \hat{A}^2\})$. *By Thm. 2, the error of the estimator is* $O_P(n_0^{-1/2}) + O_P(n_z^{-1/2}) + (1/L) \sum_{\ell=1}^L O_P(\|\mu_{1,\ell}^2 - \mu_{1,0}^2\|\|\pi_{1,\ell}^1 - \pi_{1,0}^1\|)$, *where* $n_0 := |D|$ *and* $n_z := |D_z|$ *where* $D \sim P$ *and* $D_z \sim P_z$.

**Example 6 (Application of Thm. 2 to Example 2).** *Recall that* $P(y|do(x_1, x_2)) = f(\{A_0^2, A_0^{13}\}) = \sum_{r,w \in \mathfrak{S}_{R,W}} A_0^2 A_0^{13}$. *The nuisance set for* $A_0^2$ *is* $\mu_{2,0}^2 := \mathbb{E}_{P_{x_1}}[\mathbb{1}_w(W)|R, X_2]$ *and* $\pi_{1,0}^1 := \mathbb{1}_r(R)/P_{x_1}(R|X_2)$. *Then, the estimator for* $A_0^2$ *is* $\hat{A}^2 := \hat{A}^2(\{\mu_{2,\ell}^2, \pi_{1,\ell}^1\}_{\ell \in [L]})$. *The nuisance set for* $A_0^{13}$ *is* $\mu_{13,0}^2 := \mathbb{E}_{P_{x_2}}[\mathbb{1}_{r,y}(R, Y)|R, W]$ *and* $\pi_{13,0}^1 := \frac{P_{\sigma(X_1)}(R|x_2, x_1)}{P_{\sigma(X_2)}(R|x_2)} \frac{\mathbb{1}_w(W)}{P_{\sigma(X_2)}(W|R, x_2)}$. *Then, the estimator is* $\hat{A}^{13} = \hat{A}^{13}(\{\mu_{13,0}^2, \pi_{13,0}^1\}_{\ell \in [L]})$. *Then, the estimator is constructed as Def. 4, as* $f(\{\hat{A}^{13}, \hat{A}^2\})$. *By Thm. 2, the error of the estimator is* $O_{P_{\sigma(X_1)}}(n_1^{-1/2}) + O_{P_{\sigma(X_2)}}(n_2^{-1/2}) + (1/L) \sum_{\ell=1}^L \{O_{P_{\sigma(X_1)}}(\|\mu_{2,\ell}^2 - \mu_{2,0}^2\|\|\pi_{2,\ell}^1 - \pi_{2,0}^1\|) + O_{P_{\sigma(X_2)}}(\|\mu_{13,\ell}^2 - \mu_{13,0}^2\|\|\pi_{13,\ell}^1 - \pi_{13,0}^1\|)\}$, *where* $n_1 := |D_1|$ *and* $n_2 := |D_2|$ *where* $D_1 \sim P_{x_1}$ *and* $D_2 \sim P_{x_2}$.

## 4 Experiments

In this section, we demonstrate the MR-gID estimator from Definition (4) through Examples (1,2) and Project STAR dataset[Krueger and Whitmore, 2001, Schanzenbach, 2006]. For each example, the proposed estimator is constructed using a dataset $\mathbb{D} := \{D_{\mathbf{Z}_i}, \mathbf{Z}_i \in \mathbb{Z}\}$ simulated from an underlying SCM. Our goal is to provide empirical evidence of the fast convergence behavior and the robustness property of the proposed estimator compared to competing baseline estimators. We consider two standard baselines in the literature: the 'regression-based estimator (reg)' only uses the

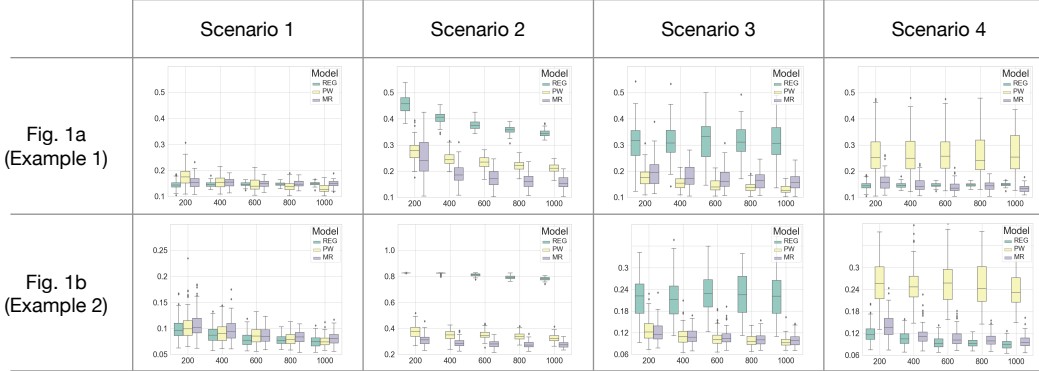

Figure 2: AAE Plots for Examples (1,2) for Scenarios {1,2,3,4} depicted in the Experimental Setup section. The $x$-axis and $y$-axis are the number of samples and AAE, respectively.

regression nuisance parameters $\mu^2$ for $\mu_0^2$ defined in Def. 2, and the 'probability weighting-based estimator (pw)' only uses the probability weighting parameters $\pi^{m-1}$ for $\pi_0^{m-1}$ defined in Def. 2, while our MR-gID uses both in estimating the g-mSBD operators $A^k$ composing $f(\{A^k\})$ in Thm. 1. Details of the regression-based ('reg') and the probability weighting-based ('pw') estimators are provided in Appendix A. The details of the simulation are in Appendix (D, E).

## 4.1 Synthetic Dataset Analysis

**Accuracy Measure.** We compare the proposed estimator ('mr') in Def. 4 to the regression-based estimator ('reg') and the probability weighting-based estimator ('pw'). In particular, we use $T^{\text{est}}(\mathbf{x})$ for est $\in \{\text{reg}, \text{pw}, \text{mr}\}$ to denote the g-ID estimators that leverage regression-based ('reg'), probability weighting-based ('pw'), and MR-gID in estimating each operator $A^k$ in the identification expression $f(\{A^k\})$ of the causal effect $P(\mathbf{y}|do(\mathbf{x}))$. We assess the quality of the estimators by computing the *average absolute error* $\text{AAE}^{\text{est}} := \frac{1}{|\mathfrak{D}_{\mathbf{X}}|} \sum_{\mathbf{x} \in \mathfrak{D}_{\mathbf{X}}} |T^{\text{est}}(\mathbf{x}) - P(\mathbf{y}|do(\mathbf{x}))|$ where $|\mathfrak{D}_{\mathbf{X}}|$ is the cardinality of $\mathfrak{D}_{\mathbf{X}}$. Nuisance functions are estimated using gradient boosting models called XGBoost [Chen and Guestrin, 2016]. We ran 100 simulations for each $n = \{200, 400, 600, 800, 1000\}$ for $n := |D_{\mathbf{Z}}|$ for $\forall \mathbf{Z} \subseteq \mathbb{Z}$. We label the box-plot for these AAEs as 'AAE-plot'.

**Experimental Setup.** We evaluate the $\text{AAE}^{\text{est}}$ for Examples (1,2) in four scenarios:

- **(Scenario 1)** There were no noises in estimating nuisances.
- **(Scenario 2)** We introduced a converging noise $\epsilon$ in estimating the nuisance, decaying at a $n^{-\alpha}$ rate (i.e., $\epsilon \sim \text{Normal}(n^{-\alpha}, N^{-2\alpha})$) for $\alpha = 1/4$ to emphasize the errors induced by the finiteness of samples. This scenario is inspired by the experimental design discussed in [Kennedy, 2020].
- **(Scenario 3)** Nuisance $\{\mu_{k,\ell}^{i+1}\}_{\ell,k,i}$ are estimated incorrectly - simulated by training the model with a random matrix having the same dimension as the input matrix.
- **(Scenario 4)** Nuisance $\{\pi_{k,\ell}^i\}_{\ell,k,i}$ are estimated incorrectly as in Scenario 3.

In Scenario 1, we aim to show that all estimators $T^{\text{reg}}, T^{\text{pw}}, T^{\text{mr}}$ are converging to the true causal quantity $P(\mathbf{y}|do(\mathbf{x}))$. In Scenario 2, we aim to show that the MR-gID estimator exhibits fast convergence behavior compared to competing estimators. In Scenario (3,4), our goal is to highlight the multiply robustness property of the MR-gID estimator.

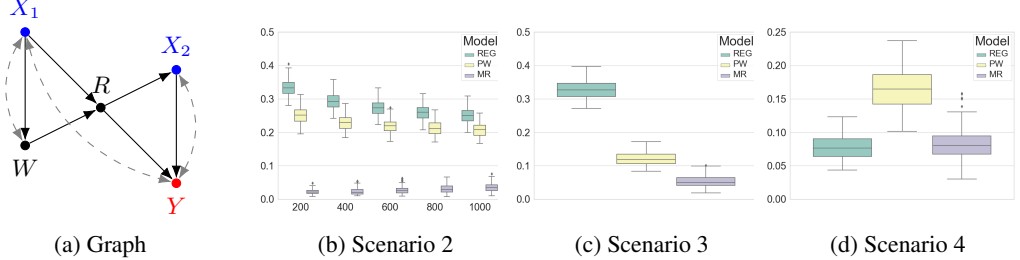

|         |         |         |         |
|---------|---------|---------|---------|
| (a) Graph | (b) Scenario 2 | (c) Scenario 3 | (d) Scenario 4 |

Figure 3: A graph and the AAE-plot for Project STAR.

**Experimental Results.** The AAE plots for all scenarios are presented in Fig. 2. All the estimators ('reg', 'pw', 'mr') converge in Scenario 1 as the sample size grows. In Scenario 2, where the estimated nuisances are controlled to be bounded in probability at $n^{-1/4}$ rate, the proposed MR-gID $\hat{\psi}$ outperforms the other two estimators by achieving fast convergence. This result corroborates the robustness property in Thm. 2. In Scenarios (3,4), where the estimated nuisances for $\{\mu^i\}_{i=2}^m$ or $\{\pi^i\}_{i=1}^{m-1}$ are wrongly specified, the MR-gID estimator converges while other estimators fail to converge. This result corroborates the multiply robustness property in Coro. 2.

## 4.2   Project STAR Dataset

This section provides an overview of the analysis using Project STAR dataset [Krueger and Whitmore, 2001, Schanzenbach, 2006]. Project STAR investigated the impact of teacher/student ratios on academic achievement for students in kindergarten through third grade. The dataset $D$ includes class size $(X_1)$, the academic outcome in kindergarten $(W)$ for kindergarten, the academic outcome in second grade $(R)$, class size $(X_2)$, and the academic outcome for the third grade $(Y)$. We assume that the SCM $\mathcal{M}$ underlying Project STAR dataset $D$ can be depicted in Figure 3a. The target quantity is $\mathbb{E}[Y|do(x_1, x_2)]$ where $P_{x_1,x_2}(y) = \sum_{r \in \mathfrak{D}_R} P_{x_1}(r) \sum_{x_1', w \in \mathfrak{D}_{X_1,W}} P_{x_2}(y|x_1', w, r, x_2) P_{x_2}(x_1', w)$. The detailed procedures are in Appendix E.

**Experimental Setup.** We generate two datasets $D_1$ and $D_2$ from the original dataset $D$ to demonstrate the gID estimation. $D_1$ is a random subsample of $D$ with only $\{X_1, W, R\}$ and follows $P_{\sigma(X_1)}(X_1, W, R)$. $D_2$ is constructed by resampling from $D$ in a way that the confounding bias between $X_1$ and $W$, and $X_1$ and $Y$ presents, following $P_{\sigma(X_2)}(X_1, W, X_2, R, Y)$. We conducted 100 simulations by generating new instances of $D_1$ and $D_2$ to create the AAE plot. Estimators were constructed solely from $D_1$ and $D_2$, with $D$ used exclusively to construct the ground-truth estimate.

**Experimental Results.** We evaluated the $\text{AAE}^{\text{est}}$ of estimators $T^{\text{est}}$ for est $\in \{\text{reg}, \text{pw}, \text{mr}\}$. The AAE plots for scenarios (2,3,4) are in Figs. (3b,3c,3d). Our findings indicate that the MR-gID estimator $T^{\text{mr}}$ consistently provided reliable estimates for the ground-truth quantity.

## 5   Conclusions

We present a framework for estimating the causal effect $P(\mathbf{y}|do(\mathbf{x}))$ by combining multiple observational and experimental datasets and a causal graph $G$. We introduce the generalized multi-outcome sequential back-door adjustment (g-mSBD) operator (Def. 1) and its operations. We show that any g-identifiable causal effects can be expressed as a function of the g-mSBD operators as specified in Algo. 1 (Thm. 1). We then develop an estimator called DR-g-mSBD (Def. 3) for the g-mSBD operator and analyze its statistical properties in Prop. 1. Based on the DR-g-mSBD estimator, we develop the MR-gID estimator (Def. 4) and analyze its statistical properties (Thm. 2 and Coro. 2) which exhibits fast convergence and multiply-robustness. Our experimental results demonstrate that the MR-gID estimator is a consistent and robust estimator of $P(\mathbf{y}|do(\mathbf{x}))$ against model misspecification and slow convergence.

## Acknowledgement

This research was supported in part by the NSF, ONR, AFOSR, DoE, Amazon, JP Morgan, and The Alfred P. Sloan Foundation.

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

# Supplement to "Estimating Causal Effects Identifiable from a Combination of Observations and Experiments"

## Contents

# A  Further Details

We restate the notation here. To clarify the relationship between the experimental samples, where randomization is applied to $\mathbf{Z} \subseteq \mathbf{V}$, and the distribution $P_{\mathbf{z}}(\mathbf{V} \backslash \mathbf{z})$, we introduce the notation $P_{\sigma(\mathbf{Z})}(\mathbf{V})$, where $\sigma(\mathbf{Z})$ indicates that $\mathbf{Z}$ has been randomized. The distribution $P_{\sigma(\mathbf{Z})}(\mathbf{V})$ is derived from the Structural Causal Model (SCM), where the original equation $Z \leftarrow f_Z(pa_z, u_z)$ for $Z \in \mathbf{Z}$ is replaced by a function that assigns a value to $Z = z$ randomly, independent of other endogenous variables. For example, assigning $Z = 1$ and $0$ with a probability of 0.5 each.

It should be noted that when considering observational data, $P := P_{\sigma(\emptyset)}$. For any sets $\mathbf{A}$, $\mathbf{B}$, and $\mathbf{Z} \subseteq \mathbf{V}$, the interventional distribution can be represented as $P(\mathbf{A} | \text{do}(\mathbf{z}), \mathbf{B}) = P_{\sigma(\mathbf{Z})}(\mathbf{A} | \mathbf{Z} = \mathbf{z}, \mathbf{B})$ according to the definition of the do-operator and the $P_{\sigma(\mathbf{Z})}$ distribution. To emphasize that the distribution is induced from randomization and conditioning on $\mathbf{Z} = \mathbf{z}$, we use $P_{\mathbf{z}}(\mathbf{A} | \mathbf{B}) := P_{\sigma(\mathbf{Z})}(\mathbf{A} | \mathbf{Z} = \mathbf{z}, \mathbf{B})$. The experimental samples obtained from randomization $\sigma(\mathbf{Z})$ lead to samples $D_{\sigma(\mathbf{Z})}$ that follow $P_{\sigma(\mathbf{Z})}(\mathbf{V})$. We denote the subsample of $D_{\sigma(\mathbf{Z})}$, where $\mathbf{Z} = \mathbf{z}$ is fixed, as $D_{\mathbf{z}}$, which follows $P_{\mathbf{z}}(\mathbf{V})$. ∎

## A.1  Example 3

We provide a detailed illustration of Example 3, demonstrating the application of Lemmas (2,3,4).

**Input**: $\mathbf{x} = \{x\}$, $\mathbf{y} := \{y\}$, $\mathbb{Z} := \{\emptyset, Z\}$. The goal is to identify $P(y | do(x))$ from $\mathbb{P}$ which contains $P$ and $P_{\sigma(Z)}(\mathbf{V})$. In the identification, $P(\mathbf{V})$ and $P_z(\mathbf{V} \backslash Z) := P_{\sigma(Z)}(\mathbf{V} | z)$ for $z \in \mathfrak{D}_Z$ are used.

**Line 3-4**: Since $\mathbf{V} \backslash \mathbf{X} = \{Z, Y\}$, $\mathbf{D} = an(Y)_{G(Z,Y)} = \{Z, Y\}$. Let $\mathbf{D}_1 := \{Z\}$ and $\mathbf{D}_2 := \{Y\}$.

We now run **Line 5-13**. We first run $\mathbf{D}_1 = \{Z\}$ and $\mathbf{Z}_1 = \emptyset$. Then,

1. **Line 7**: The c-component $\mathbf{S}_1^1 = \mathbf{V} = \{W, X, Z, Y\}$ includes $\mathbf{D}_1$.

2. **Line 8**: The c-factor $Q[\mathbf{S}_1^1]$ is identified as
$$Q[\mathbf{S}_1^1] = A_0[\mathbf{S}_1^1, \emptyset, \emptyset; \mathbb{Z}_1^1 := \emptyset, \emptyset](\mathbf{s}_1^1, \emptyset) = P(w)P(x|w)P(z|x,w)P(y|w,x,z).$$

3. **Line 9**: Run $Q[\mathbf{D}_1] = \text{SUBID}(\mathbf{D}_1, \mathbf{S}_1^1, Q[\mathbf{S}_1^1], G(\mathbf{S}_1^1))$.

   (a) **Line a.(2-3):** $\mathbf{A} = an(Z)_{G(\mathbf{V})} = \{W, X, Z\}$. Then, by Lemma 2,
   $$Q[\mathbf{A}] = \sum_{y \in \mathfrak{S}_Y} Q[\mathbf{S}] = A_0[\{W, X, Z\}, \emptyset, \emptyset; \emptyset, \emptyset](\{w, x, z\}, \emptyset) = P(w)P(x|w)P(z|x,w).$$

   (b) **Line a.(7-8):** Note $\mathbf{S} = \{W, Z\}$ is a c-component in $G(\mathbf{A})$ containing $\mathbf{D}_1 = \{Z\}$. Then,
   $$Q[\mathbf{S}] = \left( \sum_{x, z \in \mathfrak{S}_{X,Z}} Q[\mathbf{A}] \right) \times \frac{Q[\mathbf{A}]}{\sum_{z \in \mathfrak{D}_Z} Q[\mathbf{A}]}.$$
   By Lemma 2,
   $$\sum_{x, z \in \mathfrak{S}_{X,Z}} Q[\mathbf{A}] = A_0[W, \emptyset, \emptyset; \emptyset, \emptyset](w, \emptyset) = P(w),$$
   $$\sum_{z \in \mathfrak{S}_Z} Q[\mathbf{A}] = A_0[\{W, X\}, \emptyset, \emptyset; \emptyset, \emptyset](\{w, x\}, \emptyset, \emptyset) = P(w)P(x|w).$$
   By Lemma 4,
   $$\begin{aligned}
   \frac{Q[\mathbf{A}]}{\sum_{z \in \mathfrak{S}_Z} Q[\mathbf{A}]} &= \frac{A_0[\{W, X, Z\}, \emptyset, \emptyset; \emptyset, \emptyset](\{w, x, z\}, \emptyset)}{A_0[\{W, X\}, \emptyset, \emptyset; \emptyset, \emptyset](\{w, x\}, \emptyset, \emptyset)} \\
   &= A_0[Z, \emptyset, \{W, X\}; \emptyset, \emptyset](z, \{w, x\}) \\
   &= P(z | w, x).
   \end{aligned}$$

By Lemma 3,

$$Q[\mathbf{S}] = A_0[W, \emptyset, \emptyset, \emptyset; \emptyset, \emptyset](w, \emptyset) \times A_0[Z, \emptyset, \{W, X\}; \emptyset, \emptyset](z, \{w, x\})$$
$$= A_0[\{W, Z\}, \emptyset, X; \emptyset, \emptyset](\{w, z\}, x)$$
$$= P(w)P(z|w, x),$$

because the order is $W \prec_G X \prec_G Z$.

(c) **Line a.9:** Run $Q[\mathbf{D}_1] = \text{SUBID}(\mathbf{D}_1, \mathbf{S}, Q[\mathbf{S}], G(\mathbf{S}))$.

(d) **Line a.(2-3):** $\mathbf{A} = an(\mathbf{D}_1)_{G(\mathbf{S})} = \{Z\} = \mathbf{D}_1$. Then, by Lemma 2,

$$Q[\mathbf{D}_1] = \sum_{w \in \mathcal{W}} Q[\mathbf{S}]$$
$$= A_0^1 := A_0[\{W, Z\}, W, X; \emptyset, \emptyset](z, x)$$
$$= \sum_{w \in \mathfrak{S}_W} P(w)P(z|w, x).$$

We now run **Line 5-13**. We first run $\mathbf{D}_2 = \{Y\}$ and $\mathbf{Z}_1 = \emptyset$. We note that it fails since the sub-procedure $\text{subID}(\mathbf{D}_2, \mathbf{S}_1^1, Q[\mathbf{S}_1^1], G(\mathbf{S}_1^1))$ fails. Specifically, $\mathbf{A} := an(\mathbf{D}_2)_{G(\mathbf{S}_1^1)} = \mathbf{V} = \mathbf{S}_1^1$. Therefore, by **Line a.5**, the procedure fails.

We now run $\mathbf{D}_2$ with $\mathbf{Z}_2 = \{Z\}$.

1. **Line 7**: The c-component $\mathbf{S}_2^1 = \mathbf{V} \backslash Z = \{W, X, Y\}$.

2. **Line 8**: $Q[\mathbf{S}_2^1] = A_0[\mathbf{S}_2^1, \emptyset, \emptyset, ; \mathbb{Z}_2^1 = \{Z\}, \text{seq}_2^1](\mathbf{s}_2^1, \emptyset) = P_z(w)P_z(x|w)P_z(y|x, w)$, where $\text{seq}_2^1(W_j) = (z, z, z)$.

3. **Line 9**: We run $Q[\mathbf{D}_2] = \text{SUBID}(\mathbf{D}_2, \mathbf{S}_2^1, Q[\mathbf{S}_2^1], G(\mathbf{S}_2^1))$.

4. **Line a.(2-3):** $\mathbf{A} = an(\mathbf{D}_2)_{G(\mathbf{S}_2^1)} = \{Y\} = \mathbf{D}_2$. Then, by Lemma 2

$$Q[\mathbf{D}_2] = \sum_{w, x \in \mathfrak{S}_{W, X}} Q[\mathbf{S}_2^1] \tag{A.1}$$

$$= \sum_{w, x \in \mathfrak{S}_{W, X}} A_0[\{W, X, Y\}, \emptyset, \emptyset; \mathbb{Z}_2^1 = \{Z\}, \text{seq}_2^1](y; \emptyset) \tag{A.2}$$

$$= A_0[Y, \emptyset, \emptyset; \mathbb{Z}_2^1 = \{Z\}, \text{seq}_2^1](\{y\}; \emptyset) \tag{A.3}$$

$$= P_z(y). \tag{A.4}$$

Let

$$Q[\mathbf{D}_1] = A_0^1 := A_0[\{W, Z\}, W, X; \emptyset, \emptyset](z, x) \tag{A.5}$$

$$Q[\mathbf{D}_2] = A_0^2 := A_0[Y, \emptyset, \emptyset; \mathbb{Z}_2^1 = \{Z\}, \text{seq}_2^1](\{y\}; \emptyset). \tag{A.6}$$

By **Line 14**,

$$P(y|do(x)) = \sum_{z \in \mathfrak{S}_Z} Q[\mathbf{D}_1]Q[\mathbf{D}_2] = \sum_{z \in \mathfrak{S}_Z} A_0^1 A_0^2. \tag{A.7}$$

## A.2 Example 4

We provide a detailed illustration of Example 4, demonstrating the application of Lemmas (2,3,4).

**Input**: $\mathbf{x} = \{x_1, x_2\}$, $\mathbf{y} := \{y\}$, $\mathbb{Z} := \{X_1, X_2\}$. The goal is to identify $P(y|do(x_1, x_2))$ from $\mathbb{P} := \{P_{\sigma(X_1)}(\mathbf{V}), P_{\sigma(X_2)}(\mathbf{V})\}$. Specifically, two distributions $P_{x_1}(\mathbf{V} \backslash X_1)$ and $P_{x_2}(\mathbf{V} \backslash X_2)$ will be used in the identification task.

**Line 3-4**: $\mathbf{D} = an(Y)_{G(R, W, Y)} = \{R, W, Y\}$. Let $\mathbf{D}_1 := \{R\}$, $\mathbf{D}_2 := \{W\}$ and $\mathbf{D}_3 := \{Y\}$.

**Line 5-13**: Consider $\mathbf{D}_1 = \{R\}$ and $\mathbf{Z}_1 := \{X_1\}$. Note $\mathbf{S}_1^1 := \{R\} = \mathbf{D}_1$ is a c-component in $G(\mathbf{V}\backslash X_1)$. Therefore, $Q[\mathbf{D}_1] = Q[\mathbf{S}_1^1]$, where

$$Q[\mathbf{D}_1^1] = A_0^1 := A_0[R, \emptyset, X_2; \mathbb{Z}_1^1 := \{X_1\}, \mathtt{seq}_1^1](r, x_2) = P_{x_1}(r|x_2), \tag{A.8}$$

where $\mathtt{seq}_1^1 := (x_1)$.

**Line 5-13**: We now consider $\mathbf{D}_3 = \{Y\}$. Note that $Q[\mathbf{D}_3]$ is not identifiable from $P_{x_1}(\mathbf{V}\backslash X_1)$. To witness, consider the c-component $\mathbf{S}_3^1 := \{W, X_2, Y\}$ in $G(\mathbf{V}\backslash X_1)$. Then, the sub-procedure $\mathtt{subID}(\mathbf{D}_3, \mathbf{S}_3^1, Q[\mathbf{S}_3^1], G(\mathbf{S}_3^1))$ fails because failure condition in line a.5 is triggered. Specifically, $an(Y)_{G(\mathbf{S}_3^1)} = \mathbf{S}_3^1$.

Therefore, we consider $\mathbf{D}_3 = \{Y\}$ with $\mathbf{Z}_3 := x_2$. Note $\mathbf{S}_3^2 := \{Y\} = \mathbf{D}_3$ is a c-component in $G(\mathbf{V}\backslash X_2)$. Therefore, $Q[\mathbf{D}_3] = Q[\mathbf{S}_3^2]$ which is given by line 8:

$$Q[\mathbf{D}_3] = A_0[Y, \emptyset, \{R, W\}; \mathbb{Z}_3^2 := \{X_2\}, \mathtt{seq}_3^2 = (x_2)](y, \{r, w\}) \tag{A.9}$$
$$= P_{x_2}(y|r, w). \tag{A.10}$$

**Line 5-13**: Consider $\mathbf{D}_2 = \{W\}$ and $\mathbf{Z}_1 := X_1$. Note that $Q[\mathbf{D}_2]$ is not identifiable from $P_{x_2}(\mathbf{V}\backslash X_2)$. To witness, consider the c-component $\mathbf{S}_2^2 := \{X_1, W\}$ in $G(\mathbf{V}\backslash X_2)$. Then, the sub-procedure $\mathtt{subID}(\mathbf{D}_2, \mathbf{S}_2^2, Q[\mathbf{S}_2^2], G(\mathbf{S}_2^2))$ fails because failure condition in line a.5 is triggered. Specifically, $an(W)_{G(\mathbf{S}_2^2)} = \mathbf{S}_2^2$.

Therefore, we consider $\mathbf{D}_2 = \{W\}$ with $\mathbf{Z}_1 := X_1$.

1. Note $\mathbf{S}_2^1 = \{X_2, W, Y\}$ is a c-component in $G(\mathbf{V}\backslash X_1)$ containing $\mathbf{D}_2$. Then,

$$Q[\mathbf{S}_2^1] = A_0[\{X_2, W, Y\}, \emptyset, R; \mathbb{Z}_2^1 := \{X_1\}, \mathtt{seq}_2^1 = (x_1, x_1, x_1)](\{x_2, w, y\}, r) \tag{A.11}$$
$$= P_{x_1}(x_2)P_{x_1}(w|r, x_2)P_{x_1}(y|x_2, w, r). \tag{A.12}$$

2. Run $Q[\mathbf{D}_2] = \mathtt{SUBID}(\mathbf{D}_2, \mathbf{S}_2^1, Q[\mathbf{S}_2^1], G(\mathbf{S}_2^1))$.

3. **Line a.(2-3):** $\mathbf{A} = an(W)_{G(\mathbf{S}_2^1)} = \{W\} = \mathbf{D}_2$. Then,

$$Q[\mathbf{D}_2] = \sum_{y, x_2 \in \mathfrak{S}_{Y, X_2}} Q[\mathbf{S}_2^1] \tag{A.13}$$

$$= \sum_{y, x_2 \in \mathfrak{S}_{Y, X_2}} A_0[\{X_2, W, Y\}, \emptyset, R; \mathbb{Z}_2^1, \mathtt{seq}_2^1](\{x_2, w, y\}, r) \tag{A.14}$$

$$= A_0[\{X_2, W\}, X_2, R; \mathbb{Z}_2^1, \mathtt{seq}_2^1 = (x_1, x_1)](w, r) \tag{A.15}$$

$$= \sum_{x_2' \in \mathcal{X}_2} P_{x_1}(x_2)P_{x_1}(w|r, x_2'). \tag{A.16}$$

Also, by Lemma 3,

$$Q[\mathbf{D}_1]Q[\mathbf{D}_3] \tag{A.17}$$
$$= A_0^{13} \tag{A.18}$$
$$:= A_0[R, \emptyset, X_2; \mathbb{Z}_1^1 := \{X_1\}, \mathtt{seq}_1^1](r, x_2) \times A_0[Y, \emptyset, \{R, W\}; \mathbb{Z}_3^2 := \{X_2\}, \mathtt{seq}_3^2](y, \{r, w\}) \tag{A.19}$$
$$= A_0[\{R, Y\}, \emptyset, \{X_2, W\}; \mathbb{Z}^{13} := \{X_1, X_2\}, \mathtt{seq}^{13} = (x_1, x_2), G](\{r, y\}, \{x_2, w\}) \tag{A.20}$$
$$= P_{x_1}(r|x_2)P_{x_2}(y|r, w). \tag{A.21}$$

Finally,

$$P(y|do(x_1, x_2)) = \sum_{r, w \in \mathfrak{S}_{R, W}} A_0^2 A_0^{13}. \tag{A.22}$$

### A.3 Example 5

#### A.3.1 Specification of Nuisances

Recall that the topological order of the variable is $W \prec_G X \prec_G Z \prec_G Y$. Also, $P(y|do(x)) = \sum_{z \in \mathfrak{S}_Z} A_0^1 A_0^2$ where

$$A_0^1 := A_0[\{W, Z\}, W, X; \emptyset, \emptyset](z, x) = \sum_{w \in \mathfrak{S}_W} P(z|x, w)P(w) \tag{A.23}$$

$$A_0^2 := A_0[Y, \emptyset, \emptyset; \{Z\}, (z)](y, \emptyset) = P_z(y) := P_{\sigma(Z)}(y|z). \tag{A.24}$$

That is,

$$P(y|do(x)) = f(A_0^1, A_0^2) := \sum_{z \in \mathfrak{S}_Z} A_0^1 A_0^2. \tag{A.25}$$

By leveraging the definition of the nuisance in Def. 2, the nuisance composing $A_0^1$ is $\{\mu_{1,0}^2, \pi_{1,0}^1\}$ which are defined as follow:

$$\mu_{1,0}^2(X, W) := \mathbb{E}_P \left[ \mathbb{1}_z(Z)|X, W \right],$$
$$\pi_{1,0}^1(X, W) := \mathbb{1}_x(X)/P(X|W).$$

The nuisance composing $A_0^2$ is $\{\mu_{2,0}^2\}$ which is $\mu_{2,0}^2 := \mathbb{E}_{P_z} \left[ \mathbb{1}_y(Y) \right]$.

#### A.3.2 Construction of Estimators

We apply the procedure in Def. 3 to construct estimators $\hat{A}^1$ and $\hat{A}^2$ for $A_0^1$ and $A_0^2$. We choose $L = 2$. We first construct $\hat{A}^1$ for the fixed $\{z, x\} \in \mathfrak{D}_{Z,X}$. We note that $\hat{A}_\ell^1$ for $\ell \in \{1, 2\}$ is given as follow: For a fixed $z, x$,

$$\hat{A}_\ell^1 := \mathbb{E}_{D_\ell} \left[ \pi_\ell^1(X, W)\{\mathbb{1}_z(Z) - \mu_{1,\ell}^2(X, W)\} + \mu_{1,\ell}^2(x, W) \right], \tag{A.26}$$

and

$$\hat{A}^1 = 1/L \sum_{\ell=1}^L \hat{A}_\ell^1, \tag{A.27}$$

where $\pi^1, \mu^2$ are nuisances estimated using $D \backslash D_\ell$. Specifically, $\mu_{1,\ell}^2(X, W)$ is obtained by using the XGBoost [Chen and Guestrin, 2016] regression model which regresses $\mathbb{1}_z(Z)$ onto the $\{X, W\}$ using $D \backslash D_\ell$. $\mu_{1,\ell}^2(x, W)$ is evaluated from $D_\ell$ after fixing a column for $X$ to $x$. In similar, $\pi^1$ as follow: we first model $P(X|W)$ by regressing $X$ onto $W$ from the data $D \backslash D_\ell$ using the XGBoost [Chen and Guestrin, 2016]. Then, we evaluate $\pi_{1,0}^1(X, W)$ by plugging in the trained $P(X|W)$.

We now construct $\hat{A}^2$ for the fixed $z$ and $y$. We first take the subsamples $D_z$ from the experimental samples $D_{\sigma(Z)} \in \mathbb{D}$, where $D_z$ is the sample where $Z = z$. Then, we compute the following:

$$\hat{A}^2 = \mu_2^2 = \mathbb{E}_{D_z} \left[ \mathbb{1}_y(Y) \right]. \tag{A.28}$$

Then, following Def. 4, the MR-gID is constructed as follow:

$$f(\hat{A}^1, \hat{A}^2) = \sum_{z \in \mathfrak{D}_Z} \hat{A}^1 \hat{A}^2. \tag{A.29}$$

### A.4 Example 6

#### A.4.1 Specification of Nuisances

Recall that the topological order of the variable is $X_1 \prec_G X_2 \prec_G R \prec_G W \prec_G Y$. Also,

$$A_0^2 := A_0[\{X_2, W\}, \{X_2\}, R; \mathbb{Z}^2 = \{X_1\}, \mathrm{seq}^2 = (x_1, x_1)](w, r) \tag{A.30}$$

$$= \sum_{x_2' \in \mathfrak{S}_{X_2}} P_{x_1}(w|r, x_2') P_{x_1}(x_2') \tag{A.31}$$

$$A_0^{13} := A_0[\{R, Y\}, \emptyset, \{X_2, W\}; \mathbb{Z}^{13} = \{X_1, X_2\}, \mathrm{seq}^{13} = (x_1, x_2)](\{r, y\}, \{x_2, w\}) \tag{A.32}$$

$$= P_{x_1}(r|x_2) P_{x_2}(y|r, w). \tag{A.33}$$

Then,

$$P(y|do(x_1, x_2)) = \sum_{r, w \in \mathfrak{S}_{R,W}} A_0^2 A_0^{13}. \tag{A.34}$$

The nuisance composing $A_0^2$ is $\{\mu_{2,0}^2, \pi_{2,0}^1\}$ which are defined as follow:

$$\mu_{2,0}^2(R, X_2) := \mathbb{E}_{P_{x_1}}\left[\mathbb{1}_w(W)|R, X_2\right], \tag{A.35}$$

$$\pi_{2,0}^1(R, X_2) := \mathbb{1}_r(R)/P_{x_1}(R|X_2). \tag{A.36}$$

The nuisance composing $A_0^{13}$ is $\{\mu_{2,0}^2, \pi_{2,0}^1\}$ which are defined as follow:

$$\mu_{13,0}^2(R, W) := \mathbb{E}_{P_{x_2}}\left[\mathbb{1}_{r,y}(R, Y)|R, W\right] \tag{A.37}$$

$$= \mathbb{E}_{P_{\sigma(X_2)}}\left[\mathbb{1}_{r,y}(R, Y)|R, W, x_2\right] \tag{A.38}$$

$$= \mathbb{1}_r(R)\mathbb{E}_{P_{\sigma(X_2)}}\left[\mathbb{1}_y(Y)|R, W, x_2\right], \tag{A.39}$$

and

$$\pi_{13,0}^1(X_2, W) := \frac{P_{\sigma(X_1)}(R|x_2, x_1)}{P_{\sigma(X_2)}(R|x_2)} \frac{\mathbb{1}_w(W)}{P_{\sigma(X_2)}(W|R, x_2)}. \tag{A.40}$$

#### A.4.2 Construction of Estimators

We apply the procedure in Def. 3 to construct estimators $\hat{A}^2$ and $\hat{A}^{13}$ for $A_0^2$ and $A_0^{13}$. We choose $L = 2$. We first construct $\hat{A}^2$ for the fixed $\{w, r, x_1\}$. We note that $\hat{A}^2 := (1/L)\sum_{\ell=1}^L \hat{A}_\ell^2$ for $\ell \in \{1, 2\}$ where $\hat{A}_\ell^2$ is given as follow:

$$\hat{A}_\ell^2 := \mathbb{E}_{D_{x_1,\ell}}\left[\pi_{2,\ell}^1(R, X_2)\{\mathbb{1}_w(W) - \mu_{2,\ell}^2(R, X_2)\} + \mu_{2,\ell}^2(r, X_2)\right], \tag{A.41}$$

where $D_{x_1}$ is a subsample of $D_{\sigma(X_1)}$ fixing $X_1 = x_1$, and $\pi_{2,\ell}^1, \mu_{2,\ell}^2$ are nuisances trained using $D_{x_1} \backslash D_{x_1,\ell}$. We note that $\mu_{2,\ell}^2(R, X_2)$ is constructed by regressing $\mathbb{1}_w(W)$ onto $\{R, X_2\}$. Also, $\pi_{2,\ell}^1(R, X_2)$ is constructed by regressing $R$ onto $X_2$.

We now construct $\hat{A}^{13} := (1/L)\sum_{\ell=1}^L \hat{A}_\ell^{13}$ for $\ell \in \{1, 2\}$ where $\hat{A}_\ell^{13}$ is given as follow:

$$\hat{A}_\ell^{13} := \mathbb{E}_{D_{x_2,\ell}}\left[\pi_{13,\ell}^1(X_2, W)\{\mathbb{1}_{r,y}(R, Y) - \mu_{13,\ell}^2(R, W)\}\right] + \mathbb{E}_{\overline{D}_{x_1,\ell}}\left[\mu_{13,\ell}^2(R, w)\}\right], \tag{A.42}$$

where $D_{x_1}$ is a subsample of $D_{\sigma(X_1)}$ fixing $X_1 = x_1$, and $\overline{D}_{x_1}$ is a subsample of $D_{x_1}$ fixing $X_2 = x_2$. $\pi_{2,\ell}^1, \mu_{2,\ell}^2$ are nuisances trained using $\overline{D}_{x_1} \backslash \overline{D}_{x_1,\ell}$ and $\overline{D}_{x_2} \backslash \overline{D}_{x_2,\ell}$.

### A.5 Details on Regression-based (REG) and Probability Weighting-based (PW) estimators.

In this section, we provide details on two alternative g-ID estimators used in Sec. 4: $T^{\mathrm{reg}} := f(\{\hat{A}^{k,\mathrm{reg}}\}_{k=1}^K)$ ('regression-based estimators') where $\hat{A}^{k,\mathrm{reg}}$ denotes the regression-based estima-

tor for the g-mSBD operator, and $T^{\mathrm{pw}} = f(\{\hat{A}^{k,\mathrm{pw}}\}_{k=1}^{K})$ ('probability weighting-based estimators) where $\hat{A}^{k,\mathrm{reg}}$ denotes the probability weighting-based estimator for the g-mSBD operator.

### A.5.1 Regression-based Estimator

The regression-based g-mSBD estimator is defined as follows:

**Definition A.1** (**Regression-based g-mSBD Estimator**). Let $D_{\sigma(\mathbf{Z}_i)}$ for $\mathbf{Z}_i \in \mathbb{Z}$ denote the experimental samples from randomizing the variable $\mathbf{Z}_i$. Let $\overline{D}_{\mathbf{z}_i}$ for $\mathbf{z}_i \in \mathfrak{D}_{\mathbf{Z}_i}$ denote the subsamples of $D_{\sigma(\mathbf{Z}_i)}$ fixing $\mathbf{R}_0 \backslash \mathbf{Z}_i = \mathbf{r}_0 \backslash \mathbf{z}_i$ and $\mathbf{Z}_i = \mathbf{z}_i$. A regression-based estimator $\hat{A}^{\mathrm{reg}}$ for the g-mSBD adjustment $A_0[\mathbf{W}, \mathbf{C}, \mathbf{R}; \mathbb{Z}_0 \coloneqq \{\mathbf{Z}_i\}_{i=1}^{m}, \mathtt{seq} \coloneqq (\mathbf{z}_i)_{i=1}^{m}](\mathbf{w} \backslash \mathbf{c}, \mathbf{r})$ is given as follows:

1. Randomly partition $\overline{D}_{\mathbf{z}_i}$ into $\{\overline{D}_{\mathbf{z}_i,\ell}\}_{\ell \in [L]}$; i.e., $\overline{D}_{\mathbf{z}_i} = \cup_{\ell=1}^{L} \overline{D}_{\mathbf{z}_i,\ell}$, $\forall \mathbf{Z}_i \in \mathbb{Z}$ and $\mathbf{z}_i \in \mathfrak{D}_{\mathbf{Z}_i}$.

2. For each fold $\ell \in [L]$, let $\mu_{\ell}^{i+1}$ denote learned $\mu_0^{i+1}$ using $\overline{D}_{\mathbf{z}_{i+1}} \backslash \overline{D}_{\mathbf{z}_{i+1},\ell}$ for $i = m, \cdots, 2$. Define $\check{\mu}_{\ell}^{i+1} \coloneqq \mu_{\ell}^{i+1}(\overline{\mathbf{W}}^{i}, \mathbf{r}_i, \overline{\mathbf{R}}^{1:i-1})$.

3. Estimate $\hat{A}^{\mathrm{reg}} \coloneqq \hat{A}^{\mathrm{reg}}(\{\mu_{\ell}^{j+1}\}_{j \in [m-1], \ell \in [L]}) \coloneqq (1/L) \sum_{\ell=1}^{L} \hat{A}_{\ell}^{\mathrm{reg}}(\{\mu_{\ell}^{j+1}\}_{j \in [m-1]})$ where

$$\hat{A}_{\ell}^{\mathrm{reg}} \coloneqq \hat{A}_{\ell}^{\mathrm{reg}}(\{\mu_{\ell}^{j+1}\}_{j \in [m-1]}) \coloneqq \mathbb{E}_{\overline{D}_{\mathbf{z}_1,\ell}} \left[ \check{\mu}_{\ell}^2 \right]. \tag{A.43}$$

The error of the regression-based estimator is given as follows:

**Proposition A.1** (**Error Analysis of the regression-based g-mSBD estimator**). *Suppose* $\|\mu_{\ell}^2 - \mu_0^2\|_{P_{\sigma(\mathbf{Z}_1)}} = o_{P_{\sigma(\mathbf{Z}_1)}}(1)$. *Then,*

$$\hat{A}^{reg} - A_0 = O_{P_{\sigma(\mathbf{Z}_1)}}(n_1^{-1/2}) + \frac{1}{L} \sum_{\ell=1}^{L} O_{P_{\sigma(\mathbf{Z}_1)}}(\|\mu_{\ell}^2 - \mu_0^2\|). \tag{A.44}$$

***Proof of Proposition A.1.*** We note that

$$A_0 = \mathbb{E}_{P_{\sigma(\mathbf{Z}_1)}} \left[ \check{\mu}_0^2 | \mathbf{z}_1, \mathbf{r}_0 \right] \tag{A.45}$$

by the analysis in Lemma S.2. Therefore, by Lemma S.7,

$$\hat{A}_{\ell}^{\mathrm{reg}} - A_0 = \mathbb{E}_{\overline{D}_{\mathbf{z}_1,\ell} - P_{\sigma(\mathbf{Z}_1)|\mathbf{r}_0,\mathbf{z}_1}} \left[ \check{\mu}_0^2 \right] \tag{A.46}$$

$$+ \mathbb{E}_{\overline{D}_{\mathbf{z}_1,\ell} - P_{\sigma(\mathbf{Z}_1)|\mathbf{r}_0,\mathbf{z}_1}} \left[ \check{\mu}_{\ell}^2 - \check{\mu}_0^2 \right] \tag{A.47}$$

$$+ \mathbb{E}_{P_{\sigma(\mathbf{Z}_1)}} \left[ \check{\mu}_{\ell}^2 - \check{\mu}_0^2 | \mathbf{r}_0, \mathbf{z}_1 \right]. \tag{A.48}$$

By the central limit theorem,

$$\text{Eq. (A.46)} = O_{P_{\sigma(\mathbf{Z}_1)}}(n_{1,\ell}^{-1/2}), \tag{A.49}$$

where $n_{1,\ell} \coloneqq |\overline{D}_{\mathbf{z}_1,\ell}|$.

By [Kennedy et al., 2020, Lemma 2] and the given assumption that $\|\mu_{\ell}^2 - \mu_0^2\|_{P_{\sigma(Z_1)}} = o_{P_{\sigma(Z_1)}}(1)$,

$$\text{Eq. (A.47)} = O_{P_{\sigma(\mathbf{Z}_1)}}(1/n_{1,\ell}^{-1/2}). \tag{A.50}$$

Finally, by applying Cauchy-Schwarz inequality,

$$\text{Eq. (A.48)} = O_{P_{\sigma(\mathbf{Z}_1)}}(\|\check{\mu}_{\ell}^2 - \check{\mu}_0^2\|). \tag{A.51}$$

Finally,

$$\hat{A}^{\mathrm{reg}} - A_0 = \frac{1}{L}\sum_{\ell=1}^{L}(\hat{A}_\ell^{\mathrm{reg}} - A_0) \tag{A.52}$$

$$= \frac{1}{L}\sum_{\ell=1}^{L}\left(O_{P_{\sigma(\mathbf{Z}_1)}}(n^{-1/2_{1,\ell}}) + O_{P_{\sigma(\mathbf{Z}_1)}}(\|\check{\mu}_\ell^2 - \check{\mu}_0^2\|)\right) \tag{A.53}$$

$$= O_{P_{\sigma(\mathbf{Z}_1)}}(n^{-1/2_1}) + \frac{1}{L}\sum_{\ell=1}^{L}O_{P_{\sigma(\mathbf{Z}_1)}}(\|\mu_\ell^2 - \mu_0^2\|). \tag{A.54}$$

$\square$

### A.5.2 Probability-weighting based Estimator

In this section, we define and analyze the probability weighting-based g-mSBD estimator. The probability-weighting-based estimator is defined as follows:

**Definition A.2 (Probability-weighting-based g-mSBD Estimator).** Let $D_{\sigma(\mathbf{Z}_i)}$ for $\mathbf{Z}_i \in \mathbb{Z}$ denote the experimental samples from randomizing the variable $\mathbf{Z}_i$. Let $\overline{D}_{\mathbf{z}_i}$ for $\mathbf{z}_i \in \mathfrak{D}_{\mathbf{Z}_i}$ denote the subsamples of $D_{\sigma(\mathbf{Z}_i)}$ fixing $\mathbf{R}_0 \backslash \mathbf{Z}_i = \mathbf{r}_0 \backslash \mathbf{z}_i$ and $\mathbf{Z}_i = \mathbf{z}_i$. A probability weighting-based estimator $\hat{A}^{\mathrm{pw}}$ for the g-mSBD adjustment $A_0[\mathbf{W}, \mathbf{C}, \mathbf{R}; \mathbb{Z}_0 \coloneqq \{\mathbf{Z}_i\}_{i=1}^m, \mathtt{seq} \coloneqq (\mathbf{z}_i)_{i=1}^m](\mathbf{w}\backslash\mathbf{c}, \mathbf{r})$ is given as follows:

1. Randomly partition $\overline{D}_{\mathbf{z}_i}$ into $\{\overline{D}_{\mathbf{z}_i,\ell}\}_{\ell\in[L]}$; i.e., $\overline{D}_{\mathbf{z}_i} = \cup_{\ell=1}^L \overline{D}_{\mathbf{z}_i,\ell}, \ \forall \mathbf{Z}_i \in \mathbb{Z}$ and $\mathbf{z}_i \in \mathfrak{D}_{\mathbf{Z}_i}$.

2. For each fold $\ell \in [L]$, let $\pi_\ell^i$ denote learned $\pi_0^i$ using $\overline{D}_{\mathbf{z}_i}\backslash\overline{D}_{\mathbf{z}_i,\ell}$ for $i = m-1, \cdots, 1$. Let $\overline{\pi}_\ell^{m-1} \coloneqq \prod_{i=1}^{m-1}\pi_\ell^i$.

3. Estimate $\hat{A}^{\mathrm{pw}} \coloneqq \hat{A}^{\mathrm{pw}}(\{\pi_\ell^j\}_{j\in[m-1],\ell\in[L]}) \coloneqq (1/L)\sum_{\ell=1}^L \hat{A}_\ell^{\mathrm{pw}}(\{\pi_\ell^j\}_{j\in[m-1]})$ where

$$\hat{A}_\ell^{\mathrm{pw}} \coloneqq \hat{A}_\ell^{\mathrm{pw}}(\{\pi_\ell^j\}_{j\in[m-1]}) \coloneqq \mathbb{E}_{\overline{D}_{\mathbf{z}_{m,\ell}}}\left[\overline{\pi}_\ell^{m-1}\mathbb{1}_{\mathbf{w}\backslash\mathbf{c}}(\mathbf{W}\backslash\mathbf{C})\right]. \tag{A.55}$$

**Lemma S.1 (Representation of the g-mSBD operator using Probability Weighting).** *The g-mSBD adjustment $A_0$ in Def. 1 can be represented as*

$$A_0 = \mathbb{E}_{P_{\sigma(\mathbf{z}_m)}}\left[\overline{\pi}_0^{m-1}\mathbb{1}_{\mathbf{w}\backslash\mathbf{c}}(\mathbf{W}\backslash\mathbf{C})|\mathbf{z}_m, \mathbf{r}_0\right]. \tag{A.56}$$

*Proof of Lemma S.1.* It suffices to show that, for $k = m-1, \cdots, 1$,

$$\mathbb{E}_{P_{\sigma(\mathbf{z}_{k+1})}}\left[\overline{\pi}_0^k\check{\mu}_0^{k+2}|\mathbf{z}_{k+1}, \mathbf{r}_0\right] = \mathbb{E}_{P_{\sigma(\mathbf{z}_k)}}\left[\overline{\pi}_0^k\check{\mu}_0^{k+1}|\mathbf{z}_k, \mathbf{r}_0\right]. \tag{A.57}$$

If this holds, Lemma S.1 can be shown as follows:

$$\begin{aligned}
A_0 &= \mathbb{E}_{P_{\sigma(\mathbf{z}_1)}}\left[\check{\mu}_0^2|\mathbf{z}_1, \mathbf{r}_0\right] \\
&= \mathbb{E}_{P_{\sigma(\mathbf{z}_2)}}\left[\overline{\pi}_0^1\check{\mu}_0^3|\mathbf{z}_2, \mathbf{r}_0\right] \\
&= \mathbb{E}_{P_{\sigma(\mathbf{z}_m)}}\left[\overline{\pi}_0^{m-1}\check{\mu}_0^{m+1}|\mathbf{z}_m, \mathbf{r}_0\right] \\
&= \mathbb{E}_{P_{\sigma(\mathbf{z}_m)}}\left[\overline{\pi}_0^{m-1}\mathbb{1}_{\mathbf{w}\backslash\mathbf{c}}(\mathbf{W}\backslash\mathbf{C})|\mathbf{z}_m, \mathbf{r}_0\right].
\end{aligned}$$

Eq. (A.57) holds as follows:

$$\begin{aligned}
&\mathbb{E}_{P_{\sigma(\mathbf{z}_{k+1})}}\left[\overline{\pi}_0^k\check{\mu}_0^{k+2}|\mathbf{z}_{k+1}, \mathbf{r}_0\right] \\
&= \mathbb{E}_{P_{\sigma(\mathbf{z}_{k+1})}}\left[\overline{\pi}_0^k\mu_0^{k+1}|\mathbf{z}_{k+1}, \mathbf{r}_0\right] \\
&= \mathbb{E}_{P_{\sigma(\mathbf{z}_k)}}\left[\overline{\pi}_0^{k-1}\check{\mu}_0^{k+1}|\mathbf{z}_k, \mathbf{r}_0\right].
\end{aligned}$$

This completes the proof.

$\square$

Equipped with Lemma S.1, we analyze the error of the probability-weighting-based estimator as follow:

**Proposition A.2 (Error Analysis of the probability weighting-based g-mSBD estimator).** *Suppose* $\|\{\pi_\ell^1 \tau_\ell^2 - \pi_0^1 \tau_0^2\}\|_{P_{\sigma(\mathbf{Z}_2)}} = o_{P_{\sigma(\mathbf{Z}_2)}}(1)$. *Then,*

$$\hat{A}^{pw} - A_0 = O_{P_{\sigma(\mathbf{Z},)}}(n_m^{-1/2}) + \frac{1}{L}\sum_{\ell=1}^{L} O_{P_{\sigma(\mathbf{Z}_m)}}(\|\overline{\pi}_\ell^m - \overline{\pi}_0^m\|). \tag{A.58}$$

*Proof of Proposition A.2.* By Lemma S.7 and Assumption 1,

$$\hat{A}_\ell^{\mathrm{pw}} - A_0 = \mathbb{E}_{\overline{D}_{\mathbf{z}_m, \ell} - P_{\sigma(\mathbf{Z}_m)|\mathbf{z}_m, \mathbf{r}_0}} \left[\overline{\pi}_0^m \mathbb{1}_{\mathbf{w}\backslash\mathbf{c}}(\mathbf{W}\backslash\mathbf{C})\right] \tag{A.59}$$

$$+ \mathbb{E}_{\overline{D}_{\mathbf{z}_m, \ell} - P_{\sigma(\mathbf{Z}_m)|\mathbf{z}_m, \mathbf{r}_0}} \left[(\overline{\pi}^m - \overline{\pi}_0^m) \mathbb{1}_{\mathbf{w}\backslash\mathbf{c}}(\mathbf{W}\backslash\mathbf{C})\right] \tag{A.60}$$

$$+ \mathbb{E}_{P_{\sigma(\mathbf{Z}_m)|\mathbf{z}_m, \mathbf{r}_0}} \left[(\overline{\pi}^m - \overline{\pi}_0^m) \mathbb{1}_{\mathbf{w}\backslash\mathbf{c}}(\mathbf{W}\backslash\mathbf{C})\right]. \tag{A.61}$$

By the central limit theorem,

$$\text{Eq. (A.59)} = O_{P_{\sigma(\mathbf{z}_m)}}(n_{m,\ell}^{-1/2}). \tag{A.62}$$

By [Kennedy et al., 2020, Lemma 2] and the given assumption,

$$\text{Eq. (A.60)} = O_{P_{\sigma(\mathbf{z}_m)}}(n_{m,\ell}^{-1/2}). \tag{A.63}$$

Finally, by applying Cauchy-Schwarz inequality,

$$\text{Eq. (A.61)} = O_{P_{\sigma(\mathbf{z}_m)}}(\|\{\overline{\pi}^m - \overline{\pi}_0^m\}\|). \tag{A.64}$$

This completes the proof. $\qquad\square$

# B  Proofs

## B.1  Proof of Lemma 1

**Lemma 1 (c-component Identification [Jung et al., 2021b]).** *Let* $\mathbf{S}$ *denote a c-component in* $G_i := G(\mathbf{V}\backslash\mathbf{Z}_i)$ *for some* $\mathbf{Z}_i \in \mathbb{Z}$. *Let* $\mathbf{R} := pa(\mathbf{S})_{G_i}\backslash\mathbf{S}$. *Let* $(\mathbf{S}, \mathbf{R})$ *be ordered as* $(\mathbf{R}_0, S_1, \cdots, \mathbf{R}_{m-1}, S_m)$ *by* $\prec_G$. *Let* $\mathbf{A} \subseteq \mathbf{S}$ *denote a set satisfying* $\mathbf{A} = an(\mathbf{A})_{G_i(\mathbf{S})}$. *Let* $\mathbf{C} := (\mathbf{S}\backslash\mathbf{A})$. *Let* $\mathbb{Z}_0 := \{\mathbf{Z}_i\}$ *and* $\mathsf{seq}(\mathbb{Z}_0)$ *be a sequence of* $\mathbf{z}_i$ *repeating* $m$ *times. Then, the c-factor* $Q[\mathbf{A}]$ *is g-identifiable as follows:*

$$Q[\mathbf{A}] = A_0[\mathbf{S}, \mathbf{C}, \mathbf{R}; \mathbb{Z}_0 := \{\mathbf{Z}_i\}, \mathsf{seq}](\mathbf{a}, \mathbf{r}) = \sum_{\mathbf{c}\in\mathfrak{S}_\mathbf{C}} \prod_{j:V_j\in\mathbf{S}} P_{\mathbf{z}_i}(v_j|\overline{\mathbf{s}}^{j-1}, \overline{\mathbf{r}}^{j-1}\backslash\mathbf{z}_i). \tag{2}$$

*Proof of Lemma 1.* Let $\mathbf{C}_0 := \mathrm{pre}(\mathbf{A}; G(\mathbf{S})) \cap \mathbf{C}$. Let $\mathbf{C}_1 := \mathbf{C}\backslash\mathbf{C}_0$. We first note that, by [Jung et al., 2021b, Lemma 1],

$$Q[\mathbf{A}] = \sum_{\mathbf{c}_0 \mathfrak{S}_{\mathbf{C}_0}} \prod_{j:V_j\in\mathbf{A}\cup\mathbf{C}_0} P_{\mathbf{z}_i}(v_j|\overline{\mathbf{s}}^{j-1}, \overline{\mathbf{r}}^{j-1}\backslash\mathbf{z}_i). \tag{B.1}$$

Therefore, it suffices to show that

$$\text{Eq. (B.1)} = A_0[\mathbf{S}, \mathbf{R}; \mathbb{Z}_0 := \{\mathbf{Z}_i\}, \mathsf{seq}](\mathbf{s}\backslash\mathbf{c}, \mathbf{r}). \tag{B.2}$$

It holds as follows:

$$A_0[\mathbf{S}, \mathbf{R}; \mathbb{Z}_0 \coloneqq \{\mathbf{Z}_i\}, \mathtt{seq}](\mathbf{s}\backslash\mathbf{c}, \mathbf{r}) = \sum_{\mathbf{c} \in \mathfrak{S}_{\mathbf{C}}} \prod_{V_j \in \mathbf{S}} P_{\mathbf{z}_i}(v_j | \overline{\mathbf{s}}^{j-1}, \overline{\mathbf{r}}^{j-1}\backslash\mathbf{z}_i) \tag{B.3}$$

$$= \sum_{\mathbf{c}_0 \in \mathfrak{S}_{\mathbf{C}_0}} \sum_{\mathbf{c}_1 \in \mathfrak{S}_{\mathbf{C}_1}} \prod_{V_j \in \mathbf{S}} P_{\mathbf{z}_i}(v_j | \overline{\mathbf{s}}^{j-1}, \overline{\mathbf{r}}^{j-1}\backslash\mathbf{z}_i) \tag{B.4}$$

$$= \sum_{\mathbf{c}_0 \in \mathfrak{S}_{\mathbf{C}_0}} \prod_{V_j \in \mathbf{S}\backslash\mathbf{C}_1} P_{\mathbf{z}_i}(v_j | \overline{\mathbf{s}}^{j-1}, \overline{\mathbf{r}}^{j-1}\backslash\mathbf{z}_i) \tag{B.5}$$

$$= \sum_{\mathbf{c}_0 \in \mathfrak{S}_{\mathbf{C}_0}} \prod_{V_j \in \mathbf{A} \cup \mathbf{C}_0} P_{\mathbf{z}_i}(v_j | \overline{\mathbf{s}}^{j-1}, \overline{\mathbf{r}}^{j-1}\backslash\mathbf{z}_i) \tag{B.6}$$

$$= \text{Eq. (B.1).} \tag{B.7}$$

We note that the third equation holds since the all $P_{\mathbf{z}_i}(v_j | \overline{\mathbf{s}}^{j-1}, \overline{\mathbf{r}}^{j-1}\backslash\mathbf{z}_i)$ will be marginalized out if $V_j \in \mathbf{C}_1$. The fourth equation holds since $\mathbf{S} \coloneqq \mathbf{A} \uplus \mathbf{C}_0 \uplus \mathbf{C}_1$, which implies that $\mathbf{S}\backslash\mathbf{C}_1 = \mathbf{A} \cup \mathbf{C}_0$. $\qquad \square$

## B.2 Proof of Lemma 2

**Lemma 2 (Marginalization).** *Let $A_0[\mathbf{W}, \mathbf{C}, \mathbf{R}; \mathbb{Z}_0, \mathtt{seq}](\mathbf{w}\backslash\mathbf{c}, \mathbf{r})$ denote the g-mSBD operator in Def. 1. Let $\mathbf{W}_0 \subseteq \mathbf{W}\backslash\mathbf{C}$. Let $\mathbf{W}_{mar} \subseteq \{\mathbf{W}_0, \mathbf{C}\}$ denote the vector formed by the following procedure: Starting from $\mathbf{W}_{mar} = \emptyset$, for $j = m, \cdots, 1$, $\mathbf{W}_{mar} = \mathbf{W}_{mar} \cup \{W_j\}$ if (1) $W_j \in \{\mathbf{W}_0, \mathbf{C}\}$ and (2) $\exists k \in \{j, \cdots, m\}$ such that $\mathbf{R}_j, \cdots, \mathbf{R}_{k-1} = \emptyset$, $\overline{\mathbf{W}}^{k+1:m} \subseteq \mathbf{W}_{mar}$, and $\mathbf{Z}_k = \cdots = \mathbf{Z}_j$ and $\mathbf{z}_k = \cdots = \mathbf{z}_j$. Let $\mathbf{W}' \coloneqq \mathbf{W}\backslash\mathbf{W}_{mar}$, $\mathbf{R}' \coloneqq pre(\mathbf{W}'; G) \cap \mathbf{R}$ and $\mathbf{C}' \coloneqq \{\mathbf{W}_0, \mathbf{C}\}\backslash\mathbf{W}_{mar}$. Let $\mathbb{Z}' \subseteq \mathbb{Z}_0$ denote the collection of $\mathbf{Z}_i$ corresponding to the variable in $\mathbf{W}'$, and $\mathtt{seq}'$ the corresponding sequence. Then,*

$$\sum_{\mathbf{w}_0 \in \mathfrak{S}_{W_0}} A_0[\mathbf{W}, \mathbf{C}, \mathbf{R}; \mathbb{Z}_0, \mathtt{seq}](\mathbf{w}\backslash\mathbf{c}, \mathbf{r}) = A_0[\mathbf{W}', \mathbf{C}', \mathbf{R}'; \mathbb{Z}', \mathtt{seq}'](\mathbf{w}'\backslash\mathbf{c}', \mathbf{r}'). \tag{3}$$

***Proof of Lemma 2.*** Let $\mathbf{W}_{\mathrm{mar}}^c \coloneqq \{\mathbf{W}_0, \mathbf{C}\}\backslash\mathbf{W}_{\mathrm{mar}}$. We note that

$$\sum_{\mathbf{w}_0 \in \mathfrak{S}_{W_0}} A_0[\mathbf{W}, \mathbf{C}, \mathbf{R}; \mathbb{Z}_0, \mathtt{seq}](\mathbf{w}\backslash\mathbf{c}, \mathbf{r}) \tag{B.8}$$

$$= \sum_{\mathbf{w}_0, \mathbf{c}_0 \in \mathfrak{S}_{\mathbf{W}_0, \mathbf{C}_0}} \prod_{j:W_j \in \mathbf{W}} P_{\mathbf{z}_j}(w_j | \overline{\mathbf{w}}^{j-1}, \overline{\mathbf{r}}^{j-1}\backslash\mathbf{z}_j) \tag{B.9}$$

$$= \sum_{\mathbf{w}_{\mathrm{mar}}^c \mathfrak{S}_{\mathbf{W}_{\mathrm{mar}}^c}} \sum_{\mathbf{w}_{\mathrm{mar}} \mathfrak{S}_{\mathbf{W}_{\mathrm{mar}}}} \prod_{i=k+1}^{m} P_{\mathbf{z}_i}(w_i | \overline{\mathbf{w}}^{i-1}, \overline{\mathbf{r}}^{i-1}\backslash\mathbf{z}_i) P_{\mathbf{z}_j}(w_j, \cdots, w_k | \overline{\mathbf{w}}^{j-1}, \overline{\mathbf{r}}^{j-1}\backslash\mathbf{z}_j) \prod_{\ell=1}^{j-1} P_{\mathbf{z}_\ell}(w_\ell | \overline{\mathbf{w}}^{\ell-1}, \overline{\mathbf{r}}^{\ell-1}\backslash\mathbf{z}_\ell) \tag{B.10}$$

$$= \sum_{\mathbf{c}' \in \mathfrak{S}_{\mathbf{C}'}} P_{\mathbf{z}_j}(\{w_j, \cdots, w_k\}\backslash\mathbf{w}_{\mathrm{mar}} | \overline{\mathbf{w}}^{j-1}, \overline{\mathbf{r}}^{j-1}\backslash\mathbf{z}_j) \prod_{\ell=1}^{j-1} P_{\mathbf{z}_\ell}(w_\ell | \overline{\mathbf{w}}^{\ell-1}, \overline{\mathbf{r}}^{\ell-1}\backslash\mathbf{z}_\ell) \tag{B.11}$$

$$= \sum_{\mathbf{c}' \in \mathfrak{S}_{\mathbf{C}'}} \prod_{W_j \in \mathbf{W}\backslash\mathbf{W}_{\mathrm{mar}}} P_{\mathbf{z}_j}(w_j | \overline{\mathbf{w}}^{j-1}, \overline{\mathbf{r}}^{j-1}\backslash\mathbf{z}_j) \tag{B.12}$$

$$= A_0[\mathbf{W}', \mathbf{C}', \mathbf{R}'; \mathbb{Z}', \mathtt{seq}'](\mathbf{w}'\backslash\mathbf{c}', \mathbf{r}'). \tag{B.13}$$

$\qquad \square$

## B.3 Proof of Lemma 3

**Lemma 3 (Multiplication).** *Let $A_0^i \coloneqq A_0[\mathbf{W}_i, \emptyset, \mathbf{R}_i; \mathbb{Z}_i, \mathtt{seq}^i](\mathbf{w}_i, \mathbf{r}_i) \coloneqq \prod_{j=1}^{m^i} P_{\mathbf{z}_j^i}(w_{i,j} | \overline{\mathbf{w}}_i^{j-1}, \overline{\mathbf{r}}_i^{j-1}\backslash\mathbf{z}_j^i)$ for $i \in \{1, 2\}$ where $\mathtt{seq}^i \coloneqq (\mathbf{z}_j^i)_{j=1}^{m^i}$. Let $\mathbf{W} \coloneqq \mathbf{W}_1 \cup \mathbf{W}_2$. Let*

$\mathbf{R} := (\mathbf{R}_1 \cup \mathbf{R}_2) \backslash \mathbf{W}$. *Let* $(\mathbf{W}, \mathbf{R})$ *be ordered by* $\prec_G$. *Let* $\mathbb{Z} := \mathbb{Z}_1 \cup \mathbb{Z}_2$. *Assume the following: (1)* $\mathbf{W}_1 \cap \mathbf{W}_2 = \emptyset$; *and (2)* $\forall W_j \in \mathbf{W}, \exists W_{i,k} \in \mathbf{W}_i$ *such that* $(\overline{\mathbf{W}}^{j-1}, \overline{\mathbf{R}}^{j-1}) = (\overline{\mathbf{W}_i}^{k-1}, \overline{\mathbf{R}_i}^{k-1})$. *Let* $\mathtt{seq} := (\mathbf{z}_j)_{j:W_j \in \mathbf{W}}$ *where* $\mathbf{z}_j = \mathbf{z}_k^i$ *for all* $j$. *Then,*

$$A_0^1 \times A_0^2 = A_0[\mathbf{W}, \emptyset, \mathbf{R}; \mathbb{Z}, \mathtt{seq}](\mathbf{w}, \mathbf{r}) = \prod_{j:W_j \in \mathbf{W}} P_{\mathbf{z}_j}(w_j | \overline{\mathbf{w}}^{j-1}, \overline{\mathbf{r}}^{j-1} \backslash \mathbf{z}_j). \tag{4}$$

***Proof of Lemma 3.***

$$A_0[\mathbf{W}, \emptyset, \mathbf{R}; \mathbb{Z}, \mathtt{seq}](\mathbf{w}, \mathbf{r}) \tag{B.14}$$

$$= \prod_{j:W_j \in \mathbf{W}} P_{\mathbf{z}_j}(w_j | \overline{\mathbf{w}}^{j-1}, \overline{\mathbf{r}}^{j-1} \backslash \mathbf{z}_j) \tag{B.15}$$

$$= \prod_{k:W_{1,k} \in \mathbf{W}_1 \text{ s.t } W_{1,k}=W_j} P_{\mathbf{z}_k^1}(w_{1,k} | \overline{\mathbf{w}}_1^{j-1}, \overline{\mathbf{r}}_1^{j-1} \backslash \mathbf{z}_j^1) \times \prod_{k:W_{2,k} \in \mathbf{W}_2 \text{ s.t } W_{2,k}=W_j} P_{\mathbf{z}_k^2}(w_{2,k} | \overline{\mathbf{w}}_2^{j-1}, \overline{\mathbf{r}}_2^{j-1} \backslash \mathbf{z}_j^2) \tag{B.16}$$

$$= \prod_{j=1}^{m^1} P_{\mathbf{z}_j^1}(w_{1,j} | \overline{\mathbf{w}}_1^{j-1}, \overline{\mathbf{r}}_1^{j-1} \backslash \mathbf{z}_j^1) \times \prod_{j=1}^{m^2} P_{\mathbf{z}_j^2}(w_{2,j} | \overline{\mathbf{w}}_2^{j-1}, \overline{\mathbf{r}}_2^{j-1} \backslash \mathbf{z}_j^2) \tag{B.17}$$

$$= A_0^1 \times A_0^2. \tag{B.18}$$

$\square$

## B.4 Proof of Lemma 4

**Lemma 4 (Division).** *Let* $A_0^i := A_0[\mathbf{W}_i, \emptyset, \mathbf{R}_i; \mathbb{Z}_i, \mathtt{seq}^i](\mathbf{w}_i, \mathbf{r}_i) :=$ $\prod_{j=1}^{m^i} P_{\mathbf{z}_j^i}(w_{i,j} | \overline{\mathbf{w}}_i^{j-1}, \overline{\mathbf{r}}_i^{j-1} \backslash \mathbf{z}_j^i)$ *for* $i \in \{1, 2\}$ *where* $\mathtt{seq}^i := (\mathbf{z}_j^i)_{j=1}^{m^i}$. *Let* $\mathbf{W} := \mathbf{W}_1 \backslash \mathbf{W}_2$. *Let* $\mathbf{R} := (\mathbf{R}_1 \cup \mathbf{W}_2) \cap pre(\mathbf{W}; G)$. *Assume the following: (1)* $\mathbf{W}_2 \subseteq \mathbf{W}_1$; *and (2)* $\forall W_j \in \mathbf{W}, \exists W_{1,k} \in \mathbf{W}_1$ *such that* $(\overline{\mathbf{W}}^{j-1}, \overline{\mathbf{R}}^{j-1}) = (\overline{\mathbf{W}_1}^{k-1}, \overline{\mathbf{R}_1}^{k-1})$, $\mathbf{Z}_{i,k} = \mathbf{Z}_j$ *and* $\mathbf{z}_{i,k} = \mathbf{z}_j$. *Then,*

$$A_0^1 / A_0^2 = A_0[\mathbf{W}, \emptyset, \mathbf{R}; \mathbb{Z}_1, \mathtt{seq}^1](\mathbf{w}, \mathbf{r}) = \prod_{j:W_j \in \mathbf{W}} P_{\mathbf{z}_j}(w_j | \overline{\mathbf{w}}^{j-1}, \overline{\mathbf{r}}^{j-1} \backslash \mathbf{z}_j). \tag{5}$$

***Proof of Lemma 4.***

$$A_0^1 / A_0^2 = \frac{\prod_{j=1}^{m^1} P_{\mathbf{z}_j^1}(w_{1,j} | \overline{\mathbf{w}}_1^{j-1}, \overline{\mathbf{r}}_1^{j-1} \backslash \mathbf{z}_j^1)}{\prod_{j=1}^{m^2} P_{\mathbf{z}_j^2}(w_{2,j} | \overline{\mathbf{w}}_2^{j-1}, \overline{\mathbf{r}}_2^{j-1} \backslash \mathbf{z}_j^2)} \tag{B.19}$$

$$= \prod_{k:W_k \in \mathbf{W}_1 \backslash \mathbf{W}_2} P_{\mathbf{z}_k^1}(w_{1,k} | \overline{\mathbf{w}}_1^{k-1}, \overline{\mathbf{r}}_1^{k-1} \backslash \mathbf{z}_k^1) \tag{B.20}$$

$$= \prod_{k:W_k \in \mathbf{W}_1 \backslash \mathbf{W}_2} P_{\mathbf{z}_k^1}(w_{1,k} | \overline{\mathbf{w}}_1^{k-1} \cap \mathbf{w}, \overline{\mathbf{w}}_1^{k-1} \backslash \mathbf{w}, \overline{\mathbf{r}}_1^{k-1} \backslash \mathbf{z}_k^1). \tag{B.21}$$

We note that $\overline{\mathbf{W}_1}^{k-1} \cap \mathbf{W} = \overline{\mathbf{W}}^{j-1}$ for some $W_j \in \mathbf{W}$ s.t. $W_j = W_{1,k}$. Also, $(\mathbf{W}_1 \backslash \mathbf{W}) \cup \mathbf{R}_1 = \mathbf{W}_2 \cup \mathbf{R}_1$. Therefore, $\cup_{k:W_k \in \mathbf{W}_1 \backslash \mathbf{W}_2} \{\overline{\mathbf{W}_1}^{k-1} \backslash \mathbf{W}, \overline{\mathbf{R}_1}^{k-1}\} = \mathbf{R} := (\mathbf{R}_1 \cup \mathbf{W}_2) \cap pre(\mathbf{W}; G)$. Therefore,

$$A_0^1 / A_0^2 = \prod_{k:W_k \in \mathbf{W}_1 \backslash \mathbf{W}_2} P_{\mathbf{z}_k^1}(w_{1,k} | \overline{\mathbf{w}}_1^{k-1} \cap \mathbf{w}, \overline{\mathbf{w}}_1^{k-1} \backslash \mathbf{w}, \overline{\mathbf{r}}_1^{k-1} \backslash \mathbf{z}_k^1) \tag{B.22}$$

$$= \prod_{\ell:W_\ell \in \mathbf{W}} P_{\mathbf{z}_\ell}(w_\ell | \overline{\mathbf{w}}^{\ell-1}, \overline{\mathbf{r}}^{\ell-1}) \tag{B.23}$$

$$= A_0[\mathbf{W}, \emptyset, \mathbf{R}; \mathbb{Z}_1, \mathtt{seq}^1](\mathbf{w}, \mathbf{r}). \tag{B.24}$$

$\square$

## B.5 Proof of Theorem 1

**Theorem 1** (**Expression of g-Identifiable Causal Effects**). *Algo. 1 returns any g-identifiable causal effects as a function of a set $\{A_0^k\}$ of g-mSBD adjustment operators in the form*

$$P(\mathbf{y}|do(\mathbf{x})) = f(\{A_0^k\}_{k=1}^K), \tag{6}$$

*where the function $f(\cdot)$ applies marginalization, multiplication, or division over g-mSBD operators in $\{A_0^k\}$ as specified by Algo. 1.*

***Proof of Theorem 1.*** Throughout the proof, we refer to the algorithm developed in [Lee et al., 2019, Algo. 1] as the "standard gID" algorithm, in comparison to our gID algorithm presented in Algo. 1. It is established that the standard gID algorithm is sound, as stated in [Lee et al., 2019, Theorem 2]. This means that if the algorithm returns an identification expression, it must be correct. Furthermore, the standard gID algorithm is proven to be complete [Lee et al., 2019, Theorem 3]. In other words, the causal effect $P(\mathbf{y}|do(\mathbf{x}))$ is identifiable from $\mathbb{P}$ and the causal graph $G$ if and only if the standard gID algorithm does not return FAIL.

In our proof, we will show the soundness and completeness of Algo. 1 based on the foundation provided by the standard gID algorithm.

Algo. 1 is sound – If Algo. 1 returns an expression $f(\{A_0^k\}_{k=1}^K)$, then it holds that $f(\{A_0^k\}_{k=1}^K) = P(\mathbf{y}|do(\mathbf{x}))$. The soundness of Algo. 1 is derived from the soundness of Tian's c-factor operation, as demonstrated in [Tian and Pearl, 2003, Lemmas (3,4)] and Lemma 1.

We will now show that Algo. 1 is complete. Suppose there exists an input $(\mathbf{x}, \mathbf{y}, \mathbb{Z}, \mathbb{P}, G)$ for which the standard gID algorithm does not return FAIL while Algo. 1 does return FAIL. This implies the existence of $\mathbf{D}_j$ such that $Q[\mathbf{D}_j]$ is not identifiable from all $Q[\mathbf{S}_j^i]$ where $\mathbf{D}_j$ is a c-component in $G(\mathbf{D})$, and $\mathbf{S}_j^i$ is the c-component in $G(\mathbf{V}\backslash\mathbf{Z}_i)$ that contains $\mathbf{D}_j$. This observation is a consequence of the soundness and completeness of the SUBID procedure, as established in [Huang and Valtorta, 2006, Theorem 1].

It should be noted that $Q[\mathbf{D}_j]$ is not identifiable from $\mathbf{S}_j^i$ for all $\{i : \mathbf{Z}_i \in \mathbb{Z}\}$ only when there exists a c-component $\mathbf{T}_j^i$ in $G(\mathbf{S}_j^i)$ that serves as an ancestral set of $\mathbf{D}_j$ and includes $\mathbf{D}_j$. However, in such a scenario, the standard gID algorithm fails due to lines 12 and 13 of the algorithm. This contradicts the initial assumption that the standard gID algorithm does not return FAIL. Consequently, Algo. 1 returns FAIL whenever the standard gID algorithm does so. The completeness of the standard gID algorithm implies that Algo. 1 is complete in the g-identification task.

The fact that $f(\cdot)$ is a function involving marginalization, multiplications, and divisions of g-mSBD (generalized modified single back-door) operators is a consequence of applying Lemmas (2, 3, 4) within the algorithm. These lemmas establish the properties and operations of the g-mSBD operators, which are then utilized in the construction of $f(\cdot)$ in Algo. 1.

$\square$

## B.6 Proof of Proposition 1

We first restate the g-mSBD adjustment, its nuisances, and the g-mSBD estimator here:

**Definition 1** (**generalized-mSBD adjustment (g-mSBD)**). Let $(\mathbf{W}, \mathbf{R})$ be a disjoint pair in $\mathbf{V}$ topologically ordered as $(\mathbf{W}, \mathbf{R}) = \{\mathbf{R}_0, W_1, \cdots, \mathbf{R}_{m-1}, W_m, \mathbf{R}_m\}$ by $\prec_G$, where $\mathbf{R}_i$ can be empty. Let $\overline{\mathbf{W}}^{i-1} := \{W_j\}_{j=1}^{i-1}$ and $\overline{\mathbf{R}}^{i-1} := \{\mathbf{R}_j\}_{j=0}^{i-1}$ for $\forall i \in [m]$. Let $\mathbf{C} \subseteq \mathbf{W}$. Let $\mathbb{Z}_0 \subseteq \mathbb{Z}$ be some set such that $\forall \mathbf{Z} \in \mathbb{Z}_0, \mathbf{W} \cap \mathbf{Z} = \emptyset$. Let $\text{seq}(\mathbb{Z}_0)$ denote a sequence $(\mathbf{z}_1, \cdots, \mathbf{z}_m)$ where $\mathbf{z}_i$ denotes some realization of $\mathbf{Z}_i \in \mathbb{Z}_0$ (same $\mathbf{z}_i$ could appear multiple times in the sequence). Then, the g-mSBD adjustment is expressed as an operator $A_0[\mathbf{W}, \mathbf{C}, \mathbf{R}; \mathbb{Z}_0, \text{seq}, G](\mathbf{w}\backslash\mathbf{c}, \mathbf{r})$ defined by

$$A_0[\mathbf{W}, \mathbf{C}, \mathbf{R}; \mathbb{Z}_0, \text{seq}](\mathbf{w}\backslash\mathbf{c}, \mathbf{r}) := \sum_{\mathbf{c} \in \mathfrak{S}_{\mathbf{C}}} \prod_{i:W_i \in \mathbf{W}} P_{\mathbf{z}_i}(w_i|\overline{\mathbf{w}}^{i-1}, \overline{\mathbf{r}}^{i-1}\backslash\mathbf{z}_i). \tag{1}$$

**Definition 2** (**Nuisances for g-mSBD**). *Nuisances for g-mSBD $A_0$ in Eq. (1) are $\{\mu_0^{i+1}, \pi_0^i\}_{i=1}^{m-1}$ defined as follows. Let $\mu_0^{m+1} = \mu^{m+1} := \mathbb{1}_{\mathbf{w}\backslash\mathbf{c}}(\mathbf{W}\backslash\mathbf{C})$. For $i = m-1, \cdots, 1$,*

$$\mu_0^{i+1}(\overline{\mathbf{W}}^i, \overline{\mathbf{R}}^{1:i}) := \mathbb{E}_{P_{\sigma(\mathbf{Z}_{i+1})}}\left[\mu_0^{i+2}(\overline{\mathbf{W}}^{i+1}, \mathbf{r}_{i+1}, \overline{\mathbf{R}}^{1:i})|\overline{\mathbf{W}}^i, \overline{\mathbf{R}}^{1:i}, \mathbf{r}_0, \mathbf{z}_{i+1}\right] \tag{7}$$

$$\pi_0^i(\overline{\mathbf{W}}^i, \overline{\mathbf{R}}^{1:i}) := \frac{P_{\sigma(\mathbf{Z}_i)}(\overline{\mathbf{W}}^i, \overline{\mathbf{R}}^{1:i-1}|\mathbf{z}_i, \mathbf{r}_0)}{P_{\sigma(\mathbf{Z}_{i+1})}(\overline{\mathbf{W}}^i, \overline{\mathbf{R}}^{1:i-1}|\mathbf{z}_{i+1}, \mathbf{r}_0)} \frac{\mathbb{1}_{\mathbf{r}_i}(\mathbf{R}_i)}{P_{\sigma(\mathbf{Z}_{i+1})}(\mathbf{R}_i|\overline{\mathbf{W}}^i, \overline{\mathbf{R}}^{1:i-1}, \mathbf{z}_{i+1}, \mathbf{r}_0)}. \tag{8}$$

**Definition 3** (**DR-g-mSBD Estimators**). Let $D_{\sigma(\mathbf{Z}_i)}$ for $\mathbf{Z}_i \in \mathbb{Z}$ denote the experimental samples from randomizing the variable $\mathbf{Z}_i$. Let $\overline{D}_{\mathbf{z}_i}$ for $\mathbf{z}_i \in \mathfrak{D}_{\mathbf{Z}_i}$ denote the subsamples of $D_{\sigma(\mathbf{Z}_i)}$ fixing $\mathbf{R}_0\backslash\mathbf{Z}_i = \mathbf{r}_0\backslash\mathbf{z}_i$ and $\mathbf{Z}_i = \mathbf{z}_i$. The DR-g-mSBD estimator $\hat{A}$ for the g-mSBD adjustment $A_0[\mathbf{W}, \mathbf{C}, \mathbf{R}; \mathbb{Z}_0 := \{\mathbf{Z}_i\}_{i=1}^m, \mathtt{seq} := (\mathbf{z}_i)_{i=1}^m](\mathbf{w}\backslash\mathbf{c}, \mathbf{r})$ is defined as follows:

1. Randomly partition $\overline{D}_{\mathbf{z}_i}$ into $\{\overline{D}_{\mathbf{z}_i,\ell}\}_{\ell \in [L]}$; i.e., $\overline{D}_{\mathbf{z}_i} = \cup_{\ell=1}^L \overline{D}_{\mathbf{z}_i,\ell}$, $\forall \mathbf{Z}_i \in \mathbb{Z}$ and $\mathbf{z}_i \in \mathfrak{D}_{\mathbf{Z}_i}$.

2. For each fold $\ell \in [L]$, let $\mu_\ell^{i+1}$ denote learned $\mu_0^{i+1}$ using $\overline{D}_{\mathbf{z}_{i+1}}\backslash\overline{D}_{\mathbf{z}_{i+1},\ell}$ for $i = m, \cdots, 2$; and $\pi_\ell^i$ learned $\pi_0^i$ for $i = 1, \cdots, m-1$. Define $\check{\mu}_\ell^{i+1} := \mu_\ell^{i+1}(\overline{\mathbf{W}}^i, \mathbf{r}_i, \overline{\mathbf{R}}^{1:i-1})$ and $\overline{\pi}_\ell^i := \prod_{j=1}^i \pi_\ell^j$.

3. Estimate $\hat{A} := \hat{A}(\{\mu_\ell^{j+1}, \pi_\ell^j\}_{j\in[m-1],\ell\in[L]}) := (1/L)\sum_{\ell=1}^L \hat{A}_\ell(\{\mu_\ell^{j+1}, \pi_\ell^j\}_{j\in[m-1]})$ where

$$\hat{A}_\ell := \hat{A}_\ell(\{\mu_\ell^{j+1}, \pi_\ell^j\}_{j\in[m-1]}) := \sum_{j=1}^{m-1} \mathbb{E}_{\overline{D}_{\mathbf{z}_{j+1},\ell}}\left[\overline{\pi}_\ell^j\{\check{\mu}_\ell^{j+2} - \mu_\ell^{j+1}\}\right] + \mathbb{E}_{\overline{D}_{\mathbf{z}_1,\ell}}\left[\check{\mu}_\ell^2\right], \tag{9}$$

where $\mathbb{E}_{\overline{D}_{\mathbf{z}_j,\ell}}[\cdot]$ is an empirical average over samples $\overline{D}_{\mathbf{z}_j,\ell}$.

We analyze the bias of the g-mSBD estimator using the following results:

**Lemma S.2** (**Representation of g-mSBD**). *The g-mSBD adjustment $A_0$ in Def. 1 can be represented as*

$$A_0 = \sum_{i=1}^{m-1} \mathbb{E}_{P_{\sigma(\mathbf{Z}_{i+1})}}\left[\overline{\pi}_0^i\{\check{\mu}_0^{i+2} - \mu_0^{i+1}\}|\mathbf{z}_{i+1}, \mathbf{r}_0\right] + \mathbb{E}_{P_{\sigma(\mathbf{Z}_1)}}\left[\check{\mu}_0^2|\mathbf{z}_1, \mathbf{r}_0\right], \tag{B.25}$$

*where $\check{\mu}_\ell^{i+1}(\overline{\mathbf{W}}^i, \overline{\mathbf{R}}^{1:i-1}) := \mu_\ell^{i+1}(\overline{\mathbf{W}}^i, \mathbf{r}_i, \overline{\mathbf{R}}^{1:i-1})$ and $\overline{\pi}^i := \prod_{j=1}^i \pi^j$ as defined in Def. 3.*

***Proof of Lemma S.2.*** Throughout the proof, we will use $\mathbf{w}'\backslash\mathbf{c}'$ as some realization of $\mathbf{W}\backslash\mathbf{C}$. Recall that $\mathbb{1}_{\mathbf{w}\backslash\mathbf{c}}(\mathbf{w}'\backslash\mathbf{c}') = 1$ when $\mathbf{w}'\backslash\mathbf{c}' = \mathbf{w}\backslash\mathbf{c}$ and zero otherwise. We first recall that

$$A_0 = \sum_{\mathbf{w}'\in\mathfrak{S}_{\mathbf{W}}} \prod_{i:W_i\in\mathbf{W}} \mathbb{1}_{\mathbf{w}\backslash\mathbf{c}}(\mathbf{w}'\backslash\mathbf{c}')P_{\sigma(\mathbf{Z}_i)}(w_i'|\overline{\mathbf{w}'}^{i-1}, \overline{\mathbf{r}}^{i-1}, \mathbf{z}_i), \tag{B.26}$$

by the definition of the experimental distribution $P_{\sigma(\mathbf{Z}_i)}$.

For all $i = 1, \cdots, m-1$,

$$\mathbb{E}_{P_{\sigma(\mathbf{Z}_{i+1})}}\left[\overline{\pi}_0^i(\overline{\mathbf{W}}^i, \overline{\mathbf{R}}^{1:i})\{\check{\mu}_0^{i+2}(\overline{\mathbf{W}}^{i+1}, \overline{\mathbf{R}}^{1:i}) - \mu_0^{i+1}(\overline{\mathbf{W}}^i, \overline{\mathbf{R}}^{1:i})\}|\mathbf{z}_{i+1}, \mathbf{r}_0\right] \tag{B.27}$$

$$\overset{1}{=} \mathbb{E}_{P_{\sigma(\mathbf{Z}_{i+1})}}\left[\overline{\pi}_0^i(\overline{\mathbf{W}}^i, \overline{\mathbf{R}}^{1:i})\{\mathbb{E}_{P_{\sigma(\mathbf{Z}_{i+1})}}\left[\check{\mu}_0^{i+2}(\overline{\mathbf{W}}^{i+1}, \overline{\mathbf{R}}^{1:i})|\overline{\mathbf{W}}^i, \overline{\mathbf{R}}^{1:i}, \mathbf{z}_{i+1}, \mathbf{r}_0\right] - \mu_0^{i+1}(\overline{\mathbf{W}}^i, \overline{\mathbf{R}}^{1:i})\}|\mathbf{z}_{i+1}, \mathbf{r}_0\right] \tag{B.28}$$

$$\overset{2}{=} \mathbb{E}_{P_{\sigma(\mathbf{Z}_{i+1})}}\left[\overline{\pi}_0^i(\overline{\mathbf{W}}^i, \overline{\mathbf{R}}^{1:i})\{\mu_0^{i+1}(\overline{\mathbf{W}}^i, \overline{\mathbf{R}}^{1:i}) - \mu_0^{i+1}(\overline{\mathbf{W}}^i, \overline{\mathbf{R}}^{1:i})\}|\mathbf{z}_{i+1}, \mathbf{r}_0\right] \tag{B.29}$$

$$= 0, \tag{B.30}$$

where the equation $\overset{1}{=}$ holds by the total law of expectation, and $\overset{2}{=}$ holds by the definition of $\check{\mu}_0^{i+1}$.

It suffices to show that

$$\mathbb{E}_{P_{\sigma(\mathbf{Z}_1)}}\left[\mu_0^2(\overline{\mathbf{W}}^1, \overline{\mathbf{r}}^1)|\mathbf{z}_1, \mathbf{r}_0\right] = A_0 = \sum_{\mathbf{w}' \in \mathfrak{D}_{\mathbf{W}}} \mathbb{1}_{\mathbf{w}\backslash\mathbf{c}}(\mathbf{w}'\backslash\mathbf{c}') \prod_{j=1}^{m} P_{\sigma(\mathbf{Z}_j)}(w_j'|\overline{\mathbf{w}}'^{j-1}, \overline{\mathbf{r}}^{j-1}, \mathbf{z}_j).$$

(B.31)

To prove the equation, we show that, for all $k = m, m-1, \cdots, 2$,

$$\mu_0^k(\overline{\mathbf{W}}^{k-1}, \overline{\mathbf{R}}^{1:k-1}) = \sum_{\overline{\mathbf{w}}'^{k:m} \in \mathfrak{S}_{\overline{\mathbf{W}}^{k:m}}} \mathbb{1}_{\mathbf{w}\backslash\mathbf{c}}(\mathbf{w}'\backslash\mathbf{c}') \prod_{j=k}^{m} P_{\sigma(\mathbf{Z}_j)}(w_j'|\overline{\mathbf{W}}^{k-1}, \overline{\mathbf{R}}^{1:k-1}, \overline{\mathbf{w}}'^{k:j-1}, \overline{\mathbf{r}}^{k:j-1}, \mathbf{r}_0, \mathbf{z}_j).$$

(B.32)

This equation holds when $k = m$, because

$$\mu_0^m(\overline{\mathbf{W}}^{m-1}, \overline{\mathbf{R}}^{1:m-1}) := \mathbb{E}_{P_{\sigma(\mathbf{Z}_m)}}\left[\mathbb{1}_{\mathbf{w}\backslash\mathbf{c}}(\mathbf{w}'\backslash\mathbf{c}')|\overline{\mathbf{W}}^{m-1}, \overline{\mathbf{R}}^{1:m-1}, \mathbf{r}_0, \mathbf{z}_m\right]$$

(B.33)

$$= \sum_{w_m' \in \mathfrak{S}_{w_m}} \mathbb{1}_{\mathbf{w}\backslash\mathbf{c}}(\mathbf{w}'\backslash\mathbf{c}') P_{\sigma(\mathbf{Z}_m)}(w_m'|\overline{\mathbf{W}}^{m-1}, \overline{\mathbf{R}}^{1:m-1}, \mathbf{r}_0, \mathbf{z}_m).$$

(B.34)

For $k = m-1$,

$$\mu_0^{m-1}(\overline{\mathbf{W}}^{m-2}, \overline{\mathbf{R}}^{1:m-2})$$

(B.35)

$$:= \mathbb{E}_{P_{\sigma(\mathbf{Z}_{m-1})}}\left[\check{\mu}_0^m(\overline{\mathbf{W}}^{m-1}, \mathbf{r}_{m-1}, \overline{\mathbf{R}}^{1:m-2})|\overline{\mathbf{W}}^{m-2}, \overline{\mathbf{R}}^{1:m-2}, \mathbf{z}_{m-1}, \mathbf{r}_0\right]$$

(B.36)

$$= \sum_{\overline{\mathbf{w}}'^{m-1:m} \in \mathfrak{S}_{\overline{\mathbf{W}}^{m-1:m}}} \mathbb{1}_{\mathbf{w}\backslash\mathbf{c}}(\mathbf{w}'\backslash\mathbf{c}') \prod_{j=m-1}^{m} P_{\sigma(\mathbf{Z}_j)}(w_j'|\overline{\mathbf{w}}'^{m-1:j-1}, \overline{\mathbf{r}}^{m-1:j-1}, \overline{\mathbf{W}}^{m-2}, \overline{\mathbf{R}}^{1:m-2}, \mathbf{z}_j, \mathbf{r}_0)$$

(B.37)

Based on this observation, we make the following induction hypothesis: Suppose, for a fixed $k \in \{2, \cdots, m\}$, the following holds:

$$\mu_0^{k+1}(\overline{\mathbf{W}}^k, \overline{\mathbf{R}}^{1:k}) \overset{\text{induction}}{=} \sum_{\overline{\mathbf{w}}'^{k+1:m} \in \mathfrak{S}_{\overline{\mathbf{W}}^{k+1:m}}} \mathbb{1}_{\mathbf{w}\backslash\mathbf{c}}(\mathbf{w}'\backslash\mathbf{c}') \prod_{j=k+1}^{m} P_{\sigma(\mathbf{Z}_j)}(w_j'|\overline{\mathbf{W}}^k, \overline{\mathbf{R}}^{1:k}, \overline{\mathbf{w}}'^{k+1:j-1}, \overline{\mathbf{r}}^{k+1:j-1}, \mathbf{z}_j, \mathbf{r}_0).$$

(B.38)

Then, the induction hypothesis holds for $k - 1$ as follows:

$$\mu_0^k(\overline{\mathbf{W}}^{k-1}, \overline{\mathbf{R}}^{1:k-1})$$

(B.39)

$$:= \mathbb{E}_{P_{\sigma(\mathbf{Z}_{k+1})}}\left[\check{\mu}^{k+1}(\overline{\mathbf{W}}^k, \overline{\mathbf{R}}^{1:k-1}, \mathbf{r}_k)|\overline{\mathbf{W}}^{k-1}, \overline{\mathbf{R}}^{1:k-1}, \mathbf{z}_{k+1}, \mathbf{r}_0\right]$$

(B.40)

$$= \sum_{\overline{\mathbf{w}}'^{k:m} \in \mathfrak{S}_{\overline{\mathbf{W}}^{k:m}}} \mathbb{1}_{\mathbf{w}\backslash\mathbf{c}}(\mathbf{w}'\backslash\mathbf{c}') \prod_{j=k+1}^{m} P_{\sigma(\mathbf{Z}_j)}(w_j'|\overline{\mathbf{W}}^{k-1}, \overline{\mathbf{R}}^{1:k-1}, \overline{\mathbf{w}}'^{k:j-1}, \overline{\mathbf{r}}^{k:j-1}, \mathbf{z}_j, \mathbf{r}_0) P_{\sigma(\mathbf{Z}_k)}(w_k'|\overline{\mathbf{W}}^{k-1}, \overline{\mathbf{R}}^{1:k-1}, \mathbf{z}_k, \mathbf{r}_0)$$

(B.41)

$$= \sum_{\overline{\mathbf{w}}'^{k:m} \in \mathfrak{S}_{\overline{\mathbf{W}}^{k:m}}} \mathbb{1}_{\mathbf{w}\backslash\mathbf{c}}(\mathbf{w}'\backslash\mathbf{c}') \prod_{j=k}^{m} P_{\sigma(\mathbf{Z}_j)}(w_j'|\overline{\mathbf{W}}^{k-1}, \overline{\mathbf{R}}^{1:k-1}, \overline{\mathbf{w}}'^{k:j-1}, \overline{\mathbf{r}}^{k:j-1}, \mathbf{z}_j, \mathbf{r}_0).$$

(B.42)

Also, we already checked that the induction hypothesis holds for $k = m$. Therefore, the hypothesis holds for all $k = 2, \cdots, m$:

$$\mu_0^k(\overline{\mathbf{W}}^{k-1}, \overline{\mathbf{R}}^{1:k-1}) = \sum_{\overline{\mathbf{w}}'^{k:m} \in \mathfrak{S}_{\overline{\mathbf{W}}^{k:m}}} \mathbb{1}_{\mathbf{w}\backslash\mathbf{c}}(\mathbf{w}'\backslash\mathbf{c}') \prod_{j=k}^{m} P_{\sigma(\mathbf{Z}_j)}(w_j'|\overline{\mathbf{W}}^{k-1}, \overline{\mathbf{R}}^{1:k-1}, \overline{\mathbf{w}}'^{k:j-1}, \overline{\mathbf{r}}^{k:j-1}, \mathbf{z}_j, \mathbf{r}_0).$$

(B.43)

Then,

$$\mu_0^2(W_1, \mathbf{R}_1) = \sum_{\overline{\mathbf{w}}'^{2:m} \in \mathfrak{S}_{\overline{\mathbf{W}}^{2:m}}} \mathbb{1}_{\mathbf{w} \backslash \mathbf{c}}(\mathbf{w}' \backslash \mathbf{c}') \prod_{j=2}^{m} P_{\sigma(\mathbf{Z}_j)}(w_j' | W_1, \mathbf{R}_1, \overline{\mathbf{w}}'^{2:j-1}, \overline{\mathbf{r}}^{2:j-1}, \mathbf{z}_j, \mathbf{r}_0),$$

$$\text{(B.44)}$$

$$\mu_0^2(W_1, \mathbf{r}_1) = \sum_{\overline{\mathbf{w}}'^{2:m} \in \mathfrak{S}_{\overline{\mathbf{W}}^{2:m}}} \mathbb{1}_{\mathbf{w} \backslash \mathbf{c}}(\mathbf{w}' \backslash \mathbf{c}') \prod_{j=2}^{m} P_{\sigma(\mathbf{Z}_j)}(w_j' | W_1, \overline{\mathbf{w}}'^{2:j-1}, \overline{\mathbf{r}}^{j-1}, \mathbf{z}_j) \qquad \text{(B.45)}$$

Then,

$$\mathbb{E}_{P_{\sigma(\mathbf{Z}_1)}} \left[ \mu_0^2(\mathbf{W}_1, \mathbf{r}_1) | \mathbf{z}_1, \mathbf{r}_0 \right] \qquad \text{(B.46)}$$

$$= \sum_{\overline{\mathbf{w}}'^{m} \in \mathfrak{S}_{\overline{\mathbf{W}}^m}} \mathbb{1}_{\mathbf{w} \backslash \mathbf{c}}(\mathbf{w}' \backslash \mathbf{c}') \prod_{j=2}^{m} P_{\sigma(\mathbf{Z}_j)}(w_j' | w_1', \overline{\mathbf{w}}'^{2:j-1}, \overline{\mathbf{r}}^{j-1}, \mathbf{z}_j) P_{\sigma(\mathbf{Z}_1)}(w_1' | \mathbf{z}_1, \mathbf{r}_0) \qquad \text{(B.47)}$$

$$= \sum_{\overline{\mathbf{w}}'^{m} \in \mathfrak{S}_{\overline{\mathbf{W}}^m}} \mathbb{1}_{\mathbf{w} \backslash \mathbf{c}}(\mathbf{w}' \backslash \mathbf{c}') \prod_{j=1}^{m} P_{\sigma(\mathbf{Z}_j)}(w_j' | \overline{\mathbf{w}}'^{j-1}, \overline{\mathbf{r}}^{j-1}, \mathbf{z}_j) \qquad \text{(B.48)}$$

$$= A_0. \qquad \text{(B.49)}$$

$\square$

**Lemma S.3 (Bias Analysis of g-mSBD Estimators (1)).** *Let $\overline{A}$ be the quantity defined as*

$$\overline{A} := \sum_{i=1}^{m-1} \mathbb{E}_{P_{\sigma(\mathbf{z}_{i+1})}} \left[ \overline{\pi}^i \{ \check{\mu}^{i+2} - \mu^{i+1} \} | \mathbf{z}_{i+1}, \mathbf{r}_0 \right] + \mathbb{E}_{P_{\sigma(\mathbf{z}_1)}} \left[ \check{\mu}^2 | \mathbf{z}_1, \mathbf{r}_0 \right]. \qquad \text{(B.50)}$$

*For $i = 2, \cdots, m$,*

$$\mathbb{E}_{P_{\sigma(\mathbf{z}_{i+1})}} \left[ \overline{\pi}^i \{ \check{\mu}_0^{i+2} - \mu^{i+1} \} | \mathbf{z}_{i+1}, \mathbf{r}_0 \right] + \mathbb{E}_{P_{\sigma(\mathbf{z}_i)}} \left[ \overline{\pi}^{i-1} \{ \check{\mu}^{i+1} - \mu^i \} | \mathbf{z}_i, \mathbf{r}_0 \right] \qquad \text{(B.51)}$$
$$= \mathbb{E}_{P_{\sigma(\mathbf{z}_{i+1})}} \left[ \overline{\pi}^{i-1} \{ \mu_0^{i+1} - \mu^{i+1} \} \{ \pi^i - \pi_0^i \} | \mathbf{z}_{i+1}, \mathbf{r}_0 \right] + \mathbb{E}_{P_{\sigma(\mathbf{z}_i)}} \left[ \overline{\pi}^{i-1} \{ \check{\mu}_0^{i+1} - \mu^i \} | \mathbf{z}_i, \mathbf{r}_0 \right].$$

*Proof of Lemma S.3.* We first rewrite Eq. (B.51) as follows:

$$\text{Eq. (B.51)} = \mathbb{E}_{P_{\sigma(\mathbf{z}_{i+1})}} \left[ \overline{\pi}^i \{ \check{\mu}_0^{i+2} - \mu^{i+1} \} | \mathbf{z}_{i+1}, \mathbf{r}_0 \right] \qquad \text{(B.52)}$$
$$+ \mathbb{E}_{P_{\sigma(\mathbf{z}_i)}} \left[ \overline{\pi}^{i-1} \{ \check{\mu}^{i+1} - \check{\mu}_0^{i+1} \} | \mathbf{z}_i, \mathbf{r}_0 \right] \qquad \text{(B.53)}$$
$$+ \mathbb{E}_{P_{\sigma(\mathbf{z}_i)}} \left[ \overline{\pi}^{i-1} \{ \check{\mu}_0^{i+1} - \mu^i \} | \mathbf{z}_i, \mathbf{r}_0 \right]. \qquad \text{(B.54)}$$

Also,

Eq. (B.53)
$$= \mathbb{E}_{P_{\sigma(\mathbf{z}_i)}} \left[ \overline{\pi}^{i-1}(\overline{\mathbf{W}}^{i-1}, \overline{\mathbf{R}}^{1:i-1}) \{ \check{\mu}^{i+1}(\overline{\mathbf{W}}^i, \overline{\mathbf{R}}^{1:i-1}) - \check{\mu}_0^{i+1}(\overline{\mathbf{W}}^i, \overline{\mathbf{R}}^{1:i-1}) \} | \mathbf{z}_i, \mathbf{r}_0 \right]$$
$$= \mathbb{E}_{P_{\sigma(\mathbf{z}_{i+1})}} \left[ \overline{\pi}^{i-1}(\overline{\mathbf{W}}^{i-1}, \overline{\mathbf{R}}^{1:i-1}) \frac{P_{\sigma(\mathbf{Z}_i)}(\overline{\mathbf{W}}^i, \overline{\mathbf{R}}^{1:i-1} | \mathbf{z}_i, \mathbf{r}_0)}{P_{\sigma(\mathbf{Z}_{i+1})}(\overline{\mathbf{W}}^i, \overline{\mathbf{R}}^{1:i-1} | \mathbf{z}_{i+1}, \mathbf{r}_0)} \{ \check{\mu}^{i+1}(\overline{\mathbf{W}}^i, \overline{\mathbf{R}}^{1:i-1}) - \check{\mu}_0^{i+1}(\overline{\mathbf{W}}^i, \overline{\mathbf{R}}^{1:i-1}) \} | \mathbf{z}_{i+1}, \mathbf{r}_0 \right]$$
$$= \mathbb{E}_{P_{\sigma(\mathbf{z}_{i+1})}} \left[ \overline{\pi}^{i-1} \frac{P_{\sigma(\mathbf{Z}_i)}(\overline{\mathbf{W}}^i, \overline{\mathbf{R}}^{1:i-1} | \mathbf{z}_i, \mathbf{r}_0)}{P_{\sigma(\mathbf{Z}_{i+1})}(\overline{\mathbf{W}}^i, \overline{\mathbf{R}}^{1:i-1} | \mathbf{z}_{i+1}, \mathbf{r}_0)} \frac{\mathbb{1}_{\mathbf{r}_i}(\mathbf{R}_i)}{P_{\sigma(\mathbf{Z}_{i+1})}(\mathbf{R}_i | \overline{\mathbf{W}}^{i-1}, \overline{\mathbf{R}}^{1:i-1}, \mathbf{r}_0, \mathbf{z}_{i+1})} \{ \mu^{i+1} - \mu_0^{i+1} \} \Big| \mathbf{z}_{i+1}, \mathbf{r}_0 \right]$$
$$= \mathbb{E}_{P_{\sigma(\mathbf{z}_{i+1})}} \left[ \overline{\pi}^{i-1}(\overline{\mathbf{W}}^{i-1}, \overline{\mathbf{R}}^{1:i}) \pi_0^i(\overline{\mathbf{W}}^i, \overline{\mathbf{R}}^i) \{ \mu^{i+1}(\overline{\mathbf{W}}^i, \overline{\mathbf{R}}^i) - \mu_0^{i+1}(\overline{\mathbf{W}}^i, \overline{\mathbf{R}}^i) \} | \mathbf{z}_{i+1}, \mathbf{r}_0 \right]. \qquad \text{(B.55)}$$

Therefore,

Eq. (B.52) + Eq. (B.53)

$= \text{Eq. (B.52)} + \text{Eq. (B.55)}$

$= \mathbb{E}_{P_{\sigma(\mathbf{z}_{i+1})}}\left[\overline{\pi}^i\{\breve{\mu}_0^{i+2} - \mu^{i+1}\}|\mathbf{z}_{i+1},\mathbf{r}_0\right] + \text{Eq. (B.55)}$

$= \mathbb{E}_{P_{\sigma(\mathbf{z}_{i+1})}}\left[\overline{\pi}^i\{\mu_0^{i+1} - \mu^{i+1}\}|\mathbf{z}_{i+1},\mathbf{r}_0\right] + \text{Eq. (B.55)}$

$= \mathbb{E}_{P_{\sigma(\mathbf{z}_{i+1})}}\left[\overline{\pi}^i\{\mu_0^{i+1} - \mu^{i+1}\}|\mathbf{z}_{i+1},\mathbf{r}_0\right] + \mathbb{E}_{P_{\sigma(\mathbf{z}_{i+1})}}\left[\overline{\pi}^{i-1}\pi_0^i\{\mu^{i+1} - \mu_0^{i+1}\}|\mathbf{z}_{i+1},\mathbf{r}_0\right]$

$$= \mathbb{E}_{P_{\sigma(\mathbf{z}_{i+1})}}\left[\overline{\pi}^{i-1}\{\mu_0^{i+1} - \mu^{i+1}\}\{\pi^i - \pi_0^i\}|\mathbf{z}_{i+1},\mathbf{r}_0\right]. \tag{B.56}$$

Finally,

$$\text{Eq. (B.51)} = \text{Eq. (B.56)} + \text{Eq. (B.54)}.$$

$\square$

**Lemma S.4** (**Bias Analysis of g-mSBD Estimators (2)**). *Let $\overline{A}$ be the quantity defined as*

$$\overline{A} := \sum_{i=1}^{m-1} \mathbb{E}_{P_{\sigma(\mathbf{z}_{i+1})}}\left[\overline{\pi}^i\{\breve{\mu}^{i+2} - \mu^{i+1}\}|\mathbf{z}_{i+1},\mathbf{r}_0\right] + \mathbb{E}_{P_{\sigma(\mathbf{z}_1)}}\left[\breve{\mu}^2|\mathbf{z}_1,\mathbf{r}_0\right]. \tag{B.57}$$

*Then, for $k = 3, \cdots, m-1$,*

$$\overline{A} - A_0 = \sum_{r=k}^{m-1} \mathbb{E}_{P_{\sigma(\mathbf{z}_{r+1})}}\left[\overline{\pi}^{r-1}\{\mu_0^{r+1} - \mu^{r+1}\}\{\pi^r - \pi_0^r\}|\mathbf{z}_{r+1},\mathbf{r}_0\right] + \mathbb{E}_{P_{\sigma(\mathbf{z}_k)}}\left[\overline{\pi}^{k-1}\{\breve{\mu}_0^{k+1} - \mu^k\}|\mathbf{z}_k,\mathbf{r}_0\right]$$

$$+ \sum_{i=1}^{k-2} \mathbb{E}_{P_{\sigma(\mathbf{z}_{i+1})}}\left[\overline{\pi}^i\{\breve{\mu}^{i+2} - \mu^{i+1}\}|\mathbf{z}_{i+1},\mathbf{r}_0\right] + \mathbb{E}_{P_{\sigma(\mathbf{z}_1)}}\left[\breve{\mu}^2 - \breve{\mu}_0^2|\mathbf{z}_1,\mathbf{r}_0\right].$$

*Proof of Lemma S.4.* The equation holds for $k = m-1$. It can be shown as follows:

$\overline{A} - A_0$

$$= \sum_{i=1}^{m-1} \mathbb{E}_{P_{\sigma(\mathbf{z}_{i+1})}}\left[\overline{\pi}^i\{\breve{\mu}^{i+2} - \mu^{i+1}\}|\mathbf{z}_{i+1},\mathbf{r}_0\right] + \mathbb{E}_{P_{\sigma(\mathbf{z}_1)}}\left[\breve{\mu}^2 - \breve{\mu}_0^2|\mathbf{z}_1,\mathbf{r}_0\right]$$

$$= \mathbb{E}_{P_{\sigma(\mathbf{z}_{m+1})}}\left[\overline{\pi}^m\{\breve{\mu}_0^{m+2} - \mu^{m+1}\}|\mathbf{z}_{m+1},\mathbf{r}_0\right] + \mathbb{E}_{P_{\sigma(\mathbf{z}_m)}}\left[\overline{\pi}^{m-1}\{\breve{\mu}^{m+1} - \mu^m\}|\mathbf{z}_m,\mathbf{r}_0\right] \tag{B.58}$$

$$+ \sum_{i=1}^{m-2} \mathbb{E}_{P_{\sigma(\mathbf{z}_{i+1})}}\left[\overline{\pi}^i\{\breve{\mu}^{i+2} - \mu^{i+1}\}|\mathbf{z}_{i+1},\mathbf{r}_0\right] + \mathbb{E}_{P_{\sigma(\mathbf{z}_1)}}\left[\breve{\mu}^2 - \breve{\mu}_0^2|\mathbf{z}_1,\mathbf{r}_0\right]. \tag{B.59}$$

Then,

Eq. (B.58)

$$\stackrel{\text{Lemma S.3}}{=} \mathbb{E}_{P_{\sigma(\mathbf{z}_{m+1})}}\left[\overline{\pi}^{m-1}\{\mu_0^{m+1} - \mu^{m+1}\}\{\pi^m - \pi_0^m\}|\mathbf{z}_{m+1},\mathbf{r}_0\right] + \mathbb{E}_{P_{\sigma(\mathbf{z}_m)}}\left[\overline{\pi}^{m-1}\{\breve{\mu}_0^m - \mu^{m-1}\}|\mathbf{z}_m,\mathbf{r}_0\right].$$

$$\tag{B.60}$$

Therefore,

$\overline{A} - A_0$

$$= \text{Eq. (B.60)} + \sum_{i=1}^{m-2} \mathbb{E}_{P_{\sigma(\mathbf{z}_{i+1})}}\left[\overline{\pi}^i\{\breve{\mu}^{i+2} - \mu^{i+1}\}|\mathbf{z}_{i+1},\mathbf{r}_0\right] + \mathbb{E}_{P_{\sigma(\mathbf{z}_1)}}\left[\breve{\mu}^2 - \breve{\mu}_0^2|\mathbf{z}_1,\mathbf{r}_0\right]. \tag{B.61}$$

For any fixed $k + 1 \in \{m - 1, \cdots, 4\}$, suppose the following holds:

$$\overline{A} - A_0$$

$$= \sum_{r=k+1}^{m-1} \mathbb{E}_{P_{\sigma(\mathbf{z}_{r+1})}} \left[ \overline{\pi}^{r-1} \{ \mu_0^r - \mu^r \} \{ \pi^{r+1} - \pi_0^{r+1} \} | \mathbf{z}_{r+1}, \mathbf{r}_0 \right] \quad \text{(B.62)}$$

$$+ \mathbb{E}_{P_{\sigma(\mathbf{z}_{k+1})}} \left[ \overline{\pi}^k \{ \check{\mu}_0^{k+2} - \mu^{k+1} \} | \mathbf{z}_{k+1}, \mathbf{r}_0 \right] \quad \text{(B.63)}$$

$$+ \sum_{i=1}^{k-1} \mathbb{E}_{P_{\sigma(\mathbf{z}_{i+1})}} \left[ \overline{\pi}^i \{ \check{\mu}^{i+2} - \mu^{i+1} \} | \mathbf{z}_{i+1}, \mathbf{r}_0 \right] \quad \text{(B.64)}$$

$$+ \mathbb{E}_{P_{\sigma(\mathbf{z}_1)}} \left[ \check{\mu}^2 - \check{\mu}_0^2 | \mathbf{z}_1, \mathbf{r}_0 \right]. \quad \text{(B.65)}$$

We note that this holds when $k = m - 2$, as shown in Eq. (B.61). We will now show that it will hold for $k$, too. First,

Eq. (B.63) + Eq. (B.64) $\quad$ (B.66)

$$\mathbb{E}_{P_{\sigma(\mathbf{z}_{k+1})}} \left[ \overline{\pi}^k \{ \check{\mu}_0^{k+2} - \mu^{k+1} \} | \mathbf{z}_{k+1}, \mathbf{r}_0 \right] + \sum_{i=1}^{k-1} \mathbb{E}_{P_{\sigma(\mathbf{z}_{i+1})}} \left[ \overline{\pi}^i \{ \check{\mu}^{i+2} - \mu^{i+1} \} | \mathbf{z}_{i+1}, \mathbf{r}_0 \right]$$

$$= \mathbb{E}_{P_{\sigma(\mathbf{z}_{k+1})}} \left[ \overline{\pi}^k \{ \check{\mu}_0^{k+2} - \mu^{k+1} \} | \mathbf{z}_{k+1}, \mathbf{r}_0 \right] + \mathbb{E}_{P_{\sigma(\mathbf{z}_k)}} \left[ \overline{\pi}^{k-1} \{ \check{\mu}^{k+2} - \mu^{k+1} \} | \mathbf{z}_k, \mathbf{r}_0 \right]$$

$$+ \sum_{i=1}^{k-2} \mathbb{E}_{P_{\sigma(\mathbf{z}_{i+1})}} \left[ \overline{\pi}^i \{ \check{\mu}^{i+2} - \mu^{i+1} \} | \mathbf{z}_{i+1}, \mathbf{r}_0 \right],$$

$$\overset{\text{Lemma S.3}}{=} \mathbb{E}_{P_{\sigma(\mathbf{z}_{k+1})}} \left[ \overline{\pi}^{k-1} \{ \mu_0^{k+1} - \mu^{k+1} \} \{ \pi^k - \pi_0^k \} | \mathbf{z}_{k+1}, \mathbf{r}_0 \right] + \mathbb{E}_{P_{\sigma(\mathbf{z}_k)}} \left[ \overline{\pi}^{k-1} \{ \check{\mu}_0^{k+1} - \mu^k \} | \mathbf{z}_k, \mathbf{r}_0 \right]$$

$$+ \sum_{i=1}^{k-2} \mathbb{E}_{P_{\sigma(\mathbf{z}_{i+1})}} \left[ \overline{\pi}^i \{ \check{\mu}^{i+2} - \mu^{i+1} \} | \mathbf{z}_{i+1}, \mathbf{r}_0 \right]. \quad \text{(B.67)}$$

Therefore,

$$\overline{A} - A_0$$

$$= \text{Eq. (B.62)} + \text{Eq. (B.63)} + \text{Eq. (B.64)} + \text{Eq. (B.65)}$$

$$= \text{Eq. (B.62)} + \text{Eq. (B.67)} + \text{Eq. (B.65)}$$

$$= \sum_{r=k+1}^{m} \mathbb{E}_{P_{\sigma(\mathbf{z}_{r+1})}} \left[ \overline{\pi}^{r-1} \{ \mu_0^{r+1} - \mu^{r+1} \} \{ \pi^r - \pi_0^r \} | \mathbf{z}_{r+1}, \mathbf{r}_0 \right] + \mathbb{E}_{P_{\sigma(\mathbf{z}_{k+1})}} \left[ \overline{\pi}^{k-1} \{ \mu_0^{k+1} - \mu^{k_1} \} \{ \pi^k - \pi_0^k \} | \mathbf{z}_{k+1}, \mathbf{r}_0 \right]$$

$$+ \mathbb{E}_{P_{\sigma(\mathbf{z}_k)}} \left[ \overline{\pi}^{k-1} \{ \check{\mu}_0^{k+1} - \mu^k \} | \mathbf{z}_k, \mathbf{r}_0 \right]$$

$$+ \sum_{i=1}^{k-2} \mathbb{E}_{P_{\sigma(\mathbf{z}_{i+1})}} \left[ \overline{\pi}^i \{ \check{\mu}^{i+2} - \mu^{i+1} \} | \mathbf{z}_{i+1}, \mathbf{r}_0 \right]$$

$$+ \text{Eq. (B.65)}$$

Therefore, the equation holds for $k$, too. This completes the proof. $\quad\square$

**Lemma S.5 (Bias Analysis of g-mSBD Estimators (3)).**
$$\mathbb{E}_{P_{\sigma(\mathbf{z}_2)}} \left[ \pi^1 \{ \check{\mu}_0^3 - \mu^2 \} | \mathbf{z}_2, \mathbf{r}_0 \right] - \mathbb{E}_{P_{\sigma(\mathbf{z}_1)}} \left[ \check{\mu}^2 - \check{\mu}_0^2 | \mathbf{z}_1, \mathbf{r}_0 \right] = \mathbb{E}_{P_{\sigma(\mathbf{z}_2)}} \left[ \{ \mu_0^2 - \mu^2 \} \{ \pi^1 - \pi_0^1 \} | \mathbf{z}_2, \mathbf{r}_0 \right].$$

***Proof of Lemma S.5.*** Note that

$$\mathbb{E}_{P_{\sigma(\mathbf{z}_1)}} \left[ \breve{\mu}^2(\mathbf{W}_1, \mathbf{R}_0) - \breve{\mu}_0^2(\mathbf{W}_1, \mathbf{R}_0) | \mathbf{z}_1, \mathbf{r}_0 \right]$$

$$= \mathbb{E}_{P_{\sigma(\mathbf{z}_2)}} \left[ \frac{P_{\sigma(\mathbf{Z}_1)}(\mathbf{W}_1, \mathbf{R}_0 | \mathbf{z}_1, \mathbf{r}_0)}{P_{\sigma(\mathbf{Z}_2)}(\mathbf{W}_1, \mathbf{R}_0 | \mathbf{z}_2, \mathbf{r}_0)} \{ \breve{\mu}^2(\mathbf{W}_1, \mathbf{R}_0) - \breve{\mu}_0^2(\mathbf{W}_1, \mathbf{R}_0) \} \Big| \mathbf{z}_2, \mathbf{r}_0 \right]$$

$$= \mathbb{E}_{P_{\sigma(\mathbf{z}_2)}} \left[ \frac{P_{\sigma(\mathbf{Z}_1)}(\mathbf{W}_1, \mathbf{R}_0 | \mathbf{z}_1, \mathbf{r}_0)}{P_{\sigma(\mathbf{Z}_2)}(\mathbf{W}_1, \mathbf{R}_0 | \mathbf{z}_2, \mathbf{r}_0)} \frac{\mathbb{1}_{\mathbf{r}_1}(\mathbf{R}_1)}{P_{\sigma(\mathbf{Z}_2)}(\mathbf{R}_1 | \mathbf{W}_0, \mathbf{z}_2, \mathbf{r}_0)} \{ \mu^2(\mathbf{W}_1, \mathbf{R}_1) - \mu_0^2(\mathbf{W}_1, \mathbf{R}_1) \} \Big| \mathbf{z}_2, \mathbf{r}_0 \right]$$

$$= \mathbb{E}_{P_{\sigma(\mathbf{z}_2)}} \left[ \pi_0^1(\mathbf{W}_1, X_1) \{ \mu^2(\mathbf{W}_1, X_1) - \mu_0^2(\mathbf{W}_1, X_1) \} | \mathbf{z}_2, \mathbf{r}_0 \right].$$

Therefore,

$$\mathbb{E}_{P_{\sigma(\mathbf{z}_2)}} \left[ \pi^1 \{ \breve{\mu}_0^3 - \mu^2 \} | \mathbf{z}_2, \mathbf{r}_0 \right] - \mathbb{E}_{P_{\sigma(\mathbf{z}_1)}} \left[ \breve{\mu}^2 - \breve{\mu}_0^2 | \mathbf{z}_1, \mathbf{r}_0 \right]$$

$$= \mathbb{E}_{P_{\sigma(\mathbf{z}_2)}} \left[ \pi^1 \{ \mu_0^2 - \mu^2 \} | \mathbf{z}_2, \mathbf{r}_0 \right] - \mathbb{E}_{P_{\sigma(\mathbf{z}_1)}} \left[ \breve{\mu}^2 - \breve{\mu}_0^2 | \mathbf{z}_1, \mathbf{r}_0 \right]$$

$$= \mathbb{E}_{P_{\sigma(\mathbf{z}_2)}} \left[ \pi^1 \{ \mu_0^2 - \mu^2 \} | \mathbf{z}_2, \mathbf{r}_0 \right] - \mathbb{E}_{P_{\sigma(\mathbf{z}_2)}} \left[ \pi_0^1 \{ \mu_0^2 - \mu^2 \} | \mathbf{z}_2, \mathbf{r}_0 \right]$$

$$= \mathbb{E}_{P_{\sigma(\mathbf{z}_2)}} \left[ \{ \mu_0^2 - \mu^2 \} \{ \pi^1 - \pi_0^1 \} | \mathbf{z}_2, \mathbf{r}_0 \right].$$

$\square$

**Lemma S.6 (Bias Analysis of g-mSBD Estimators).** *Let $\overline{A}$ be the quantity defined as*

$$\overline{A} := \sum_{i=1}^{m-1} \mathbb{E}_{P_{\sigma(\mathbf{z}_{i+1})}} \left[ \overline{\pi}^i \{ \breve{\mu}^{i+2} - \mu^{i+1} \} | \mathbf{z}_{i+1}, \mathbf{r}_0 \right] + \mathbb{E}_{P_{\sigma(\mathbf{z}_1)}} \left[ \breve{\mu}^2 | \mathbf{z}_1, \mathbf{r}_0 \right], \quad \text{(B.68)}$$

*where $\pi^i, \mu^i$ are arbitrary nuisances for true nuisances $\pi_0^i$ and $\mu_0^i$ defined in Def. 2. Let $A_0$ denote the g-mSBD in Def. 1. Then,*

$$\overline{A} - A_0 = \sum_{r=1}^{m-1} O_{P_{\sigma(\mathbf{z}_{r+1})}} \left( \| \pi^r - \pi_0^r \| \| \mu^{r+1} - \mu_0^{r+1} \| \right). \quad \text{(B.69)}$$

***Proof of Lemma S.6.*** By Lemmas (S.3,S.4,S.5).

$\square$

We also use the following results which are used by Kennedy et al. [2020].

**Lemma S.7 (Decomposition).** *Let $\mathcal{D} \sim P$ denote a finite sample set following a distribution $P$. Let $h(V; \eta)$ denote an arbitrary random function taking $\eta$ as a nuisance. For any $\eta, \eta_0$,*

$$\mathbb{E}_{\mathcal{D}} \left[ h(\mathbf{V}; \eta) \right] - \mathbb{E}_P \left[ h(\mathbf{V}; \eta_0) \right] \quad \text{(B.70)}$$

$$= \mathbb{E}_{\mathcal{D}-P} \left[ h(V; \eta_0) \right] + \mathbb{E}_{\mathcal{D}-P} \left[ h(V; \eta) - h(V; \eta_0) \right] + \mathbb{E}_P \left[ h(V; \eta) - h(V; \eta_0) \right]. \quad \text{(B.71)}$$

***Proof of Lemma S.7.***

$$\mathbb{E}_{\mathcal{D}} \left[ h(\mathbf{V}; \eta) \right] - \mathbb{E}_P \left[ h(\mathbf{V}; \eta_0) \right] \quad \text{(B.72)}$$

$$= \mathbb{E}_{\mathcal{D}-P} \left[ h(\mathbf{V}; \eta_0) \right] + \mathbb{E}_{\mathcal{D}} \left[ h(\mathbf{V}; \eta) - h(\mathbf{V}; \eta_0) \right] \quad \text{(B.73)}$$

$$= \mathbb{E}_{\mathcal{D}-P} \left[ h(\mathbf{V}; \eta_0) \right] + \mathbb{E}_{\mathcal{D}-P} \left[ h(\mathbf{V}; \eta) - h(\mathbf{V}; \eta_0) \right] + \mathbb{E}_P \left[ h(\mathbf{V}; \eta) - h(\mathbf{V}; \eta_0) \right]. \quad \text{(B.74)}$$

$\square$

**Lemma S.8 (Continuous Mapping Theorem for $L_2(P)$).** *Let $X_n, X$ denote a random sequence defined on a metric space $S$. Suppose a function $g : S \to S'$ (where $S'$ is another metric space) is bounded and continuous almost everywhere. Then,*

$$X_n \overset{L_2(P)}{\to} X \implies g(X_n) \overset{L_2(P)}{\to} g(X). \quad \text{(B.75)}$$

**Proof of Lemma S.8.** We first note that $X_n \overset{L_2(P)}{\to} X$ implies $X_n \overset{p}{\to} X$. Then, by continuous mapping theorem, $g(X_n) \overset{p}{\to} g(X)$. Then,

$$\lim_{n\to\infty} \|g(X_n) - g(X)\|^2 = \lim_{n\to\infty} \int_{\mathcal{X}} |g(X_n) - g(X)|^2 \ d[P] \overset{*}{=} \int_{\mathcal{X}} \lim_{n\to\infty} |g(X_n) - g(X)|^2 \ d[P] = 0,$$

(B.76)

where the equation $\overset{*}{=}$ holds by the dominated convergence theorem, which is applicable since $g(X_n), g(X)$ are bounded functions (from the given condition) and $X_n \overset{p}{\to} X$. $\qquad\square$

**Proposition 1** (**Asymptotic Analysis of g-mSBD Estimators**). *Assume that the nuisance estimates* $\mu_\ell^i$ *and* $\pi_\ell^i$ *are* $L_2$-*consistent; i.e.,* $\|\mu_\ell^{i+1} - \mu_0^{i+1}\|_{P_{\sigma(\mathbf{z}_{i+1})}} = o_{P_{\sigma(\mathbf{z}_{i+1})}}(1)$, $\|\check{\mu}_\ell^{i+2} - \check{\mu}_0^{i+2}\|_{P_{\sigma(\mathbf{z}_{i+1})}} = o_{P_{\sigma(\mathbf{z}_{i+1})}}(1)$ *and* $\|\pi_\ell^i - \pi_0^i\|_{P_{\sigma(\mathbf{z}_{i+1})}} = o_{P_{\sigma(\mathbf{z}_{i+1})}}(1)$ *for* $i = 1, \cdots, m-1$, *and* $\|\check{\mu}_\ell^2 - \check{\mu}_0^2\|_{P_{\sigma(\mathbf{Z}_1)}} = o_{P_{\sigma(\mathbf{Z}_1)}}(1)$. *Let* $n_i := |\overline{D}_{\mathbf{z}_i}|$ *for* $i \in \{1, \cdots, m\}$. *Then,*

$$\hat{A} - A_0 = \sum_{i=1}^m R_i + \frac{1}{L} \sum_{\ell=1}^L \sum_{i=1}^{m-1} O_{P_{\sigma(\mathbf{Z}_{i+1})}} \left( \|\mu_\ell^{i+1} - \mu_0^{i+1}\| \|\pi_\ell^i - \pi_0^i\| \right),$$

(10)

*where* $R_i$ *is a random variable such that* $n_i^{1/2} R_i$ *converges in distribution to a mean-zero normal random variable.*

**Proof of Proposition 1.** We start the proof by noting that

$$\hat{A} - A_0 = \frac{1}{L} \sum_{\ell=1}^L \{\hat{A}_\ell - A_0\},$$

(B.77)

where $\hat{A}_\ell$ is defined in Def. 3. This proof focuses on analyzing $\hat{A}_\ell - A_0$.

Also, we recall that $\overline{D}_{\mathbf{z}_i,\ell}$ for all $\mathbf{z}_i$ follows the distribution $P_{\sigma(\mathbf{Z}_i)}(\mathbf{V}|\mathbf{r}_0, \mathbf{z}_i)$. Throughout the proof, we will denote

$$P_{\sigma(\mathbf{Z}_{i+1})|\mathbf{z}_{i+1},\mathbf{r}_0}(\mathbf{V}) := P_{\sigma(\mathbf{Z}_{i+1})}(\mathbf{V}|\mathbf{z}_{i+1}, \mathbf{r}_0).$$

(B.78)

Then, each $\hat{A}_\ell - A_0$ in Eq. (B.77) is given as follow:

$$\hat{A}_\ell - A_0$$

$$= \sum_{i=1}^{m-1} \mathbb{E}_{\overline{D}_{\mathbf{z}_{i+1},\ell} - P_{\sigma(\mathbf{Z}_{i+1})|\mathbf{z}_{i+1},\mathbf{r}_0}} \left[ \overline{\pi}_0^i \{ \check{\mu}_0^{i+2} - \mu_0^{i+1} \} \right] + \mathbb{E}_{\overline{D}_{\mathbf{z}_1,\ell} - P_{\sigma(\mathbf{Z}_1)|\mathbf{z}_1,\mathbf{r}_0}} \left[ \check{\mu}_0^2 \right]$$

(B.79)

$$+ \sum_{i=1}^{m-1} \mathbb{E}_{\overline{D}_{\mathbf{z}_{i+1},\ell} - P_{\sigma(\mathbf{Z}_{i+1})|\mathbf{z}_{i+1},\mathbf{r}_0}} \left[ \overline{\pi}_\ell^i \{ \check{\mu}_\ell^{i+2} - \mu_\ell^{i+1} \} - \overline{\pi}_0^i \{ \check{\mu}_0^{i+2} - \mu_0^{i+1} \} \right]$$

$$+ \mathbb{E}_{\overline{D}_{\mathbf{z}_1,\ell} - P_{\sigma(\mathbf{Z}_1)|\mathbf{z}_1,\mathbf{r}_0}} \left[ \check{\mu}_\ell^2 - \check{\mu}_0^2 \right]$$

(B.80)

$$+ \sum_{i=1}^{m-1} \mathbb{E}_{P_{\sigma(\mathbf{Z}_{i+1})}} \left[ \overline{\pi}_\ell^i \{ \check{\mu}_\ell^{i+2} - \mu_\ell^{i+1} \} - \overline{\pi}_0^i \{ \check{\mu}_0^{i+2} - \mu_0^{i+1} \} | \mathbf{z}_{i+1}, \mathbf{r}_0 \right] + \mathbb{E}_{P_{\sigma(\mathbf{Z}_1)}} \left[ \check{\mu}_\ell^2 - \check{\mu}_0^2 | \mathbf{z}_1, \mathbf{r}_0 \right].$$

(B.81)

Define

$$R_{1,\ell}^a := \mathbb{E}_{\overline{D}_{\mathbf{z}_1,\ell} - P_{\sigma(\mathbf{Z}_1)|\mathbf{z}_1,\mathbf{r}_0}} \left[ \check{\mu}_0^2 \right]$$

(B.82)

and for $i = 1, \cdots, m-1$,

$$R_{i+1,\ell}^a := \mathbb{E}_{\overline{D}_{\mathbf{z}_{i+1},\ell} - P_{\sigma(\mathbf{Z}_{i+1})|\mathbf{z}_{i+1},\mathbf{r}_0}} \left[ \overline{\pi}_0^i \{ \check{\mu}_0^{i+2} - \mu_0^{i+1} \} \right].$$

(B.83)

By the central limit theorem, we note that $R_{i,\ell}^a$ for $i = 1, \cdots, m$ is a random variable such that $n_{i,\ell}^{1/2} R_{i,\ell}^a$ converges in distribution to a mean-zero normal random variable. Therefore,

$$\text{Eq. (B.79)} = \sum_{i=1}^{m} R_{i,\ell}^a, \tag{B.84}$$

also behaves the same.

We now analyze the second term. Define

$$R_{1,\ell}^b := \mathbb{E}_{\overline{D}_{\mathbf{z}_{1,\ell}} - P_{\sigma(\mathbf{z}_1)|\mathbf{z}_1, \mathbf{r}_0}} \left[ \check{\mu}_\ell^2 - \check{\mu}_0^2 \right], \tag{B.85}$$

and for $i = 1, 2, \cdots, m-1$,

$$R_{i+1,\ell}^b := \mathbb{E}_{\overline{D}_{\mathbf{z}_{i+1,\ell}} - P_{\sigma(\mathbf{z}_{i+1})|\mathbf{z}_{i+1}, \mathbf{r}_0}} \left[ \overline{\pi}_\ell^i \{ \check{\mu}_\ell^{i+2} - \mu_\ell^{i+1} \} - \overline{\pi}_0^i \{ \check{\mu}_0^{i+2} - \mu_0^{i+1} \} \right]. \tag{B.86}$$

By [Kennedy et al., 2020, Lemma 2] and the continuous mapping theorem in Lemma S.8,

$$R_{1,\ell}^b = O_{P_{\sigma(\mathbf{z}_1)}} \left( \frac{\| \check{\mu}_\ell^2 - \check{\mu}_0^2 \|}{\sqrt{n_{1,\ell}}} \right), \tag{B.87}$$

and for $i = 1, \cdots, m-1$,

$$R_{i+1,\ell}^b = O_{P_{\sigma(\mathbf{z}_{i+1})}} \left( \frac{\| \| \overline{\pi}_\ell^i \{ \check{\mu}_\ell^{i+2} - \mu_\ell^{i+1} \} - \overline{\pi}_0^i \{ \check{\mu}_0^{i+2} - \mu_0^{i+1} \} \| \|}{\sqrt{n_{i+1,\ell}}} \right). \tag{B.88}$$

Under the given assumption, for $i = 1, 2, \cdots, m$,

$$R_{i,\ell}^b = o_{P_{\sigma(\mathbf{z}_i)}}(1), \tag{B.89}$$

and

$$\text{Eq. (B.80)} = \sum_{i=1}^{m} o_{P_{\sigma(\mathbf{z}_i)}}(1). \tag{B.90}$$

Define

$$R_{i,\ell} := R_{i,\ell}^a + R_{i,\ell}^b. \tag{B.91}$$

Then, $R_{i,\ell}$ is also a random variable such that $n_{i,\ell}^{1/2} R_{i,\ell}$ converges in distribution to a mean-zero normal random variable, by Slutsky's theorem.

We now analyze the third term. By Lemma S.6, the third term can be analyzed as follow:

$$\text{Eq. (B.81)} = \sum_{i=1}^{m-1} O_{P_{\sigma(\mathbf{z}_{i+1})}} \left( \| \mu_\ell^{i+1} - \mu_0^{i+1} \| \| \pi_\ell^i - \pi_0^i \| \right). \tag{B.92}$$

Therefore,

$$\hat{A}_\ell - A_0 = \text{Eq. (B.79)} + \text{Eq. (B.80)} + \text{Eq. (B.81)} \tag{B.93}$$

$$= \sum_{i=1}^{m} R_{i,\ell} + \sum_{i=1}^{m-1} O_{P_{\sigma(\mathbf{z}_{i+1})}} \left( \| \mu_\ell^{i+1} - \mu_0^{i+1} \| \| \pi_\ell^i - \pi_0^i \| \right). \tag{B.94}$$

Define $R_i := (1/L) \sum_{\ell=1}^{L} R_{i,\ell}$. We note that $R_i$ is a random variable such that $n_i^{1/2} R_i$ converges in distribution to a mean-zero normal random variable. Then,

$$\hat{A} - A_0 = \frac{1}{L} \sum_{\ell=1}^{L} \{\hat{A}_\ell - A_0\} \tag{B.95}$$

$$= \sum_{i=1}^{m} R_i + \frac{1}{L} \sum_{\ell=1}^{L} \sum_{i=1}^{m-1} O_{P_{\sigma(\mathbf{Z}_{i+1})}} \left( \|\pi_\ell^i - \pi_0^i\| \|\mu_\ell^{i+1} - \mu_0^{i+1}\| \right). \tag{B.96}$$

$\square$

## B.7 Proof of Theorem 2

We first restate the definition of the MR-gID estimator, the theorem, and its corresponding assumptions.

**Definition 4** (**MR-gID Estimator**). The MR-gID estimator $\hat{\psi}$ for the identification expression of the causal effect $\psi_0 := f(\{A_0^k\}_{k=1}^K)$ in Theorem 1 is given as follows: For each $A_0^k$ composing $f(\{A_0^k\}_{k=1}^K)$, let $\hat{A}^k := \hat{A}^k(\{\mu_{k,\ell}^{j+1}, \pi_{k,\ell}^j\}_{j \in [m^k - 1], \ell \in [L]})$ denote the DR-g-mSBD estimator with nuisance estimates $\{\mu_{k,\ell}^{j+1}, \pi_{k,\ell}^j\}$ for the true nuisances $\{\mu_{k,0}^{j+1}, \pi_{k,0}^j\}$. Then,

$$\hat{\psi} := f(\{\hat{A}^k\}_{k=1}^K). \tag{11}$$

**Assumption 2** (**Analysis of MR-gID**). *The identification function* $f(\{A^k\}_{k=1}^K)$ *in Thm. 1 and each nuisances* $\{\mu_{k,\ell}^{i+1}, \pi_{k,\ell}^i\}_{k,\ell}$ *for* $\hat{A}^k$ *satisfy the following properties:*

1. *Twice differentiability*: $f(\{A^k\}_{k=1}^K)$ *is twice continuously Frechet differentiable w.r.t.* $\{A^k\}_{k=1}^K$ *w.r.t.* $\{A^k\}_{k=1}^K$.

2. *Boundedness*: $\forall k \in [K]$ *and* $\forall \mathbf{Z}_i \in \mathbb{Z}$, $\nabla_{A^k} f(\{A_0^j\}_{j=1}^K)[\hat{A}^k - A_0^k] = O_{P_{\sigma(\mathbf{z}_i)}}(\hat{A}^k - A_0^k)$.

3. $L_2$-*Consistency*: $\|\mu_{k,\ell}^{i+1} - \mu_{k,0}^{i+1}\|_{P_{\sigma(\mathbf{z}_{i+1}^k)}} = o_{P_{\sigma(\mathbf{z}_{i+1}^k)}}(1)$, $\|\breve{\mu}_{k,\ell}^{i+2} - \breve{\mu}_{k,0}^{i+2}\|_{P_{\sigma(\mathbf{z}_{i+1}^k)}} = o_{P_{\sigma(\mathbf{z}_{i+1}^k)}}(1)$, $\|\pi_{k,\ell}^i - \pi_{k,0}^i\|_{P_{\sigma(\mathbf{z}_{i+1}^k)}} = o_{P_{\sigma(\mathbf{z}_{i+1}^k)}}(1)$, *and* $\|\breve{\mu}_{k,\ell}^2 - \breve{\mu}_{k,0}^2\|_{P_{\sigma(\mathbf{z}_1^k)}} = o_{P_{\sigma(\mathbf{z}_1^k)}}(1)$.

**Theorem 2** (**Asymptotic Analysis of MR-gID**). *Suppose Assumption 2 holds. Let* $n_{k,i} := |\overline{D}_{\mathbf{z}_i^k}|$ *for* $\mathbf{Z}_i^k \in \mathbb{Z}$ *and* $\mathbf{z}_i^k \in \mathfrak{D}_{\mathbf{Z}_i^k}$. *Let* $\hat{\psi}$ *denote the MR-gID estimator in Def. 4 for the causal effect* $\psi_0 := f(\{A_0^k\}_{k=1}^K)$ *in Theorem 1. Then, the error of* $\hat{\psi}$ *is given as*

$$\hat{\psi} - \psi_0 = \sum_{k=1}^{K} \sum_{i=1}^{m^k} O_{P_{\sigma(\mathbf{z}_i^k)}}(n_{k,i}^{-1/2}) + \frac{1}{L} \sum_{k=1}^{K} \sum_{\ell=1}^{L} \sum_{i=1}^{m^k-1} O_{P_{\sigma(\mathbf{z}_i^k)}}(\|\mu_{k,\ell}^{i+1} - \mu_{k,0}^{i+1}\| \|\pi_{k,\ell}^i - \pi_{k,0}^i\|). \tag{12}$$

***Proof of Theorem 2.*** We first define the following notation. For a map $g(x)$, we will use $\nabla_x g(x_0)[h] := \lim_{t \to 0} (g(x_0 + th) - g(x_0))/t$. We first note that by the definition of Fréchet Derivative-based Taylor expansion [Blanchard and Brüning, 2015, Def. 34.1] and the given assumption ('Twice differentiability'), the error $\hat{\psi} - \psi_0$ can be represented as follow:

$$\hat{\psi} - \psi_0 = \sum_{k=1}^{K} \nabla_{A^k} f(\{A_0^j\}_{j=1}^K)[\hat{A}^k - A_0^k] + o(\hat{A}^k - A_0^k), \tag{B.97}$$

where, by Prop. 1,

$$\hat{A}^k - A_0^k = \sum_{i=1}^{m^k} O_{P_{\sigma(\mathbf{z}_i^k)}}(n_{k,i}^{-1/2}) + \frac{1}{L} \sum_{\ell=1}^{L} \sum_{i=1}^{m^k-1} O_{P_{\sigma(\mathbf{z}_{i+1}^k)}}(\|\mu_{k,\ell}^{i+1} - \mu_{k,0}^{i+1}\| \|\pi_{k,\ell}^i - \pi_{k,0}^i\|). \tag{B.98}$$

Therefore, by the big O in probability calculus [Van der Vaart, 2000, Chap. 2],

$$o(\hat{A}^k - A_0^k) = o\left(\sum_{i=1}^{m^k} O_{P_{\sigma(\mathbf{z}_i^k)}}(n_{k,i}^{-1/2}) + \frac{1}{L}\sum_{\ell=1}^{L}\sum_{i=1}^{m^k-1} O_{\overline{P}_{\mathbf{z}_{i+1}^k}}(\|\mu_{k,\ell}^{i+1} - \mu_{k,0}^{i+1}\|\|\pi_{k,\ell}^i - \pi_{k,0}^i\|).\right)$$
(B.99)

$$= \sum_{i=1}^{m^k} o_{P_{\sigma(\mathbf{z}_i^k)}}(n_{k,i}^{-1/2}) + \frac{1}{L}\sum_{\ell=1}^{L}\sum_{i=1}^{m^k-1} o_{P_{\sigma(\mathbf{z}_{i+1}^k)}}(\|\mu_{k,\ell}^{i+1} - \mu_{k,0}^{i+1}\|\|\pi_{k,\ell}^i - \pi_{k,0}^i\|).$$
(B.100)

By applying the given assumption ('Boundedness'), we have the following:

$$\nabla_{A^k} f(\{A_0^j\}_{j=1}^m)[\hat{A}^k - A_0^k] = \sum_{i=1}^{m^k} O_{P_{\sigma(\mathbf{z}_i^k)}}(n_{k,i}^{-1/2})$$

$$+ \frac{1}{L}\sum_{\ell=1}^{L}\sum_{i=1}^{m^k-1} O_{P_{\sigma(\mathbf{z}_{i+1}^k)}}(\|\mu_{k,\ell}^{i+1} - \mu_{k,0}^{i+1}\|\|\pi_{k,\ell}^i - \pi_{k,0}^i\|). \quad \text{(B.101)}$$

Therefore,

$$\hat{\psi} - \psi_0 = \sum_{k=1}^{K} \nabla_{A^k} f(\{A_0^j\}_{j=1}^K)[\hat{A}^k - A_0^k] + o(\hat{A}^k - A_0^k) \tag{B.102}$$

$$= \sum_{k=1}^{K}\sum_{i=1}^{m^k} O_{P_{\sigma(\mathbf{z}_i^k)}}(n_{k,i}^{-1/2}) + \frac{1}{L}\sum_{k=1}^{K}\sum_{\ell=1}^{L}\sum_{i=1}^{m^k-1} O_{P_{\sigma(\mathbf{z}_i^k)}}(\|\mu_{k,\ell}^{i+1} - \mu_{k,0}^{i+1}\|\|\pi_{k,\ell}^i - \pi_{k,0}^i\|)$$
(B.103)

$$+ \sum_{k=1}^{K}\sum_{i=1}^{m^k} o_{P_{\sigma(\mathbf{z}_i^k)}}(n_{k,i}^{-1/2}) + \frac{1}{L}\sum_{k=1}^{K}\sum_{\ell=1}^{L}\sum_{i=1}^{m^k-1} o_{P_{\sigma(\mathbf{z}_i^k)}}(\|\mu_{k,\ell}^{i+1} - \mu_{k,0}^{i+1}\|\|\pi_{k,\ell}^i - \pi_{k,0}^i\|)$$
(B.104)

$$= \sum_{k=1}^{K}\sum_{i=1}^{m^k} O_{P_{\sigma(\mathbf{z}_i^k)}}(n_{k,i}^{-1/2}) + \frac{1}{L}\sum_{k=1}^{K}\sum_{\ell=1}^{L}\sum_{i=1}^{m^k-1} O_{P_{\sigma(\mathbf{z}_i^k)}}(\|\mu_{k,\ell}^{i+1} - \mu_{k,0}^{i+1}\|\|\pi_{k,\ell}^i - \pi_{k,0}^i\|).$$
(B.105)

$$\square$$

## B.8   Proof of Corollary 2

**Corollary 2** (**Multiply Robustness** (Corollary of Thm. 2)). *Suppose (1) Assumption 2 holds; (2) Either $\pi_{k,\ell}^i = \pi_{k,0}^i$ or $\mu_{k,\ell}^j = \mu_{k,0}^j$ for $j = i+1, \cdots, m^k$ for all $i, \ell, k$; and (3) all nuisances $\{\pi_{k,\ell}^i, \mu_{k,\ell}^{i+1}\}_{i,\ell,k}$ are bounded by some constant. Then, the MR-gID $\hat{\psi}$ (Def. 4) is consistent to $\psi_0$.*

***Proof of Corollary 2.*** We first note that $f$ is a continuous function under the twice differentiability condition in Assumption 2. Suppose each $\hat{A}^k$ is a consistent estimator of $A_0^k$ under given conditions. Then, by the continuous mapping theorem, $f(\{\hat{A}^k\}_{k=1}^K)$ is consistent to $P_{\mathbf{x}}(\mathbf{y}) = f(\{A_0^k\}_{k=1}^K)$. Therefore, it suffices to show that each $\hat{A}^k$ is a consistent estimator of $A_0^k$.

We first recall that $\hat{A}^k \coloneqq (1/L) \sum_{\ell=1}^{L} \hat{A}_\ell^k$ by Def. 4. By applying Lemma S.7, $\hat{A}_\ell^k - A_0^k$ can be rewritten as follow:

$$\hat{A}_\ell^k - A_0$$

$$= \sum_{i=1}^{m-1} \mathbb{E}_{\overline{D}_{\mathbf{z}_{i+1}^k, \ell} - P_{\sigma(\mathbf{z}_{i+1}^k)|\mathbf{z}_{i+1}^k, \mathbf{r}_0}} \left[ \overline{\pi}_{k,0}^i \{ \check{\mu}_{k,0}^{i+2} - \mu_{k,0}^{i+1} \} \right] + \mathbb{E}_{\overline{D}_{\mathbf{z}_{1,\ell}^k} - P_{\sigma(\mathbf{z}_1^k)|\mathbf{z}_1^k, \mathbf{r}_0}} \left[ \check{\mu}_{k,0}^2 \right] \qquad \text{(B.106)}$$

$$+ \sum_{i=1}^{m-1} \mathbb{E}_{\overline{D}_{\mathbf{z}_{i+1}^k, \ell} - P_{\sigma(\mathbf{z}_{i+1}^k)|\mathbf{z}_{i+1}^k, \mathbf{r}_0}} \left[ \overline{\pi}_{k,\ell}^i \{ \check{\mu}_{k,\ell}^{i+2} - \mu_{k,\ell}^{i+1} \} - \overline{\pi}_{k,0}^i \{ \check{\mu}_{k,0}^{i+2} - \mu_{k,0}^{i+1} \} \right]$$

$$+ \mathbb{E}_{\overline{D}_{\mathbf{z}_{1,\ell}^k} - P_{\sigma(\mathbf{z}_1^k)|\mathbf{z}_1^k, \mathbf{r}_0}} \left[ \check{\mu}_{k,\ell}^2 - \check{\mu}_{k,0}^2 \right] \qquad \text{(B.107)}$$

$$+ \sum_{i=1}^{m-1} \mathbb{E}_{P_{\sigma(\mathbf{z}_{i+1}^k)}} \left[ \overline{\pi}_{k,\ell}^i \{ \check{\mu}_{k,\ell}^{i+2} - \mu_{k,\ell}^{i+1} \} - \overline{\pi}_{k,0}^i \{ \check{\mu}_{k,0}^{i+2} - \mu_{k,0}^{i+1} \} | \mathbf{r}_0, \mathbf{z}_{i+1}^k \right]$$

$$+ \mathbb{E}_{P_{\sigma(\mathbf{z}_1^k)}} \left[ \check{\mu}_{k,\ell}^2 - \check{\mu}_{k,0}^2 | \mathbf{r}_0, \mathbf{z}_{i+1}^k \right] \qquad \text{(B.108)}$$

We first note that all term in Eq. (B.106) converges in the mean-zero normal distribution and is bounded in probability at $n_{i+1,\ell,k}^{-1/2}$ rate. Therefore,

$$\text{Eq. (B.106)} = \sum_{i=1}^{m^k} O_{P_{\sigma(\mathbf{z}_i^k)}}(n_{i,\ell,k}^{-1/2}). \qquad \text{(B.109)}$$

We now analyze the second term. We note that, by [Kennedy et al., 2020, Lemma 2],

$$\mathbb{E}_{\overline{D}_{\mathbf{z}_{i+1}^k, \ell} - P_{\sigma(\mathbf{z}_{i+1}^k)|\mathbf{z}_{i+1}^k, \mathbf{r}_0}} \left[ \overline{\pi}_{k,\ell}^i \{ \check{\mu}_{k,\ell}^{i+2} - \mu_{k,\ell}^{i+1} \} - \overline{\pi}_{k,0}^i \{ \check{\mu}_{k,0}^{i+2} - \mu_{k,0}^{i+1} \} \right] \qquad \text{(B.110)}$$

$$= O_{P_{\sigma(\mathbf{z}_{i+1}^k)}} \left( \frac{\| \overline{\pi}_{k,\ell}^i \{ \check{\mu}_{k,\ell}^{i+2} - \mu_{k,\ell}^{i+1} \} - \overline{\pi}_{k,0}^i \{ \check{\mu}_{k,0}^{i+2} - \mu_{k,0}^{i+1} \} \|}{\sqrt{n_{i+1,\ell,k}}} \right). \qquad \text{(B.111)}$$

We note that $\| \overline{\pi}_{k,\ell}^i \{ \check{\mu}_{k,\ell}^{i+2} - \mu_{k,\ell}^{i+1} \} - \overline{\pi}_{k,0}^i \{ \check{\mu}_{k,0}^{i+2} - \mu_{k,0}^{i+1} \} \|$ is bounded by some constant by the given condition. Therefore, it is bounded in probability at $n_{i+1,\ell,k}^{-1/2}$ rate. By the same analysis, $\mathbb{E}_{\overline{D}_{\mathbf{z}_{1,\ell}^k} - P_{\sigma(\mathbf{z}_1^k)|\mathbf{z}_1^k, \mathbf{r}_0}} \left[ \check{\mu}_{k,\ell}^2 - \check{\mu}_{k,0}^2 \right]$ is bounded in probability at $1/\sqrt{n_{1,\ell,k}}$-rates. Therefore,

$$\text{Eq. (B.107)} = \sum_{i=1}^{m^k} O_{P_{\sigma(\mathbf{z}_i^k)}}(n_{i,\ell,k}^{-1/2}). \qquad \text{(B.112)}$$

Finally, under the given assumption, the third term can be analyzed by Lemma S.6 and is zero:

$$\text{Eq. (B.108)} = \sum_{i=1}^{m^k-1} O_{P_{\sigma(\mathbf{z}_{i+1}^k)}} \left( \| \mu_{k,\ell}^{i+1} - \mu_{k,0}^{i+1} \| \| \pi_{k,\ell}^i - \pi_{k,0}^i \| \right) = 0. \qquad \text{(B.113)}$$

Therefore,

$$\hat{A}_\ell^k - A_0 = \text{Eq. (B.106)} + \text{Eq. (B.107)} + \text{Eq. (B.108)} = \sum_{i=1}^{m} \sum_{i=1}^{m^k} O_{P_{\sigma(\mathbf{z}_i^k)}}(n_{i,\ell,k}^{-1/2}), \qquad \text{(B.114)}$$

and, finally,

$$\hat{A} - A_0 = \frac{1}{L} \sum_{\ell=1}^{L} \{\hat{A}_\ell^k - A_0\} \tag{B.115}$$

$$= \frac{1}{L} \sum_{\ell=1}^{L} \sum_{i=1}^{m^k} O_{P_{\sigma(\mathbf{z}_i^k)}}(n_{i,\ell,k}^{-1/2}) \tag{B.116}$$

$$= \sum_{i=1}^{m^k} O_{P_{\sigma(\mathbf{z}_i^k)}}(n_{i,k}^{-1/2}) \tag{B.117}$$

$$= \sum_{i=1}^{m} o_{P_{\sigma(\mathbf{z}_i^k)}}(1), \tag{B.118}$$

where the last equation holds since $O_P(n^{-\alpha}) = o_P(1)$ when $\alpha > 0$. $\qquad\square$

## C   Discussion

### C.1   Relaxation of Discreteness Assumption

In this paper, we made the strong assumption that all variables are discrete. However, this assumption does not hold in general. In this section, we relax the assumption to a certain degree while ensuring that the proposed estimators and corresponding error analysis remain applicable without sacrificing generality.

First, we define a set of variables, denoted as `disc`, which must be discrete in order to apply the proposed estimators and leverage the error analyses presented in the paper.

**Definition 4** (**Discreteness set `disc`**)**.** For some given inputs $(\mathbf{x}, \mathbf{y}, \mathbb{Z}, \mathbb{P}, G)$, suppose

$$f(\{A_0^k\}_{k=1}^{K}) = \texttt{GID}(\mathbf{x}, \mathbf{y}, \mathbb{Z}, \mathbb{P}, G), \tag{C.1}$$

where each $A_0^k$ is specified as

$$A_0^k \coloneqq A_0[\mathbf{W}^k, \mathbf{C}^k, \mathbf{R}^k; \mathbb{Z}^k, \texttt{seq}^k](\mathbf{w}^k \backslash \mathbf{c}^k, \mathbf{r}_k), \tag{C.2}$$

for $\mathbf{W}^k, \mathbf{R}^k \subseteq \mathbf{V}$ and $\mathbb{Z}^k \subseteq \mathbb{Z}$. Then, the discreteness set $\texttt{disc}(\{A_0^k\}_{k=1}^{K})$ is defined as follow:

$$\texttt{disc}(\{A_0^k\}_{k=1}^{K}) \coloneqq \bigcup_{k=1}^{K} \{(\mathbf{W}^k \backslash \mathbf{C}^k) \cup \mathbf{R}^k \cup \mathbb{Z}^k\}. \tag{C.3}$$

For Example 1, the discreteness set is given as

$$\texttt{disc}(\{A_0^1, A_0^2\}) = (Z, X) \cup (Y, Z) = \{Z, X, Y\} = \mathbf{V} \backslash \{W\}. \tag{C.4}$$

For Example 2, the discreteness set is given as

$$\texttt{disc}(\{A_0^2, A_0^{13}\}) = (W, R, X_1) \cup (R, Y, X_2, W, X_1) = \{X_1, X_2, R, W, Y\} = \mathbf{V} \tag{C.5}$$

Equipped with the discreteness set, we relax the assumption as follows:

**Assumption 1.1** (**Relaxed Regularity**)**.** *For variables* $\mathbf{V}$ *and the Radon-Nikodym derivative* $p_{\sigma(\mathbf{Z})}$ *of* $P_{\sigma(\mathbf{Z})}$ *for* $\mathbf{Z} \in \mathbb{Z}$*, the following conditions hold:*

1. *All variables in* $\texttt{disc}(\{A_0^k\})$ *are discrete;*

2. $p_{\sigma(\mathbf{Z})}(\mathbf{v}) > c, \forall \mathbf{v} \in \mathfrak{D}_{\mathbf{V}}$ *for some* $c \in (0, 1)$*.*

We note that the proposed estimator is well-defined and corresponding error analyses Theorem 2 and Corollary 2 hold true under the relaxed assumption:

**Lemma S.9** (**Well-defined MR-gID Estimator under Relaxed Regularity**). *The MR-gID estimator in Definition 4 is pathwise-differentiable under Assumption 1.1 and Assumption 2.*

*Proof of Lemma S.9.* For $k = 1, \cdots, K$, define

$$\overline{A}^k := \sum_{i=1}^{m^k-1} \mathbb{E}_{P_{\sigma(\mathbf{Z}_{i+1}^k)}} \left[ \overline{\pi\mathbf{k}}^i \{ \check{\mu}_k^{i+2} - \mu_k^{i+1} \} | \mathbf{z}_{i+1}^k, \mathbf{r}_0^k \right] + \mathbb{E}_{P_{\sigma(\mathbf{Z}_1^k)}} \left[ \check{\mu}_k^2 | \mathbf{z}_1^k, \mathbf{r}_0^k \right]. \tag{C.6}$$

To establish the pathwise-differentiability of the MR-gID estimator $f(\{\hat{A}^k\}_{k=1}^K)$ as defined in Definition 4, it is sufficient to ensure the pathwise-differentiability of individual $\overline{A}^k$. Under Assumption 1.1, $\mu_k^{m^k+1} := \mathbb{1}_{\mathbf{w}^k \setminus \mathbf{c}^k}(\mathbf{W}^k \setminus \mathbf{C}^k)$ is well-defined since $(\mathbf{W}^k \setminus \mathbf{C}^k) \in \mathtt{disc}(\{A_0^k\}_{k=1}^K)$ are discrete. Also, each $\mathbb{1}_{\mathbf{r}_i^k}(\mathbf{R}_i^k)$ in each $\pi_k^i$ are well-defined since $\mathbf{R}_i^k \in \mathtt{disc}(\{A_0^k\}_{k=1}^K)$ are discrete. Finally, the conditional expectation $\mathbb{E}_{P_{\sigma(\mathbf{Z}_i^k)}}[\cdot | \mathbf{z}_i^k, \mathbf{r}_0^k]$ is well-defined since $\mathbf{Z}_i^k \in \mathtt{disc}(\{A_0^k\}_{k=1}^K)$ are discrete. Also, under the positivity condition stated in Assumption 1.1, $\overline{A}^k$ in Eq. (C.6) is pathwise-differentiable. By combining this with Assumption 2, we conclude that the MR-gID estimator is pathwise-differentiable. $\qquad\square$

## C.2 Sequential Doubly Robustness: $2^{m-1}$ robustness versus $m$-robustness

In this section, we discuss the practical properties of the proposed doubly robust g-mSBD estimator in Def. 3. We recall that the estimator is doubly robust by the analysis in Lemma S.6 as follows:

**Lemma S.6** (**Bias Analysis of g-mSBD Estimators**). *Let $\overline{A}$ be the quantity defined as*

$$\overline{A} := \sum_{i=1}^{m-1} \mathbb{E}_{P_{\sigma(\mathbf{z}_{i+1})}} \left[ \overline{\pi}^i \{ \check{\mu}^{i+2} - \mu^{i+1} \} | \mathbf{z}_{i+1}, \mathbf{r}_0 \right] + \mathbb{E}_{P_{\sigma(\mathbf{Z}_1)}} \left[ \check{\mu}^2 | \mathbf{z}_1, \mathbf{r}_0 \right], \tag{B.68}$$

*where $\pi^i, \mu^i$ are arbitrary nuisances for true nuisances $\pi_0^i$ and $\mu_0^i$ defined in Def. 2. Let $A_0$ denote the g-mSBD in Def. 1. Then,*

$$\overline{A} - A_0 = \sum_{r=1}^{m-1} O_{P_{\sigma(\mathbf{z}_{r+1})}} \left( \| \pi^r - \pi_0^r \| \| \mu^{r+1} - \mu_0^{r+1} \| \right). \tag{B.69}$$

The term in Eq. (B.69) exhibits doubly robustness, becoming zero when either $\pi^r = \pi_0^r$ or $\mu^{r+1} = \mu_0^{r+1}$ hold for all $r = 1, 2, \ldots, m-1$. This phenomenon is referred to as the sequential doubly robustness [Luedtke et al., 2017], or $2^{m-1}$ robustness in the sense that there are $2^{m-1}$ ways to make Eq. (B.69) zero [Vansteelandt et al., 2007, Rotnitzky et al., 2017].

While the proposed doubly robust g-mSBD estimator defined in Definition 3 exhibits doubly robustness, as shown in Proposition 1, it does not satisfy $2^{m-1}$ robustness. This is due to the dependencies between $\mu_\ell^r$ and $\mu_\ell^s$ for $s \in \{r+1, \cdots, m\}$. Specifically, if $\mu_\ell^s$ is misspecified for some $s > r$, it renders the case $\mu_\ell^r = \mu_0^r$ impossible. Consequently, instead of having $2^{m-1}$ possibilities, there are only $m$ ways to make Eq. (B.69) equal to zero. For each $r = 1, \cdots, m-1$, this requires either $\pi_\ell^r = \pi_0^r$ or $\mu_\ell^s = \mu_0^s$ for $s > r$. This condition is referred to as $m$-robustness. In summary, the doubly robust g-mSBD estimator achieves $m$-robustness instead of $2^{m-1}$ robustness. We acknowledge that an interesting open direction is to explore ways to enhance the doubly robust g-mSBD estimator to attain $2^{m-1}$ robustness, building upon the findings presented in Luedtke et al. [2017].

# D  Details of Experiments

As described in Sec. 4, we used the XGBoost [Chen and Guestrin, 2016] as a model for estimating nuisances $\mu, \pi, \{\mu^i\}_{i=2}^m, \{\pi^i\}_{i=1}^m$. We implemented the model using Python. In modeling nuisance using the XGBoost, we used the command

`xgboost.XGBClassifier(eval_metric='logloss')`[1] to use the XGBoost with the default parameter settings. In estimating the weight, we set the weight $\pi_\ell^i = 10$ whenever the estimated weight is over 10 [Crump et al., 2009]. For Example 1, the dimension of $W$ is set as $|W| = 10$. We chose $L = 2$. All variables are set to be binary. We compute the effect of $P(Y = 1|do(\mathbf{x}))$.

## D.1 Designs of Simulations

In this section, we present the structural causal models (SCMs) utilized for generating the dataset. Furthermore, we include a segment of the code employed to generate the dataset.

### D.1.1 Example 1

We define the following structural causal models for Example 1:

$$
\begin{aligned}
U_W, U_{WZ}, U_{WY}, U_{XY} &\sim \texttt{normal}(0,1), \\
\mathbf{W} &:= f_W(U_{WZ}, U_{WY}, U_W), \\
X &:= f_X(\mathbf{W}, U_{XY}), \\
Z &:= f_Z(W, X, U_{WZ}), \\
Y &:= f_Y(Z, U_{WY}, U_{XY}),
\end{aligned}
$$

where

$$
\begin{aligned}
f_W(U_{WZ}, U_{WY}) &:= \left\lfloor \frac{1}{1 + \exp(U_W - U_{WZ} + U_{WY})} \right\rfloor, \\
f_X(\mathbf{W}, U_{XY}) &:= \left\lfloor \frac{1}{1 + \exp(\mathbf{c}_X^\intercal \mathbf{W} + U_{XY})} \right\rfloor, \\
f_Z(W, X, U_{WZ}) &:= \left\lfloor \frac{1}{1 + \exp(U_{WZ} \cdot \mathbf{c}_Z^\intercal \mathbf{W} + 4X - 2 + U_{WZ})} \right\rfloor \\
f_Y(Z, U_{WY}, U_{XY}) &:= \left\lfloor \frac{1}{1 + \exp(4Z - 2 + 0.5U_{WY} - U_{XY})} \right\rfloor,
\end{aligned}
$$

where the coefficient vector $\mathbf{c}_X, \mathbf{c}_Z$ are such that their $i$th element is 0 if $i$ is an even number and 1 otherwise.

### D.1.2 Example 2

We define the following structural causal models for Example 2:

$$
\begin{aligned}
U_{X_1,X_2}, U_{X_1,W}, U_{X_1,R}, U_{X_2,W}, U_{X_2,Y} &\sim \texttt{normal}(0,1), \\
X_1 &:= f_{X_1}(U_{X_1,X_2}), \\
X_2 &:= f_{X_2}(U_{X_1,X_2}), \\
R &:= f_R(U_{X_1,R}, X_1, X_2), \\
W &:= f_W(U_{X_1,W}, U_{X_2,W}, X_1, R), \\
Y &:= f_Y(U_{X_2,Y}, X_2, R, W),
\end{aligned}
$$

---

[1]Detailed parametrization of parameters including learning rates, maximum depth of the trees, etc. are explained in `https://xgboost.readthedocs.io/en/stable/python/python_api.html#xgboost.XGBClassifier`.

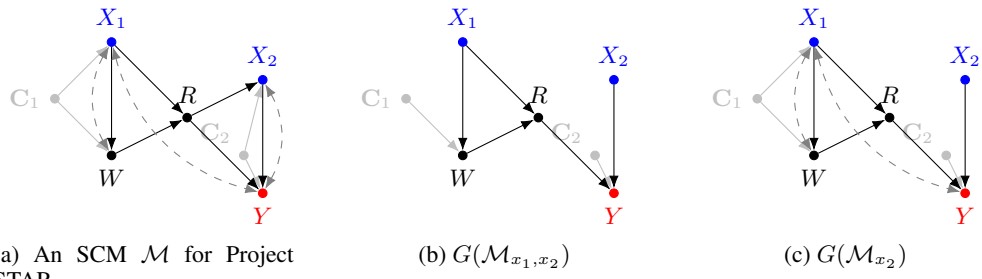

(a) An SCM $\mathcal{M}$ for Project STAR

(b) $G(\mathcal{M}_{x_1, x_2})$

(c) $G(\mathcal{M}_{x_2})$

Figure E.4: Example causal graphs for Section E. Nodes representing the treatment and outcome are marked in blue and red, respectively.

where

$$f_{X_1}(U_{X_1, X_2}) := \left\lfloor \frac{1}{1 + \exp(2U_{X_1, X_2} - 1)} \right\rfloor,$$

$$f_{X_2}(U_{X_1, X_2}) := \left\lfloor \frac{1}{1 + \exp(3U_{X_1, X_2} + 1)} \right\rfloor,$$

$$f_R(U_{X_1, R}, X_1, X_2) := \left\lfloor \frac{1}{1 + \exp(U_{X1, R}(2X_2 - 1) + 2X_1 + 3X_2 + U_{X1, R} - 4)} \right\rfloor,$$

$$f_W(U_{X_1, W}, U_{X_2, W}, X_1, R) := \left\lfloor \frac{1}{1 + \exp(U_{X1_W}(2X_1 - 1) + (4R - 2) + U_{X_2, W})} \right\rfloor,$$

$$f_Y(U_{X_2, Y}, X_2, R, W) := \left\lfloor \frac{1}{1 + \exp(0.5U_{X_2, Y}(2R - 1) - 2X_2 + 2W + U_{X_2, Y} - 2)} \right\rfloor.$$

# E  Project STAR: Estimating Joint Effects of Class Sizes to Academic Outcomes

We applied the proposed estimators to the Project STAR dataset [Krueger and Whitmore, 2001, Schanzenbach, 2006]. Project STAR is an experimental study investigating teacher/student ratios' impact on academic achievement for kindergarten through third-grade students. In the study, students were randomly assigned to three different class sizes: small-size classes, regular classes, and large-size classes. The objective was to evaluate how class size affects academic outcomes [Schanzenbach, 2006]. In our analysis, we used the dataset introduced in the online complement of Stock et al. [2003].

**Project STAR Dataset.** We denote the Project STAR dataset as $D$. The dataset $D$ includes the following information: class size for kindergarten ($X_1$), the academic outcome in kindergarten ($W$), the academic outcome in second grade ($R$), class size for third grade ($X_2$), the academic outcome in the third grade ($Y$), free lunch receiving for kindergarten ($C_1$), gender ($C_2$), ethnicity ($C_3$) and free lunch receiving for the third grade ($C_4$). We will use $\mathbf{C}_1 := (C_1, C_2)$ and $\mathbf{C}_2 := (C_3, C_4)$.

**Assumption on Dataset.** We assume that the SCM $\mathcal{M}$ generating the variables $(X_1, W, R, X_2, Y, \mathbf{C}_1, \mathbf{C}_2)$ induces a causal graph depicted in Figure E.4a. Here, we mark $\mathbf{C}_1, \mathbf{C}_2$ as gray to denote that these variables will be considered latent; i.e., these variables will not be used in the data analysis.

Project STAR dataset $D$ is a longitudinal experimental study randomizing $X_1$ and $X_2$; i.e., the dataset is induced by the submodel $M_{x_1, x_2}$ for $x_1, x_2 \in \mathfrak{D}_{X_1, X_2}$, represented in Fig. E.4b. The samples for variables $\{X_1, W\}$ follow a distribution $P_{\sigma(X_1)}(X_1, W, R) = P_{\sigma(X_1, X_2)}(X_1, W, R)$, and the samples for variables $\{X_1, W, R, X_2, Y\}$ follow a distribution $P_{\sigma(X_1, X_2)}(X_1, W, R, X_2, Y)$. To demonstrate, we will describe how to generate the sample fol-

lowing a distribution $P_{\sigma(X_2)}(X_1, W, R, X_2, Y)$ by creating unmeasured confounding bias between $X_1$ and $W$, and $X_1$ and $Y$, which is depicted by Fig. E.4c.

**Creation of Datasets from Marginal Experiments.** In this empirical study, we create two datasets from this dataset: $D_1$ and $D_2$. The dataset $D_1$ is a random subsample of $D$ only including $\{X_1, W\}$. Then, $D_1$ follows $P_{\sigma(X_1)}(X_1, W, R)$.

To construct the dataset $D_2$ following the marginal experimental distribution $P_{\sigma(X_2)}(X_1, W, R, X_2, Y)$, the confounding bias between $X_1$ and $W$ and $X_1$ and $Y$ should be introduced. To do so, we follow a standard procedure for introducing confounding bias from experimental studies used in Hill [2011], Louizos et al. [2017], Zhang and Bareinboim [2019], Gentzel et al. [2021].

A setting for the standard procedure is the following. For any arbitrary random variable $X, Y, Z, W$ such that $X \to Y$, $Z \to Y$, and there are no arrows into $X$, the dataset $D := \{(X_{(i)}, Y_{(i)}, Z_{(i)}, W_{(i)} : i = 1, \cdots, n\}$ is given. In $D$, $Z$ is not a confounding variable since $Z \perp\!\!\!\perp X$. The goal is to generate a new dataset $D' := \{(X_{(j)}, Y_{(j)}, Z_{(j)} : j = 1, \cdots, n'\}$ where $Z$ serves as a confounding variable between $X$ and $Y$. The procedure named $\texttt{IntroduceConfounding}(X, Y; Z, D)$ is given as follows: Initialize $D' = \{\}$. For $i = 1, \cdots, n$, do the followings:

1. Generate the Bernoulli Random $B_{(i)}$ with parameter $P(X_{(i)}|Z_{(i)})$.
2. If $B_{(i)} = 1$, include $(X_{(i)}, Y_{(i)}, Z_{(i)}, W_{(i)})$ in $D'$.

Finally, we exclude $Z$ is removed from $D'$. By doing so, we introduce unmeasured confounding bias between $X$ and $Y$ in $D'$; i.e., $D' = \texttt{IntroduceConfounding}(X, Y; Z, D)$.

To generate the dataset $D_2 \sim P_{\sigma(X_2)}(X_1, W, R, X_2, Y)$ from $D \sim P_{\sigma(X_1, X_2)}(X_1, W, R, X_2, Y, \mathbf{C}_1, \mathbf{C}_2)$, we have to introduce the unmeasured confounding bias between $X_1$ and $W$, and $X_1$ and $Y$. We do this by $D_2' = \texttt{IntroduceConfounding}(X_1, W; \mathbf{C}_1, D)$. Then, $D_2'$ is a variable containing $(X_1, W, R, X_2, \mathbf{C}_2, Y)$ and there is a confounding bias between $X_1$ and $W$. Then, we set $D_2 = \texttt{IntroduceConfounding}(X_1, Y; \mathbf{C}_2, D_2')$. Then, $D_2$ is a variable containing $(X_1, W, R, X_2, Y)$ and there is a confounding bias between $X_1$ and $Y$, and $X_1$ and $W$. A causal graph Fig. E.4c depicted the dependencies in $D_2$.

**Goal.** In this empirical study, we aim to study the joint effect of the class size for kindergarten ($X_1$) and the third grade ($X_2$) on the third grade's academic outcome ($Y$); i.e., $\mathbb{E}[Y|do(x_1, x_2)]$. Since $D$ is a longitudinal experimental dataset following $P_{\text{rand}(X_1, X_2)}(C, X_1, W, X_2, Y)$, the ground-truth $\mathbb{E}[Y|do(x_1, x_2)]$ is estimated as $\mathbb{E}_D[Y \mathbb{1}_{x_1, x_2}(X_1, X_2)] / \mathbb{E}_D[\mathbb{1}_{x_1, x_2}(X_1, X_2)]$.

**Causal Effect Identification.** We identify $P(y|do(x_1, x_2))$ through Algo. 1.

1. **Line 3**: Set $\mathbf{D} := an(Y)_{G(\mathbf{V} \setminus \{X_1, X_2\})} = \{Y, R, W\}$.
2. **Line 4**: Set $\mathbf{D}_1 := \{R\}, \mathbf{D}_2 := \{W\}, \mathbf{D}_3 := \{Y\}$.

Each $Q[\mathbf{D}_1] = Q[W], Q[\mathbf{D}_2] = Q[R], Q[\mathbf{D}_3] = Q[Y]$ are identified as follows: For $Q[\mathbf{D}_1]$,

1. **Line 7**: For $\mathbf{D}_1, \mathbf{Z}_1 := \{X_1\}, \mathbf{z}_1 := \{x_1\}, \mathbf{S}_1^1 := \{W\}$.
2. **Line 8**: Set $Q[\mathbf{S}_1^1] = A_0[\mathbf{S}_1^1, \emptyset, \mathbf{R}_1^1; \mathbb{Z}_1^1 = \{X_1\}, \texttt{seq}_1^1 = (x_1)](w, x_1)$ where $\mathbf{R}_1^1 := \emptyset$.
3. **Line a.4**: Since $\mathbf{S}_1^1 = \mathbf{D}_1$, we set

$$
\begin{aligned}
Q[\mathbf{D}_1] &= Q[\mathbf{S}_1^1] \\
&= A_0^W \\
&:= A_0[W, \emptyset, \emptyset; \mathbb{Z}_1^1 = \{X_1\}, \texttt{seq}_1^1 = (x_1)](w, \emptyset) \\
&= P_{\sigma(X_1)}(w|x_1) = P_{x_1}(w).
\end{aligned}
$$

For $Q[\mathbf{D}_2] = Q[R]$,

1. **Line 7**: For $\mathbf{D}_2$, $\mathbf{Z}_2 := \{X_1\}$, $\mathbf{z}_2 := \{x_1\}$, $\mathbf{S}_2^1 := \{R\}$.
2. **Line 8**: Set $Q[\mathbf{S}_2^1] = A_0[\mathbf{S}_2^1, \emptyset, \mathbf{R}_2^1; \mathbb{Z}_2^1 = \{X_1\}, \mathtt{seq}_2^1 = (x_1)](r, x_1)$ where $\mathbf{R}_1^1 := \{W\}$.
3. **Line a.4**: Since $\mathbf{S}_2^1 = \mathbf{D}_2$, we set

$$
\begin{aligned}
Q[\mathbf{D}_2] &= Q[\mathbf{S}_2^1] \\
&= A_0^R \\
&:= A_0[\mathbf{S}_2^1, \emptyset, \{W\}; \mathbb{Z}_2^1 = \{X_1\}, \mathtt{seq}_2^1 = (x_1)](w, r) \\
&= P_{\sigma(X_1)}(r | r, x_1) = P_{x_1}(r | w).
\end{aligned}
$$

For $Q[\mathbf{D}_3] = Q[Y]$,

1. **Line 7**: For $\mathbf{D}_3$, $\mathbf{Z}_3 := \{X_2\}$, $\mathbf{z}_3 := \{x_2\}$, $\mathbf{S}_1^2 := \{Y, X_1, W\}$.
2. **Line 8**: Set $Q[\mathbf{S}_1^2] = A_0[\mathbf{S}_1^2, \emptyset, \mathbf{R}_1^2; \mathbb{Z}_1^2 = \{X_2\}, \mathtt{seq}_1^2 = (x_2)](y, r)$ where $\mathbf{R}_1^2 := \{W\}$; i.e.,

$$
\begin{aligned}
Q[\mathbf{S}_1^2] &= A_0[\{X_1, W, Y\}, \emptyset, \{R\}; \mathbb{Z}_1^2 = \{X_2\}, \mathtt{seq}_1^2 = (x_2)]((x_1, w, y), r) \\
&= P_{\sigma(X_2)}(y | x_1, w, r, x_2) P_{\sigma(X_2)}(x_1, w | x_2).
\end{aligned}
$$

3. **Line a.2**: Set $\mathbf{A} := an(\mathbf{D}_3)_{G(\mathbf{S}_1^2)} = \{Y\}$.
4. **Line a.3**: $Q[\mathbf{A}] = A_0[\{Y\}, \{X_1, W\}, \{R\}; \mathbb{Z}_1^2 = \{X_2\}, \mathtt{seq}_1^2 = (x_2)](y, r)$.
5. **Line a.4**: Since $\mathbf{A} = \mathbf{D}_2$, we set

$$
\begin{aligned}
Q[\mathbf{D}_2] &= Q[\mathbf{A}] \\
&= A_0^Y \\
&:= A_0[\{Y\}, \{X_1, W\}, \{R\}; \mathbb{Z}_1^2 = \{X_2\}, \mathtt{seq}_1^2 = (x_2)](y, r) \qquad \text{(E.1)} \\
&= \sum_{x_1', w \in \mathfrak{D}_{X_1, W}} P_{\sigma(X_2)}(y | x_1', w, r, x_2) P_{\sigma(X_2)}(x_1', w | x_2).
\end{aligned}
$$

Then, by **Line 14**,

$$
\begin{aligned}
P(y | do(x_1, x_2)) &= \sum_{r, w \in \mathfrak{D}_{R, W}} Q[W] Q[R] Q[Y] \\
&= \sum_{r, w \in \mathfrak{D}_{R, W}} A_0^W A_0^R A_0^Y.
\end{aligned}
$$

By Lemma 3,

$$
\begin{aligned}
A_0^{WR} &:= A_0^W A_0^R \\
&= A_0[\{R, W\}, \emptyset, \emptyset; \mathbb{Z}_2^1 = \{X_1\}, \mathtt{seq}_2^1 = (x_1, x_1)]((w, r), \emptyset) \\
&= P_{\sigma(X_1)}(w, r | x_1) = P_{x_1}(w, r).
\end{aligned}
$$

Therefore,

$$
\begin{aligned}
P(y | do(x_1, x_2)) &= \sum_{r, w \in \mathfrak{D}_{R, W}} A_0^W A_0^R A_0^Y \\
&= \sum_{r, w \in \mathfrak{D}_{R, W}} A_0^{WR} A_0^Y \\
&= \sum_{r \in \mathfrak{D}_R} \Big( \sum_{w \in \mathfrak{D}_W} A_0^{WR} \Big) A_0^Y,
\end{aligned}
$$

where the last equation holds since $A_0^Y$ is not a function of $W$. By Lemma 2

$$
A_0^{R'} := \sum_{w \in \mathfrak{D}_W} A_0^{WR} = A_0[R, \emptyset, \emptyset; \{X_1\}, (x_1)](r, \emptyset) = P_{\sigma(X_1)}(r | x_1) = P_{x_1}(r).
$$

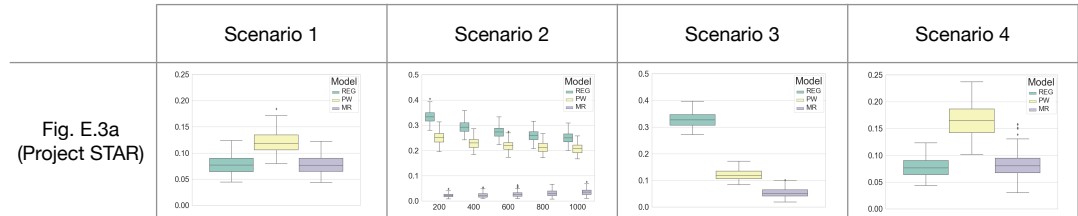

Figure E.5: AAE Plot for Project STAR dataset analysis for Scenarios {1,2,3,4}.

Therefore,

$$P(y|do(x_1, x_2)) = \sum_{r \in \mathfrak{D}_R} (\sum_{w \in \mathfrak{D}_W} A_0^{WR}) A_0^Y$$

$$= \sum_{r \in \mathfrak{D}_R} A_0^{R'} A_0^Y.$$

**Causal Effect Estimation.** Here, we only describe the nuisance for $A_0^Y$ in Eq. (E.1), since estimating $A_0^{R'} = P_{x_1}(r)$ is trivial. We define the nuisance as follows: For the fixed $x_2 \in \mathfrak{D}_{X_2}$,

$$\mu_0(X_1, W, R) := \mathbb{E}_{P_{x_2}}[Y|X_1, W_1, C], \tag{E.2}$$

$$\pi_0(R|X_1, W) := \frac{\mathbb{1}_r(R)}{P_{x_2}(R|X_1, W)}. \tag{E.3}$$

Then, $A_0^Y$ in Eq. (E.1) can be expressed as follows:

$$\text{Eq. (E.1)} \tag{E.4}$$

$$= \mathbb{E}_{P_{x_2}}\left[Y \frac{\mathbb{1}_r(R)}{\pi_0(R|X_1, W)}\right], \text{ or }, \tag{E.5}$$

$$= \mathbb{E}_{P_{x_2}}[\mu_0(X_1, W, r)], \text{ or }, \tag{E.6}$$

$$= \mathbb{E}_{P_{x_2}}\left[\frac{\mathbb{1}_r(R)}{\pi_0(R|X_1, W)}\{Y - \mu_0(X_1, W, R)\} + \mu_0(X_1, W, r)\right]. \tag{E.7}$$

We then construct the regression-based, probability weighting-based, and MR-gID (MR) $T^{\text{reg}}, T^{\text{pw}}, T^{\text{mr}}$ using the following procedure.

1. For each fixed $x_2 \in \mathfrak{D}_{X_2}$ and a sample set $D_{x_2}$ for $i \in \{1, 2\}$, randomly split the sample as $D_{x_2,t}$ and $D_{x_2,e}$.

2. Use $D_{x_2,t}$ to train the model for learning nuisances in Eq. (E.2) and Eq. (E.3). Let $\mu(X_1, W, R)$ and $\pi(R|X_1, W)$ denote the learnt models. We use the XGBoost [Chen and Guestrin, 2016] to learn the model.

3. Then, each estimator is defined as follows:

$$T^{\text{reg}} := \mathbb{E}_{D_{x_2,e}}[\mu(X_1, W, r)] \tag{E.8}$$

$$T^{\text{pw}} := \mathbb{E}_{D_{x_2,e}}\left[\frac{\mathbb{1}_r(R)}{\pi(R|X_1, W)}Y\right] \tag{E.9}$$

$$T^{\text{mr}} := \mathbb{E}_{D_{x_2,e}}\left[\frac{\mathbb{1}_r(R)}{\pi(R|X_1, W)}\{Y - \mu(X_1, W, R)\}\right] + \mathbb{E}_{D_{x_2,e}}[\mu(X_1, W, r)]. \tag{E.10}$$

**Experimental Results** As described in the Experimental Setup section (Sec. 4), we evaluated the $\text{AAE}^{\text{est}}$ of estimators $T^{\text{est}}$ for $\text{est} \in \{\text{reg}, \text{pw}, \text{mr}\}$ in Scenarios $\{1, 2, 3, 4\}$. The AAE plots for all

cases can be seen in Fig. E.5. In this particular scenario, the sample size was not varied since the sample itself was externally given.

In Case 2, we introduced variation by adjusting the size of the converging noise $\epsilon$, which follows a normal distribution $\text{Normal}(n^{-\alpha}, n^{-2\alpha})$ for $n \in \{200, 400, 600, 800, 1000\}$. It was observed that the MR-gID estimator $T^{\text{mr}}$ outperformed the other two estimators by achieving fast convergence, as demonstrated in Theorem 2. For Scenarios {3, 4}, the DML estimator $T^{\text{mr}}$ exhibited doubly robust properties, as illustrated in Corollary 2.

