# OpenReview forum: "Estimating Causal Effects Identifiable from a Combination of Observations and Experiments"
_NeurIPS.cc/2023/Conference — NeurIPS 2023 poster_

### Official Review · Reviewer_8jnJ · 2023-07-04

**Soundness:** 4 excellent
**Presentation:** 1 poor
**Contribution:** 4 excellent
**Rating:** 6
**Confidence:** 3

**Summary:**

The paper presents a method for causal effect estimation through a combination of experimental and observational data.

**Strengths:**

- The paper addresses a very important problem
- The solution appears to be mathematically sound
- Experimental results appear to show improvement over other methods

**Weaknesses:**

- The paper is incredibly  hard to parse. Pages 4-8 are just a wall of mathematical notation with little to no explanation.
- The paper would improve significantly if after each definition / lemma / proof a plain word intuition was given.
- Running an example through an algorithm in the manner its done in page 6 is borderline incomprehensible.
- Figure 2 is too small to read even when pdf is zoomed

  Honestly this is a great paper that is severely limited by the way its presented.

**Questions:**

No real questions just a request that the paper is rewritten to increase clarity and improve presentation

**Limitations:**

Little discussion is included about the limitations of the method

---

> ### Author Rebuttal · Authors · 2023-08-09
>
> We appreciate your feedback, thank you.
>
> ---
>
> > W1. The paper is incredibly hard to parse. Pages 4-8 are just a wall of mathematical notation with little to no explanation.
>
> > W2. The paper would improve significantly if after each definition / lemma / proof a plain word intuition was given.
>
> Given the intricate and technical nature of this topic that spans multiple disciplines (theories of causal effect identification and semiparametric inference), presenting the necessary background in a digestible manner is quite challenging. Given the constraints on the length of NeurIPS papers, we have strived to strike a balance, emphasizing our novel contribution and demonstrating it in a condensed manner. We recognize the density of our presentation and apologize for it. We plan to enhance the clarity of our results by adding more verbal explanations for our technical results.
>
> ---
>
> > W3. Running an example through an algorithm in the manner its done in page 6 is borderline incomprehensible.
>
> We would like to note that a detailed step-by-step procedure for following Algorithm 1 is provided in Appendix A: example 3 in Appendix A.1 and Example 4 in Appendix A.2.
>
> ---
>
> > W4. Figure 2 is too small to read even when pdf is zoomed
>
> It was an oversight and we apologize for that. We included the enlarged figure in the 1-page attached pdf in the general response above. We will incorporate the same in the updated version of the manuscript, thank you for bringing this up.

---

> > ### Comment · Reviewer_8jnJ · 2023-08-11
> >
> > I have read the authors rebuttal and I will maintain my score of acceptance.
> > I appreciate the argument regarding the page limits of the conference, however a greater effort could and should be given in presenting the methods, increasing, thus, the potential impact of the work.
> > As I've said in my review, I believe this is a good paper that could have been much much better if it was presented in the right way.

---

> > > ### Author Response · Authors · 2023-08-11
> > > **Response to Reviewer 8jnJ**
> > >
> > > Thank you for your response and positive assessment of our paper. We agree and, as promised, will improve the presentation of our results to enhance understanding and potential impact.

---

### Official Review · Reviewer_sTQB · 2023-07-04

**Soundness:** 3 good
**Presentation:** 3 good
**Contribution:** 3 good
**Rating:** 6
**Confidence:** 2

**Summary:**

This paper presents a new framework for estimating the causal effect when multiple observational and experimental datasets are available. The authors proved that any g-identifiable causal effects can be written as a functional of "g-mSBD" operators, based on which they further developed a doubly robust estimator. Asymptotic properties of this estimator are then analyzed, and the performance of this estimator is confirmed in simulation studies.

**Strengths:**

This paper tackles a hard problem with theoretical soundness and practical applicability.

**Weaknesses:**

I have two questions/comments.

The first regards the assumption imposed in their Proposition 2 as well as Assumption 2 -- the authors assumed the nuance estimates are L2-consistent. While I understand that such a property would make the theoretical analysis more amenable, I am curious and would appreciate it if the authors can provide a concrete (and maybe idealized) example, under which the assumption can be verified.

My second question is about practical performance of the proposed estimator. While I am convinced about its utility in the simulation study, it would be great if the authors can apply their methodology in a real dataset and verify it still outperforms its competitors.


**Questions:**

Please see the "Weakness" section.

**Limitations:**

Please see the "Weakness" section.

---

> ### Author Rebuttal · Authors · 2023-08-09
>
> We thank the reviewer for the valuable feedback provided and hope that the answers below help to address the raised issues. Still, we would be happy to provide further clarification if you find that this was not the case.
>
> ---
>
> > W1. The first regards the assumption imposed in their Proposition 2 as well as Assumption 2 -- the authors assumed the nuance estimates are L2-consistent. While I understand that such a property would make the theoretical analysis more amenable, I am curious and would appreciate it if the authors can provide a concrete (and maybe idealized) example, under which the assumption can be verified.
>
> Good point. This assumption is plausible in practice in the era of modern machine learning and deep neural networks. Thanks to the flexibility of these models (e.g., the universal approximation theorem of neural networks), it is plausible that the estimates will converge to the true nuisance whenever the sample size grows. Specifically, a recent study reveals that if the true nuisance function satisfies some mild smoothness assumptions, then the deep neural network model-based estimators will be L2-consistent [1].
>
> [1] Farrell, Max H., Tengyuan Liang, and Sanjog Misra. "Deep neural networks for estimation and inference." Econometrica 89.1 (2021): 181-213.
>
> ---
>
> > W2. My second question is about practical performance of the proposed estimator. While I am convinced about its utility in the simulation study, it would be great if the authors can apply their methodology in a real dataset and verify it still outperforms its competitors.
>
> Sure. We will prepare a real-world study in the revised revision.

---

> > ### Comment · Reviewer_sTQB · 2023-08-13
> >
> > Thanks the authors for the clarification of my questions. I would like to keep my score.

---

### Official Review · Reviewer_t8rU · 2023-07-05

**Soundness:** 3 good
**Presentation:** 3 good
**Contribution:** 2 fair
**Rating:** 5
**Confidence:** 4

**Summary:**

The authors study a class of causal effect that are called “g-identifiable” (gID) following Lee et al. (UAI 2019). There are two main results.
1. Expressing these causal effects as transformations of generalized multi outcome sequential back door adjustments (g-mSBDs).
2. Proposing doubly robust estimators for the adjustments, which then imply estimators for the causal effects if the transformations are smooth.

It appears that Jung et al (ICML 2023) previously provide these types of results for a subset of gID causal effects. The contribution is to extend the previous results for the full class of gID causal effects.

**Strengths:**

1. Originality: The first contribution is, in my view, the stronger one. Its connection to prior work is clearly stated. It appears that the workhorse is Lemmas 2, 3, and 4, which demonstrate the good properties of Definition 1. The second contribution follows quite closely from prior work.

2. Quality: The introduction was generally well written, though it would make more sense to choose Examples 1 and 2 to be examples that are not in the class of Jung et al (ICML 2023).

3. Clarity: Overall, the organization of the paper was very logical.

4. Significance: The contribution seems sound but possibly incremental.

**Weaknesses:**

I would be willing to improve the score if these weaknesses are addressed by the authors.

1. I find the title and framing too broad. There are many papers that could have this title, and it does not clearly convey what is in this paper but not in other papers.

2. Line 56 gives a list of doubly robust estimators. At the very least, the sequential regression paper by Bang and Robins (Biometrics 2005), the targeted learning in data science textbook (van der Laan and Rose 2018), and the debiased machine learning paper by Chernozhukov et al. (2018 Econometrics Journal) should be cited.

3. More relevant are the sequential doubly robust estimators such as Luedtke et al. (arXiv:1705.02459, 2017) and Chernozhukov et al. (arXiv:2203.13887, 2022) and the references in those papers. Definition 3 and Proposition 1, are very closely related to the prior works listed above, so explicit comparisons would be helpful here.

4. The delineation between Lee et al. (2019)’s algorithm and this algorithm should be clearer. The sentences in lines 127 and 156 seem to clash in explaining the relationship to the prior work.

5. The notation in Definition 1, Lemma 1, Lemma 2, and Lemma 3 is very heavy, and no interpretation is assigned to any of the variable symbols. It would help to revisit Examples 1 and 2 here.

6. In the appendix, the data generating processes are not described analytically; instead, python code is pasted. Please describe them. Please explicitly write what are the functions mu and pi.

**Questions:**

I would be willing to improve the score if these questions are addressed by the authors.

1. Algorithm 1 is a modification of Lee et al. (2019)’s complete algorithm. Is it the case that Algorithm 1 is precisely the previous algorithm but with Lemmas 2, 3 and 4? Does it remain complete?

2. Since the transformation f is shown to be marginalization, multiplication, or division, wouldn’t we already know that aspects of Assumption 2 are automatically satisfied? Phrased another way, when is each condition in Assumption 2 binding?

3. Are the proposed estimators semiparametrically efficient?

4. What does scenario 2 mean? Is this measurement error, or is it the usual error of a regression problem?

5. Suggested corrections:
line 8: multiply --> multiple.
line 86: specify norm when writing convergence of f_hat-f when they are functions.
line 127: “(and rewrote in Algo. 1) rewrites” is awkward phrasing.
line 248: Fretchet --> Frechet.

**Limitations:**

Yes

---

> ### Author Rebuttal · Authors · 2023-08-09
>
> Thank you for your feedback and for the opportunity to provide further elaboration. We respectfully ask the reviewer to reconsider our paper based on the clarifications provided below. If there are any points that remain unclear, we would be glad to offer additional explanations.
>
> ---
>
> > Weakness 1
>
> The current title, "Estimating Causal Effects Identifiable from Combination of Observations and Experiments," reflects what we believe is the broad scope of the paper. In particular, we introduce the first family of estimators designed for (1) any mix of observational and experimental distributions and (2) any given causal diagram, extending beyond the back-door case. Our aim was to convey comprehensiveness. However, based on your feedback, we are contemplating titles like "Estimating G-identifiable Causal Effects." The term "G-identifiable" highlights the paper's focus on the flexibility of combining observational and experimental distributions. To the best of our knowledge, no other paper can estimate such a diverse coverage of effects, input distributions, and diagrams. Technically, our results match the completeness results on the identification side, which means that any effect that is in fact 'g-identifiable' is also 'estimable' via our proposed method (Theorems 1 and 2). While there may be other papers that claim generality, we are not aware of any research covering all g-identifiable effects. Having said that, we are certainly open to any suggestions you might have regarding a more apt title for our paper.
>
> ---
>
> > Weakness {2,3}
>
> Indeed, this was an oversight, and while one citation is in the appendix, we will consolidate them and cite all the referred papers. Nice catch, thank you.
>
> ---
>
> > Question 1 and Weakness 4
>
> Lee's algorithm and our Algorithm 1 are equivalent. This implies that Algorithm 1 also provides a sound and complete procedure for identifying causal effects from a causal graph and the corresponding mix of observational and interventional distributions. The formal proof of this equivalence can be found in the proof of Theorem 1.
>
> The motivation behind Algorithm 1's formulation is to express the g-identifiable causal effect as a combination of marginalizations, multiplications, and divisions of g-mSBD. Unlike Lee's algorithm, Algorithm 1 is formulated in terms of c-factors, which can be expressed as a function of g-mSBDs, as shown in Lemma 1. Since each c-factor is expressible as a g-mSBD, our algorithm has an amenable form that allows for the application of marginalizations, multiplications, and divisions, as required by Lemmas 2, 3, and 4. To align the sentences in lines 127 and 177, we will clearly state this point by replacing the sentence in line 157 as follows:
>
> "We have rewritten the identification algorithm proposed by Lee et al. (2019) as Algorithm 1 to express the g-identifiable causal effect as a combination of marginalizations, multiplications, and divisions of g-mSBD."
>
> ---
>
> > Weakness 5
>
> Thank you for the feedback. We will add an explanation and detailed illustrations with Examples 1 and 2 in the Appendix.
>
> ---
>
> > Weakness 6
>
> We have updated the data generation process analytically based on your feedback. This update will be available in future revisions. Additionally, the exact and explicit representations of mu and pi can be found in Sections A.5 and A.6 of the submitted Appendix.
>
> ---
>
> **We now consider the "Questions". The first question was answered above.**
>
> > Question 2
>
> Thank you for the good question. The estimator $f(\{\hat{A}^{k}\})$ can be infinite if the estimated nuisance $\pi$ for the weight is zero. For example, if $f(\{\hat{A}^{1}, \hat{A}^2\}) := \hat{A}^1 / \hat{A}^2$ and $\hat{A}^2$ is zero, the function $f$ may not be differentiable, and the boundedness in Assumption 2 may not hold. Assumption 2 is a set of conditions excluding such cases and a sufficient condition for representing an upper bound of the error in terms of products of the two sets of nuisances for g-mSBD estimators.
>
> ---
>
> > Question 3
>
> We appreciate the question. The answer is no; the proposed estimators have not been shown to be semiparametric efficient. Let us elaborate further. A semi-Markovian causal graph, characterized by directed and bidirected edges, implies two types of constraints in the probability distribution: conditional independence and Verma constraints (also called Nested Markov factorization). To the best of our knowledge, constructing the tangent space for a semiparametric statistical model that takes these constraints into account remains an unsolved problem. In other words, while we can construct doubly robust estimators if the distribution is compatible with a semi-Markovian model, building a semiparametric efficient estimator is still an open research question.
>
> ---
>
> > Question 4
>
> This simulates sampling error induced by the finiteness of samples, which we believe is what you meant by the usual error in regression. In Scenario 2, the noise was added to the regression estimates, which decay as the sample size grows, to emphasize the errors induced by the finiteness of samples, following the technique outlined in Kennedy's 2020 paper. Specifically, we assumed that the error converges at a rate of N^{-1/4} to highlight the fast convergence of the proposed estimator. Another usual error in a regression problem is a specification error, which is caused by choosing the wrong regression models in estimating nuisances. Scenarios 3 and 4 simulate cases of specification errors. We have incorporated this discussion into the updated manuscript. Thank you.
>
> [Kennedy20] Kennedy, Edward H. "Towards optimal doubly robust estimation of heterogeneous causal effects." arXiv preprint arXiv:2004.14497 (2020).
>
>
> ---
>
> > Question 5
>
> Good catch, thank you! We revised the paper accordingly.

---

> > ### Comment · Reviewer_t8rU · 2023-08-21
> >
> > I do not see a new draft to verify the extent to which these changes were made. Each suggested change is important. Based on the rebuttal alone, I have increased the score.

---

> > > ### Author Response · Authors · 2023-08-21
> > > **Response to Reviewer t8rU**
> > >
> > > Dear Reviewer t8rU, thank you for engaging in the reviewing process; we hope to have addressed all the issues you raised. As we understand the rules this year, authors are not allowed to upload revised versions of their manuscripts during this stage, which is why you were unable to see a new version of our paper. Your comments have been addressed and integrated into the revised version, which will be made available after revision.

---

### Official Review · Reviewer_PJpp · 2023-07-06

**Soundness:** 3 good
**Presentation:** 2 fair
**Contribution:** 3 good
**Rating:** 7
**Confidence:** 3

**Summary:**

The paper aims to fuse the recent literature on causal effect identification and estimation. It expresses any identification (in the general sense) of a causal effect in terms of generalized mSBD adjustments, and then develops a corresponding estimator. For the latter robustness properties and empirical studies are shown.

**Strengths:**

The contribution made by this paper is well-situated within the problem space and it's a natural generalization of other recent papers. All the results appear to be sound.

**Weaknesses:**

I believe the contribution to be valuable but the methods are not novel; they seem to be iterative applications of existing ones.
In many places it's too dense: for example, Examples 3, 4 are not illuminating, and readers may just skip them.

**Questions:**

Can you include a future work section? My perception of the significance of the work would be improved if I could see some directions for how to build on this.

**Limitations:**

Yes, they have (there is ample discussion of e.g. the assumptions relied on).

---

> ### Author Rebuttal · Authors · 2023-08-09
>
> We thank the reviewer for the time and valuable feedback, and appreciate the positive assessment of our work.
>
> ---
>
> > W1. I believe the contribution to be valuable but the methods are not novel; they seem to be iterative applications of existing ones. In many places it's too dense: for example, Examples 3, 4 are not illuminating, and readers may just skip them.
>
>
> Thank you for the opportunity to clarify this issue. One of our primary research findings (as detailed in Theorem 1) reveals that the gID function can be expressed as a function of g-mSBDs. We believe this realization is non-trivial. This insight paved the way for the development of our proposed estimator, into which each DR-g-mSBD estimator is plugged in. It represents a natural and effective estimator built upon our main findings.
>
> Given the intricate nature of this topic, which spans multiple disciplines, and the tight constraints in terms of the number of pages,  presenting the necessary background in a gentle manner is non-trivial.  We apologize for that. Still, detailed explanations of the examples can be found in the Appendix; a step-by-step discussion of Example 3 is presented in Appendix A.1, and Example 4 is elaborated in Appendix A.2.
>
> ---
>
> > Q1. Can you include a future work section? My perception of the significance of the work would be improved if I could see some directions for how to build on this.
>
> Sure. We will discuss potential future directions in the updated manuscript. For now, a summary of possible research directions is as follows:
> 1. Investigate more generalized scenarios that account for the presence of selection bias or the challenges of transportability/generalizability across changing conditions;
>
> 2. Relax the restrictions of this paper, such as the assumption of discrete variables;
>
> 3. Develop a semiparametric efficient estimator;
>
> 4. Construct computationally efficient estimation procedures;
>
> 5. Develop approximated estimators for whenever the effect is not identifiable.

---

> > ### Comment · Reviewer_PJpp · 2023-08-15
> >
> > Thanks, this response is helpful.

---

### Official Review · Reviewer_T9Rz · 2023-07-23

**Soundness:** 3 good
**Presentation:** 2 fair
**Contribution:** 3 good
**Rating:** 7
**Confidence:** 1

**Summary:**

This paper builds on prior work in causal effect estimation, from observational as well as experimental data, by providing a more general framework with desirable properties. This more generalized framework is shown to be robust and consistent in its estimation and overcomes the limitations of some of the prior work in this area. The new, general framework, operating in the realm of generalized identification or g-identification, exhibits multiply robust properties, which is experimentally validated.

**Strengths:**

The new proposed estimator could be now the go-to framework for any g-identifiable causal functionals, potentially representing a significant theoretical advance in the field of learning causal effects.
The paper is well-scoped and specific in what is being advanced and corroborates the desirable properties of the estimator via simulations.
The exposition is quite clear and well-written.

**Weaknesses:**

It is not clear what challenges remain for g-identification in practice with the proposed new estimator. Some guidance for practitioners and other researchers as to the context of problems and types of data they can consider using the new estimator would be highly useful.

**Questions:**

1. Is there any scenarios where Jung et al 2023 is still useful -- i.e., are there complementary notions of use to that framework and the one proposed here, or does the framework here subsumes Jung 2023 completely? Is it possible to compare the two frameworks in a particular setting to empirically demonstrate the relationship between the two frameworks?

2. What practical challenges remain for g-identification when using the proposed estimator?

Suggestions:

- Remove citation from abstract, just let the 'in the literature' stand, and cite instead when first mentioned in the introduction.
- Claim in Line 286 -- "We consider two standard baselines in the literature" --  must be backed by citations, especially the notion of being "standard".

**Limitations:**

It is not clear what are the limitations of this work and the proposed new framework, and a lack of such an elucidation might hurt the uptake of what seems to be a promising new framework. There does not seem to be any discussion of any kind of limitations or potential broader impacts.

---

> ### Author Rebuttal · Authors · 2023-08-09
>
> Thank you for sharing your thoughts and feedback!
>
> ---
>
> > W1. It is not clear what challenges remain for g-identification in practice with the proposed new estimator. Some guidance for practitioners and other researchers as to the context of problems and types of data they can consider using the new estimator would be highly useful.
>
> > Q2. What practical challenges remain for g-identification when using the proposed estimator?
>
> > L1. It is not clear what are the limitations of this work and the proposed new framework, and a lack of such an elucidation might hurt the uptake of what seems to be a promising new framework. There does not seem to be any discussion of any kind of limitations or potential broader impacts.
>
> We appreciate you for raising this issue and thank you for the opportunity to provide further clarification. We stay by the generality of our results and are genuinely excited about them. However, we do not claim that our paper has resolved every issue; there are several challenges that remain for the community to address. First, a critical assumption in our work is that the endogenous (observed) variables are discrete. This assumption limits the generality of our claims. While we offer some preliminary ideas on how to relax this limitation in Section C1 of the Appendix, the issue largely remains unresolved. Second, our proposed estimator is not shown to be semiparametric efficient due to the constraints implied by causal graphs. This is despite its doubly robust nature and fast convergence rates. Addressing these challenges would be highly non-trivial but important extensions of the work proposed in this submission.
>
> ---
>
> > Q1. Is there any scenarios where Jung et al 2023 is still useful -- i.e., are there complementary notions of use to that framework and the one proposed here, or does the framework here subsumes Jung 2023 completely? Is it possible to compare the two frameworks in a particular setting to empirically demonstrate the relationship between the two frameworks?
>
> This work encompasses all the results presented in Jung et al., 2023. To clarify their relationship, note that under the graphical settings and assumptions imposed in Jung et al., 2023 (for example, no unmeasured confounders between outcomes in multiple experiments when input distributions are multiple experimental distributions), the proposed g-mSBD method reduces to Jung et al., 2023.
>
> ---
>
> > Suggestion 1. Remove citation from abstract, just let the 'in the literature' stand, and cite instead when first mentioned in the introduction.
>
> We will revise it accordingly, thank you for the suggestion.
>
> ---
>
> > Suggestion 2. Claim in Line 286 -- "We consider two standard baselines in the literature" -- must be backed by citations, especially the notion of being "standard".
>
> Since the gID estimation problem has not been addressed in prior literature, no existing estimators were available. To enable comparison with our proposed estimator and emphasize its doubly robust properties, we developed ‘baseline estimators’ using regression and probability-weighting methods. That is why this sentence was not backed by the citations. These methods are intuitively conceivable and frequently employed in standard settings where the sole input distribution is observational (hence our reference to them as "standard in the literature"). We appreciate you raising the issue. We would like to note that some discussions explaining the baseline estimators are in Section A5 in Appendix. Thank you for raising this point.

---

> > ### Comment · Reviewer_T9Rz · 2023-08-10
> >
> > Thank you for the response. I officially confirm I have read the authors' response and am satisfied they have addressed my questions and concerns. I am updating and increasing my review score.
> >
> > For authors: the discussion of the challenges/limitations of the introduced method that are elucidated in your response should ideally make it to the paper itself since it would be very valuable for the readers.

---

> > > ### Author Response · Authors · 2023-08-11
> > > **Response to Reviewer T9Rz**
> > >
> > > We appreciate the positive assessment of our responses. We are glad to hear that it addressed your concerns. As you suggested, we will add further discussion reflecting the challenges and limitations of our work to pave a new path for future researchers.

---

### Author Rebuttal · Authors · 2023-08-09

Thank you for your valuable feedback. We responded to each review individually. Here, we uploaded an enlarged plot for our simulation result (Figure 2).

---

### Decision · Program_Chairs · 2023-09-21

**Decision:**

Accept (poster)

**Comment:**

The paper introduces a new framework for estimating causal effects using both experimental and observational data. It extends the class of g-identifiable causal effects and provides a doubly robust estimator for these effects. The theoretical foundations are solid, and the simulation results suggest the effectiveness of the proposed method. However, there are concerns about the presentation and clarity of the paper, and there's a need for more practical application and discussion of limitations. The paper would benefit from addressing these comments.